# Direct measurements of interfacial adhesion in 2D materials and van der Waals heterostructures in ambient air

Hossein Rokni[1] & Wei Lu [1,2✉]

Interfacial adhesion energy is a fundamental property of two-dimensional (2D) layered materials and van der Waals heterostructures due to their intrinsic ultrahigh surface to volume ratio, making adhesion forces very strong in many processes related to fabrication, integration and performance of devices incorporating 2D crystals. However, direct quantitative characterization of adhesion behavior of fresh and aged homo/heterointerfaces at nanoscale has remained elusive. Here, we use an atomic force microscopy technique to report precise adhesion measurements in ambient air through well-defined interactions of tip-attached 2D crystal nanomesas with 2D crystal and $SiO_x$ substrates. We quantify how different levels of short-range dispersive and long-range electrostatic interactions respond to airborne contaminants and humidity upon thermal annealing. We show that a simple but very effective precooling treatment can protect 2D crystal substrates against the airborne contaminants and thus boost the adhesion level at the interface of similar and dissimilar van der Waals heterostructures. Our combined experimental and computational analysis also reveals a distinctive interfacial behavior in transition metal dichalcogenides and graphite/$SiO_x$ heterostructures beyond the widely accepted van der Waals interaction.

[1] Department of Mechanical Engineering, University of Michigan, Ann Arbor, MI 48109, USA. [2] Department of Materials Science and Engineering, University of Michigan, Ann Arbor, MI 48109, USA. ✉email: weilu@umich.edu

Two-dimensional layered materials (2DLMs), such as graphene, hexagonal boron nitride (hBN), transition metal dichalcogenides (TMDs: e.g., $MoS_2$ and $WS_2$), and many others, with strong in-plane covalent bonding and weak out-of-plane van der Waals (vdW) interactions exhibit a unique combination of high elasticity, high mechanical flexibility, visual transparency, and superior (opto)electronic performance, making them ideally suited to modern devices, such as photovoltaic devices, hybrid electrochemical capacitors, lithium- and sodium-ion batteries, hydrogen evolution catalysis, transistors, photodetectors, DNA detection, and memory devices[1]. However, intrinsic ultrahigh surface-to-volume ratio in 2DLMs requires intimate knowledge of interfacial adhesion between two adjacent layers of similar or dissimilar 2DLMs and also between 2DLMs and their supporting substrate. In particular, such a fundamental mechanical property plays a central role not only in synthesis, transfer, and manipulation of 2DLMs but also in fabrication, integration, and performance of 2DLM-incorporated devices.

In general, fabrication of 2D systems involves transferring 2DLMs from their growth substrate or bulk stamp onto a target substrate using different transfer-printing techniques. A better understanding of the adhesion energy between 2DLMs and the various substrates involved is highly desired as an essential step toward enhancing the transfer efficiency and thickness uniformity of printed flakes and thus producing high-quality, large-scale 2D electronic device arrays at microscale and nanoscale[2]. The interfacial adhesion between 2DLMs and their neighbors is also an important parameter for the mechanical integrity of the device whose operation is highly influenced by slippage and delamination of 2DLMs during thermal and mechanical loadings. As such, a 2DLM needs to make secure contact not only with supporting substrates and metallic interconnects in 2DLM-based devices but also with other 2DLMs and encapsulation layers in vdW heterostructure devices[3,4].

Newly emerged vdW heterostructures—stacks of individual monolayer flakes of different 2DLMs assembled layer by layer—offer a variety of new physical properties, thanks to the full spectrum of electronic properties in 2DLMs, from conducting graphene to semiconducting TMDs, to insulating hBN. An essential feature of such heterostructures is atomically clean interfaces to achieve the best device performance—any interfacial contamination (e.g., blisters) results in deterioration of transport properties[5]. As such, wet transfer and dry pick-and-lift transfer techniques are widely used for assembly of vdW heterostructures. However, both direct mechanical assembly techniques rely strongly on vdW interactions between the 2D crystals, and as a result, an accurate quantification of interfacial adhesion between different 2DLMs is crucial for the mass production of blister-free vdW hetersostructures.

Fascinating interlayer vdW-dependent properties of similar and dissimilar 2DLMs provide a unique opportunity to study the nature of electronic structure and band alignment, interfacial thermal and electrical resistance, ion intercalation and deintercalation process, interfacial nanofluidic transport and drug delivery behavior, photon absorption and photocurrent/photovoltaic production, interfacial charge polarization and redistribution, spin–orbit coupling, and many others in layered material-based devices[6–10]. Notably, interfacial electrical, mechanical, optoelectronic, magnetic, and thermal properties of layered materials can also interact in a rather complex way. For instance, formation of any delamination-motivated surface corrugations in 2DLMs can give rise to local strain distribution and curvature-induced rehybridization, which modify the electronic structure and local charge distribution; create polarized carrier puddles and dipole moment; induce pseudomagnetic fields; and thus alter magnetic, optical, and electrical properties as well as chemical surface reactivity[11]. Moreover, the vdW interaction as a key medium for the stress transfer both within and across the interface of 2DLMs can highly impact their thermal and electrical properties in such a way that a 2D layered system can act as a heat conductor or insulator and/or a semimetal or electrical insulator through strain engineering[12–14].

The interfacial physical and chemical behavior of layered materials becomes even more complicated when we consider that airborne contaminants are an inevitable part of any vdW heterostructures, and therefore addressing quantitatively to what degree their interfacial adhesion energy (IAE) is influenced by interfacial contaminants and nanoblisters and how to effectively remove them is of fundamental and technological importance for the continued development of such promising materials. However, many attempts have been made over the past six decades to measure the IAE of 2D crystals either in high vacuum or under a contamination-free environment. Among them, few direct IAE measurements of 2D crystals have been reported with a particular focus on graphite (G) crystal[15–21]. For instance, the IAE at the intact G/G homointerface was reported using micro-force sensing probe measurements on 4-μm-wide square mesas ($0.37 \pm 0.01\ \mathrm{J\,m^{-2}}$, ref. [15]) and atomic force microscopy (AFM)-assisted shearing measurements on 3-μm-wide square mesas ($0.35\ \mathrm{J\,m^{-2}}$, ref. [16]) and circular mesas of 100–600 nm in diameter ($0.227 \pm 0.005\ \mathrm{J\,m^{-2}}$, ref. [17]). Moreover, there is only one measured IAE value of $0.22\ \mathrm{J\,m^{-2}}$ at the $MoS_2/MoS_2$ homointerface using a nanomechanical cleavage technique[18], whereas, to the best of our knowledge, no IAE measurement at the hBN/hBN homointerface yet exists. Also, the vdW interaction at G/hBN and G/$MoS_2$ heterointerfaces was studied using a G-wrapped sharp tip with an unknown contact area, allowing the measurements of critical adhesion forces between G/G, G/hBN, and G/$MoS_2$ in high vacuum at room temperature[19]. Although a considerable number of experimental and theoretical methods have been proposed to study the IAE of 2D crystals in general and G crystal specifically, there is significant diversity in the reported IAE values, where the exact cause of the variation in their IAE values has still remained to be elucidated by a comprehensive and accurate experimental technique.

In this work, the first direct nanoscale quantification of IAE at both fresh and aged 2D vdW homointerfaces/heterointerfaces is performed under different annealing temperatures. To this end, force–displacement ($F$–$d$) curves with piconewton–subnanometer resolution are recorded upon retraction of AFM tip-attached 2D crystal nanomesas from tens-of-nm-thick fresh and aged 2D crystal substrates and bare $SiO_x$/Si substrate under controlled ambient conditions in the near equilibrium regime (Fig. 1a). The annealing temperature of nanocontact interfaces is precisely controlled in the range of −15–300 °C by a microheater on the top (left inset of Fig. 1a) and a cooling stage underneath the $SiO_x$/Si substrate. Aged substrates are prepared by two different aging conditions where the freshly exfoliated 2D crystal and bare $SiO_x$/Si substrates are either exposed to the ambient air directly (hereafter referred to simply as untreated substrates) or kept at subzero temperature, followed by the air exposure (referred to as precooling-treated substrates). Among many different combinations of dissimilar 2DLMs, we focus on the interlayer vdW behavior of conducting G, insulating hBN, and semiconducting $MoS_2$ crystals as a model system for a large class of 2D vdW heterostructure systems. Since direct nanoscale probing of weak vdW interactions in 2DLMs and vdW heterostructures requires a unique combination of high-resolution imaging, precise mechanical manipulation, and accurate in situ interfacial adhesion measurements, nano-sized square (circular) 2D crystal mesas with a width (diameter) of 55–65 nm are attached to an in situ flattened AFM tip, which is precoated with an ultrathin adhesive

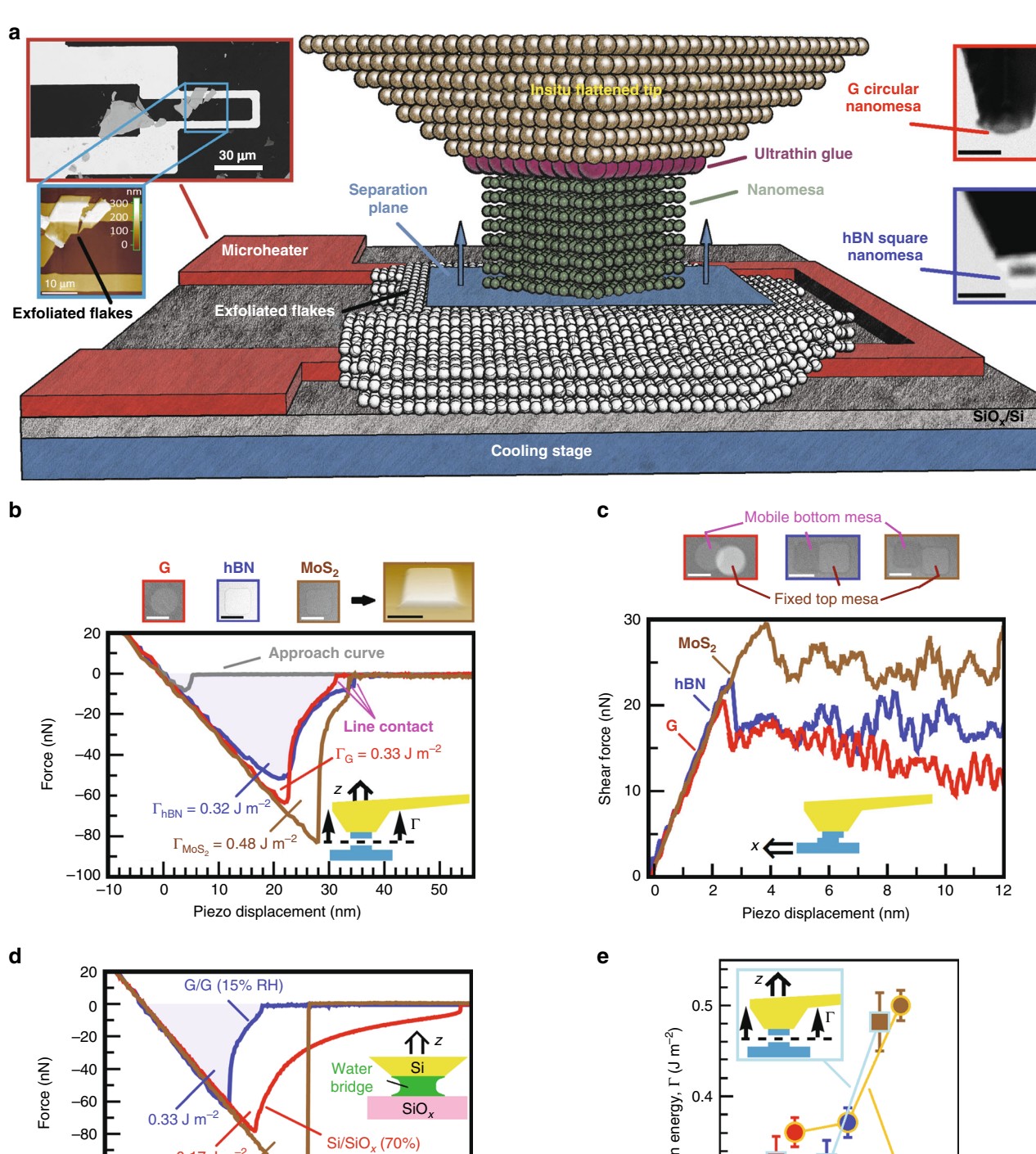

polymer at the apex (right inset of Fig. 1a and Supplementary Note 1). Using nano-sized 2D crystal tips with a very well-defined geometric shape parallel to the substrate together with accurate determination of spring constant of the probe is an essential prerequisite for the detailed characterization of nanoscale vdW interfaces and the accurate extraction of interfacial adhesion properties of 2DLMs (Supplementary Note 2).

## Results

**Cohesion energy in 2D crystals.** During the attachment of nanomesas to the glue-coated tip, $F$–$d$ curves can be recorded as the tip is gently pulled away from the substrate surface in a direction perpendicular to the single basal plane of 2D crystal, leading to pulling off the upper section of the nanomesa (attached to the tip apex) from the lower section (fixed to the 2D crystal

**Fig. 1 AFM-assisted experimental set-up and cohesion energy measurements. a** Schematic of the AFM set-up used to perform interfacial adhesion measurements under different annealing temperatures. Left inset: SEM image of the microheater with an $MoS_2$ flake exfoliated on the heating line whose corresponding AFM image was taken by the G crystal tip. Right Inset: SEM images of the tip-attached G circular nanomesa (top image) and hBN square nanomesa (bottom image). Scale bars indicate 100 nm. **b** Typical retraction $F–d$ curves recorded at the intact homointerface of G (in red), hBN (in blue), and $MoS_2$ (in brown) crystals and also the approach $F–d$ curve (in gray) recorded at the hBN tip/substrate homointerface. The light blue-shaded area under the retraction curve at the hBN homointerface represents the cohesion energy in units of Joules. Each raw data set was given an offset to provide the same equilibrium position for all $F–d$ curves. Top panel: SEM images of lower section of the nanomesas on their corresponding bulk substrate after the full tip retraction. Perspective AFM image corresponding to the SEM image of the $MoS_2$ nanomesa is also shown. Scale bars indicate 50 nm. **c** Typical shear force–lateral piezo displacement curves recorded at the intact homointerface of G (in red), hBN (in blue), and $MoS_2$ (in brown) crystals. Schematic inset shows that the 2D crystal substrate moves along the long axis of the cantilever tip at zero contact force. Top panel: Corresponding SEM images of the sheared G, hBN, and $MoS_2$ nanomesas. Scale bars indicate 50 nm. **d** Typical retraction $F–d$ curves recorded at the interface of tip-attached G nanomesa/G substrate (in blue), bare Si tip-attached water nanomeniscus/$SiO_x$ substrate (in blue), and tip-attached G mesa/Cu substrate (in brown) at the relative humidity of 15, 70, and 15%, respectively. Schematic inset shows the meniscus rupture between the in situ flattened Si tip and the $SiO_x$/Si substrate, where the energy required to rupture the nanomeniscus is roughly obtained by dividing the area under the $F–d$ curve over the area of the flat tip. **e** Cohesion energy of G, hBN, and $MoS_2$ crystals obtained by normal force measurements (squares with cyan borders) and shear force measurements (circles with orange borders) at room temperature. Data are presented as average ± standard deviation.

substrate). Similarly, the shear $F–d$ curves are recorded as the nanomesa is sheared along the long axis of the cantilever tip rather than perpendicular to its long axis to obtain more accurate shear force measurements. Figure 1b, c illustrate typical retraction $F–d$ and shear $F–d$ curves, respectively, at the intact G, hBN, and $MoS_2$ homointerfaces. After complete retraction of the tip, our scanning electron microscopy (SEM) and AFM inspection of the lower section of the nanomesas on the 2D crystal substrate reveals an atomically flat and defect-free surface at the separation plane (top panel of Fig. 1b, c). From retraction force measurements (Fig. 1b), we observe a relatively gradual reduction of the adhesion force (rather than a snap-back to zero force), in particular, at the G and hBN homointerfaces, which looks at the first glance, fairly similar to the AFM rupture force curves of capillary nanobridges. However, the hydrophobic nature of 2D crystal nanomesas along with our $F–d$ approach curves, which display a small jump-to-contact force of 8–12 nN at a small relative tip–sample distance of 5–6 nm (see, for instance, hBN/hBN approach curve in Fig. 1b), suggest dry contact at the interface with negligible effect of tip–sample capillary forces on the retraction curves. For comparison purposes, we recorded the rupture force curve of a water nanomeniscus formed between the in situ flattened Si tip and the $SiO_x$/Si substrate at the relative humidity (RH) of 70% (red curve in Fig. 1d). By closer inspection of the $F–d$ curves, we notice three fundamental differences in the separation mechanism between 2DLMs (e.g., blue curve in Fig. 1d) and capillary nanobridges (e.g., red curve in Fig. 1d). First, the separation range in the 2DLMs (typically 5–10 nm) is almost an order of magnitude shorter than that in the capillary nanobridges (typically 50–80 nm), further supporting the claim that the short-range vdW interaction (rather than the long-range nanobridge deformation) is the major separation mechanism in 2DLMs. Second, contrary to the case of the capillary nanobridges where the adhesion strongly depends on the retraction speed of the piezo[22], our $F–d$ analysis under various tip retraction rates in the range of 1–1000 nm/s reveals no appreciable effect on the IAE of 2DLMs, indicative of the absence of any dynamic (viscous) forces in the separation mechanism of 2DLMs. Third, an abrupt drop in the retraction force curves of nanobridges just prior to the complete separation could reflect the pinch-off process of the unstable meniscus neck, whereas a relatively fast transition from surface contact to line contact during the separation process in 2DLMs can eventually lead to the sudden break of the line contact and thereby an abrupt force drop at the very end of the separation process.

We also note that our shear force measurements (Fig. 1c) exhibit fluctuations in plateau regions for all 2D crystals, which

can be attributed to the stick–slip friction of the tip-attached top mesa on the mobile bottom mesa, indicating that the present axial shear force microscopic technique can provide the shear force resolution up to the subnano level compared to the conventional lateral shear force microscopic technique. This is due to the fact that the spring constant of the probe in the axial shear force microscopy ($8.60\,N\,m^{-1}$) is an order of magnitude smaller than that in the conventional lateral shear force microscopy ($83.8\,N\,m^{-1}$; Supplementary Fig. 9).

Figure 1e shows the intrinsic IAE (i.e., cohesion energy) of G, hBN, and $MoS_2$ crystals at room temperature with an average value of, respectively, $0.328 \pm 0.028$, $0.326 \pm 0.026$, and $0.482 \pm 0.032\,J\,m^{-2}$ using normal force microscopy, matching well with the corresponding average value of $0.361 \pm 0.014$, $0.372 \pm 0.015$, and $0.501 \pm 0.017\,J\,m^{-2}$ using the shear force microscopy. Slightly larger IAE values obtained by the shear measurement technique might be attributed to the small contribution of friction forces to the overall interfacial shear strength of 2D crystals. Nonetheless, both IAE measurement techniques indicate that the strongest interaction occurs between adjacent $MoS_2$ layers due to dipolar, partially ionic Mo–S bonds, whereas nonpolar C–C bonds and highly polar B–N bonds offer a roughly similar level of interaction at the G and hBN homointerfaces, respectively. In fact, stronger interactions with a faster detachment at the homointerface of $MoS_2$ than G and hBN suggest more electron sharing and thus stronger interlayer bonding in $MoS_2$ beyond a simple vdW-only interaction. Similar to the adhesion behavior at the $MoS_2$ homointerface, a sudden detachment of 2D crystal tips from metal substrates (e.g., Ni, Cu, Pt, and Au) is observed with strong interfacial adhesion (see, for instance, the $F–d$ curve of G tip on the Cu substrate in Fig. 1d), suggesting that metal atoms share electrons with carbon atoms.

It is also worth making a comparison between our IAE results and those in the literature. There are several experimental methods to measure the IAE of G crystal with the reported values ranging from 0.15 to $0.72\,J\,m^{-2}$ (see Supplementary Table 1). Among few direct measurements of intrinsic IAE values, we found that our measurements for cohesion energy of G crystal are in excellent agreement with micro-force sensing probe measurements on 4-μm-wide square mesas ($0.37 \pm 0.01\,J\,m^{-2}$, ref. [15]) and AFM-assisted shearing measurements on 3-μm-wide square mesas ($0.35\,J\,m^{-2}$, ref. [16]) but inconsistent with the AFM-assisted shearing measurements on circular mesas of 100–600 nm in diameter ($0.227 \pm 0.005\,J\,m^{-2}$, ref. [17]). We revisited the lateral stiffness calibration of all probes used in ref. [17] by means of a three-dimensional (3D) finite element simulation, predicting consistently stiffer (~1.5 times) probes than those described in the

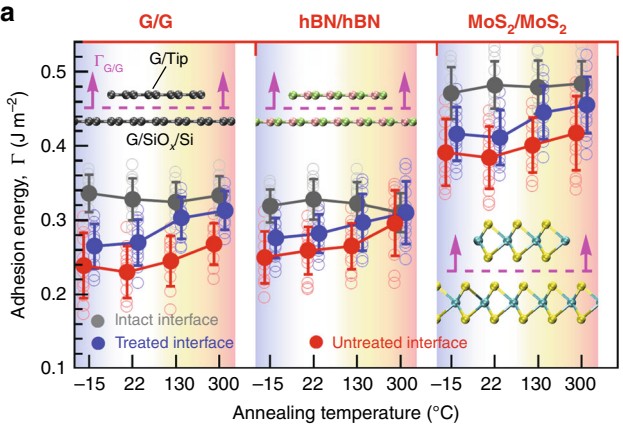

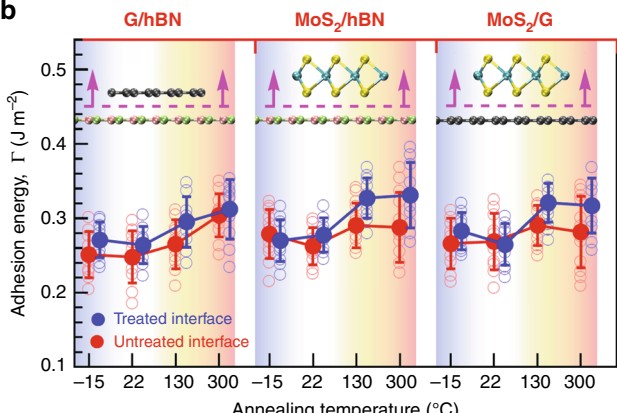

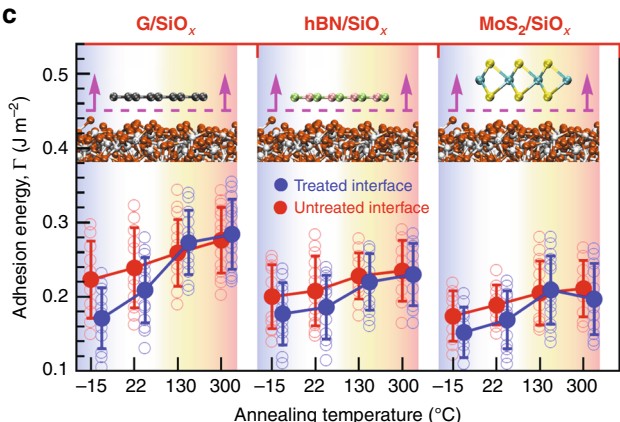

**Fig. 2 Interfacial adhesion energy measurements.** IAE values as a function of annealing temperatures at the **a** G, hBN, and MoS$_2$ homointerfaces; **b** G/hBN, MoS$_2$/hBN, and MoS$_2$/G heterointerfaces; and **c** G/SiO$_x$, hBN/SiO$_x$, and MoS$_2$/SiO$_x$ heterointerfaces using normal force microscopy technique. Filled gray circles in **a** denote the intrinsic IAE values at the intact G, hBN, and MoS$_2$ homointerfaces, whereas filled blue (red) circles in **a**–**c** denote the IAE values between 2D crystal tips and precooling-treated (untreated) substrates. Each open transparent gray circle in **a** represents a single IAE measurement at the intact homointerfaces, whereas each data point shown in open transparent blue and red circles in **a**, **b** represents the IAE of the tips on an individual 2D crystal flake averaged over ten measurements from different locations of the flake surface. Similarly, each data point shown in open transparent blue and red circles in **c** represents the average IAE value obtained from 10 measurements within an individual small region (1 μm × 1 μm) of SiO$_x$ substrate. Each filled circle in **a**–**c** is presented as average of all corresponding open circles ± standard deviation. Insets in **a**–**c** illustrate ball-and-stick representation of various tip/substrate interfaces where carbon, boron, nitrogen, molybdenum, sulfur, silicon, and oxygen atoms are shown in gray, green, pink, cyan, yellow, white, and orange, respectively.

with IAE measurements at the intact and aged 2D crystal homointerfaces under different annealing temperatures (Fig. 2a). It is evident from the gray circles in Fig. 2a (and also Supplementary Table 5) that, upon the attachment of nanomesas to the AFM tip after the thermal annealing, the measured cohesion energy at the intact homointerfaces is, within our experimental accuracy, independent of the annealing temperatures. However, after exposing the freshly exfoliated 2D crystal flakes to the ambient air, the IAE between similar vdW heterostructures (red circles in Fig. 2a) is consistently lower than their corresponding intrinsic value, mainly due to the possible adsorption of airborne contaminants (e.g., water and hydrocarbon molecules) onto the fresh surface of crystals, thereby reducing their overall free surface (Gibbs) energy. A 30 and 19% drop in the IAE of G/G and hBN/hBN, respectively, at room temperature suggests that G is more influenced by the airborne contaminants than hBN of similar lattice structure with only slightly larger (~1.8%) lattice constant. Although a mild annealing temperature (130 °C) coupled with relatively strong vdW interactions at the interface can provide a sufficient driving force to push the trapped water molecules away from the contact interface and thus to slightly improve the IAE of the crystals (up to ~5%), a higher annealing temperature is required to build up larger pressure at the interface to drive out the majority of hydrocarbons as the main source of such IAE drop, leading to nearly full recovery of the intrinsic IAE only at the hBN homointerface upon annealing at 300 °C. Interestingly, despite stronger vdW interaction of MoS$_2$/MoS$_2$ and similar level of interaction of G/G compared to hBN/hBN, the full aggregation of such contaminants into nanobubbles at the G and MoS$_2$ homointerfaces can only be triggered at a much elevated temperature, implying that hydrocarbons have a stronger interaction with G and MoS$_2$ than hBN.

We next perform a series of measurements to study the effect of precooling treatment of the substrate on the IAE of the homointerfaces (blue circles in Fig. 2a). Surprisingly, such a precooling treatment can significantly improve the IAE of the hBN, G, and MoS$_2$ crystals regardless of the subsequent annealing temperature. While such an IAE improvement upon 130 °C and 300 °C thermal annealing could be intuitively understood by hypothesizing that the formation of ice-like monolayer/bilayer on the freshly cleaved 2D crystals can be effectively leveraged as a self-release underlying film for the facile removal of the subsequent hydrocarbon adsorptions, this hypothesis might not be supported by our findings at room temperature and −15 °C

original work (Supplementary Fig. 11). Using this modified lateral spring constant yields an IAE value of 0.348 ± 0.008 J m$^{-2}$, more consistent with our measurements. We also note that, to the best of our knowledge, no IAE measurement on the hBN homointerface yet exists, while there is only one measured IAE value of 0.22 J m$^{-2}$ at the MoS$_2$ homointerface using a nanomechanical cleavage technique[18], which is much lower than our values. Given that the bending stiffness of TMDs is reported in the range of 10–16 eV[23], we believe that a very low bending stiffness value of 0.92 eV used in their calculations for the monolayer MoS$_2$ has resulted in such a low IAE value.

**Adhesion between similar vdW heterostructures.** Figure 2 presents IAE values of both fresh and aged vdW heterostructures at the intact, precooling-treated and untreated heterointerfaces as a function of annealing temperatures. We begin our discussion

where the ice-like layer is still stable and tightly bonded to the underlying crystal surface. Notably, however, our observations can be fully supported for all range of temperatures by a recent study, showing that water adsorption on graphitic surfaces can significantly slow down the hydrocarbon adsorption rate[24], thus making the nanometer-thick ice-like water an excellent protective layer against the airborne contamination for several hours. We also note that an increase of the annealing temperature from 130 to 300 °C exhibits further improvement of the IAE, implying that the ice-like layer that is completely removed at $T \leq 130$ °C cannot fully cover the crystal surface, still leaving unprotected areas with adsorbed high boiling point hydrocarbons.

**Adhesion between dissimilar vdW heterostructures**. Our IAE measurements on dissimilar vdW heterostructures exposed to air at room temperature (red circles in Fig. 2b) reveal that the adhesion level at the untreated G/hBN interface remains roughly the same as that at the untreated G and hBN homointerfaces over the temperature range of −15–300 °C, whereas the IAE value of MoS$_2$ on the untreated G and hBN substrates is considerably smaller than that on the untreated MoS$_2$ substrate. During the approach–retract course, we observe a relatively stronger adhesive response of the G nanomesa to G than hBN substrate within our experimental accuracy, suggesting that the IAE of G/hBN is governed by a lower level of dispersion energy at the interface with a negligible contribution from the electrostatic interactions of hBN, which are absent at the G/G interface. However, this is not the case at the contact interface between MoS$_2$ and hBN(G) where different crystal structures and different static polarizabilities of the constituent atoms dictate very different levels of short-range dispersive (vdW) and long-range electrostatic (Coulombic) interactions at the MoS$_2$/MoS$_2$ and MoS$_2$/hBN(G) interfaces. Such a different interaction energy level is further confirmed by recent cross-sectional scanning transmission electron microscopy imaging of vdW heterostructure interfaces, showing different interlayer separations between MoS$_2$/hBN and MoS$_2$/MoS$_2$[25].

Notably, we observe that, unlike very high-quality interface of untreated MoS$_2$/MoS$_2$ upon annealing at 300 °C, the untreated MoS$_2$/hBN(G) interface does not show any further improvement, implying an absent or even negative impact of such a high annealing temperature on the IAE of MoS$_2$/hBN(G). This counterintuitive observation can be explained by a trade-off between interface self-cleansing mechanisms driven by the vdW forces and MoS$_2$ oxidation process triggered by relatively high temperatures (>130 °C) in the ambient air. On one hand, considerably weaker vdW interaction in MoS$_2$/hBN(G) than in MoS$_2$/MoS$_2$ can still provide sufficient driving forces of similar magnitude to those of G/G, hBN/hBN, and G/hBN for the segregation of the contaminants to the localized nanobubbles, leading to the enhanced IAE of MoS$_2$/hBN(G) at 300 °C. On the other hand, the weaker interaction of MoS$_2$/hBN(G) can facilitate the oxygen interfacial diffusion and thereby the oxidation process, which is initiated from the edges, grain boundaries, and intrinsic atomic defects of MoS$_2$ and gradually penetrates into the MoS$_2$ grains at the interface. This is consistent with the low-temperature surface oxidation of MoS$_2$, which is initiated at ~100 °C and significantly increases at 300 °C[26], resulting in the negative impact of oxygen adsorption on the mobility and homogeneity of MoS$_2$/G heterostructure devices after annealing above 150 °C[27]. We hypothesize two possible interfacial oxidation mechanisms responsible for the weaker interaction of MoS$_2$/hBN (G) at 300 °C: (1) replacement of sulfur atoms with oxygen atoms results in a lower surface energy in the oxidized MoS$_2$ (MoO$_3$) than unreacted MoS$_2$; (2) partial protrusions (0.36 ± 0.25 nm[28]) at

the interface due to formation of interfacial MoO$_3$ patches along with the presence of gaseous reaction products (e.g., SO$_2$, which cannot diffuse out of interface owing to very high vdW pressure on the trapped molecular layers[29]) can give rise to local interlayer decoupling of unreacted MoS$_2$ crystal from underlying hBN and G substrates.

Similar to the precooling-treated G, hBN, and MoS$_2$ homo-interfaces, it is evident from the blue circles in Fig. 2b that precooling treatments can effectively protect the crystal substrates against the airborne contaminants and thus boost the adhesion level at the interface of dissimilar vdW heterostructures at much lower annealing temperature of 130 °C. However, such a protective layer offers no appreciable improvement in the IAE of MoS$_2$/hBN(G) at 300 °C, further confirming the possible destructive effect of interfacial contaminations/oxygen diffusion on the MoS$_2$ oxidation at higher temperatures (see Supplementary Fig. 10 for our X-ray photoelectron spectroscopy (XPS) on both freshly exfoliated and pre-annealed MoS$_2$ samples).

During the revision of this paper, Li et al. reported the vdW interactions of G/hBN and G/MoS$_2$ heterostructures, where a Si AFM tip wrapped with a thin G flake is brought into contact with pre-annealed G, hBN, and MoS$_2$ substrates in high vacuum at room temperature[19]. Using a G-wrapped sharp tip with an unknown contact area only allows the measurements of the critical adhesion forces (i.e., pull-off forces, $P$) between G/G, G/hBN, and G/MoS$_2$. Qualitatively speaking, their measurements showed that G experiences a weaker vdW interaction with hBN and G than MoS$_2$, yielding a critical adhesion force ratio of $P_{G/MoS_2}/P_{G/hBN} = 1.079$ and $P_{G/MoS_2}/P_{G/G} = 1.028$. Similarly, our IAE ratios of $\Gamma_{G/MoS_2}/\Gamma_{G/hBN}$ and $\Gamma_{G/MoS_2}/\Gamma_{G/G}$ for the roughly similar experimental conditions (i.e., precooling-treated heterointerfaces annealed at 130 °C) are 1.088 and 1.059, respectively, which are qualitatively in good agreement with their findings.

**Adhesion between 2D crystals and SiO$_x$**. Despite many experimental and theoretical studies devoted to the IAE determination of 2D crystals/SiO$_x$ heterostructures, no experimental data, to our knowledge, are available on the interaction of hBN/SiO$_x$, whereas the reported IAE data on the interaction of G and MoS$_2$ with SiO$_x$ are very diverse, ranging from 0.14 to 0.90 J m$^{-2}$ at the G/SiO$_x$ interface (Supplementary Table 3) and 0.17–0.48 J m$^{-2}$ at the MoS$_2$/SiO$_x$ interface (Supplementary Table 4). To gain an atomic-level understanding of interaction mechanisms at the 2D crystals/SiO$_x$ heterointerfaces, we first report in Fig. 2c an AFM quantitative characterization of the interlayer interactions of 2D crystals on both untreated and precooling-treated SiO$_x$ substrates. Similar to untreated 2D crystals, the thermal annealing can effectively remove the water and hydrocarbons from the untreated SiO$_x$ surface, leading to the higher IAE at the 2D crystal/SiO$_x$ interface. Surprisingly, however, unlike the case of 2D crystal substrates, the precooling treatment results in the weaker interaction of the 2D crystals with the SiO$_x$ substrate at both −15 °C and room temperature. This weaker interaction may be explained by the hydrophilic nature of SiO$_x$ that can adsorb a homogeneous and flat water film of thickness 2–3 nm (~6–10 monolayers of water) on its silanol (Si–OH)-rich surface when storing at −15 °C (corresponding to 100% RH)[30]. As such, our IAE measurements at −15 °C (i.e., 0.171 ± 0.041, 0.177 ± 0.042, and 0.152 ± 0.034 J m$^{-2}$ for the G, hBN, and MoS$_2$ crystals on SiO$_x$, respectively) essentially take place at the 2D crystal/water interface rather than at the 2D crystal/SiO$_x$ interface. In addition, a larger IAE of 2D crystal/treated SiO$_x$ at room temperature compared to that at −15 °C can be attributed to the presence of a mixture of ice-like monolayer/bilayer structure (fully H-bonded

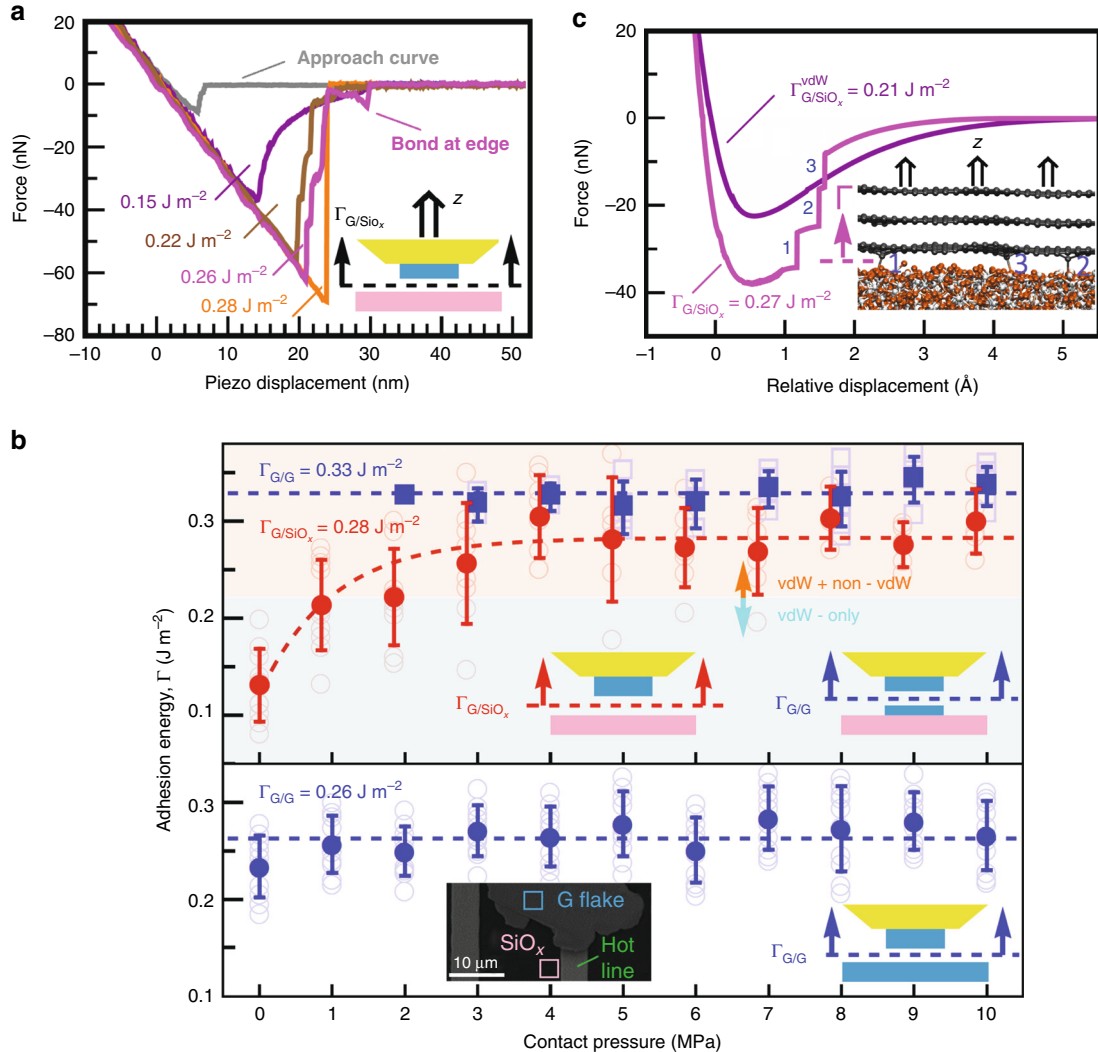

**Fig. 3 Interaction mechanism at G/SiO$_x$ heterointerface. a** Typical retraction $F$–$d$ curves of the G/SiO$_x$ interface recorded at no contact pressure (in purple) and 5 MPa (in brown, magenta and orange) and also a typical approach $F$–$d$ curve (in gray). **b** IAE values as a function of contact pressure at the intact G homointerfaces (filled blue squares in top panel), G/SiO$_x$ interfaces (filled red circles in top panel), and aged G homointerfaces (filled blue circles in bottom panel). Each open transparent symbol represents a single IAE measurement at the given contact pressure. Each filled symbol is presented as average of all corresponding open symbols ± standard deviation. From 110 IAE measurements shown in the top panel, 33 and 77 contacts result in the separation of G crystal tip across the thickness of the nanomesa (open blue squares) and from the SiO$_x$ surface (open red circles), respectively. In the top panel, cyan and orange shaded regions indicate the vdW-only and vdW+non-vdW interactions of G/SiO$_x$, respectively, where the lift-off in the vdW-only region is relatively gradual in comparison to the vdW+non-vdW region. Dashed blue lines denote an overall average IAE value of 0.328 ± 0.022 and 0.263 ± 0.032 J m$^{-2}$ at the intact G homointerface (top panel) and aged G homointerface (bottom panel), respectively. Dashed red line in the top panel represents the best fit to the data, indicating the pressure dependence of IAE at the interface with an average value of 0.284 ± 0.046 J m$^{-2}$ taken within the pressure-independent region (i.e., ≥3 MPa). Inset of bottom panel: SEM image showing 2 μm × 2 μm square regions of G (in turquoise) and SiO$_x$ (in rose) substrates on which all measurements are performed in close proximity to the microheater. **c** MD-calculated force versus relative displacement curves for the vdW-only interaction of G/SiO$_x$ (in purple) and vdW+non-vdW interaction of G/SiO$_x$ (in magenta) with three covalent C–O–Si bonds at the interface. Each force jump labeled by 1, 2, and 3 represents the break of the corresponding covalent bond, as illustrated by the MD pull-off simulation in the inset.

to the silanol groups) and liquid-like few-layer structure on top of the ice-like layer at room temperature. Such trapped liquid-like film can segregate into isolated nano-sized bubbles[31] by means of the contact pressure and the interlayer vdW forces, bringing the 2D crystals into closer proximity with the SiO$_x$ surface and thus enhancing the IAE at the 2D crystal/SiO$_x$ interface. We also note that the thermal annealing at 130 and 300 °C makes no appreciable difference in the interfacial adhesion level between 2D crystal/precooling-treated SiO$_x$ and 2D crystal/untreated SiO$_x$, confirming the formation of the protective ice-like layer on the untreated SiO$_x$ substrate due to 15% RH in the ambient air.

Similar to the interaction of MoS$_2$ crystal with the other 2D crystals, the high annealing temperature of 300 °C can reduce the adhesion level at the MoS$_2$/treated-SiO$_x$ interface, which lends additional support to the hypothesis of MoS$_2$ oxidation at higher temperatures due to the chemical reaction of MoS$_2$ with the trapped water and the diffused oxygen. Such significantly reduced IAE of MoS$_2$/treated SiO$_x$ relative to MoS$_2$/hBN(G) at 300 °C can be understood as a direct consequence of the strong hydrophilic property of SiO$_x$, where the MoS$_2$ crystal undergoes an additional chemical reaction with the interfacial water layer, resulting in the partial etching of the MoS$_2$ interface layer and needle-like

protrusions due to the formation of $MoO_3 \cdot H_2O$ on the $MoS_2$ surface[32]. It is worth pointing out that although oxygen plasma treatment can completely remove any water and hydrocarbon molecules, leaving a contamination-free $SiO_x$ surface terminated with more silanol groups, we observed that stronger interaction between 2D crystals and plasma-cleaned $SiO_x$ surface leads to the exfoliation of 2D crystals across the thickness of nanomesa, making a direct IAE measurement at 2D crystals/plasma-treated $SiO_x$ interfaces inaccessible.

**Adhesion at G/SiOx interface: beyond vdW interaction**. During interfacial adhesion measurements of $G/SiO_x$ at the annealing temperatures of 130 and 300 °C, we unexpectedly observed abrupt detachment of G nanomesa with single/multiple force jumps in the $F–d$ retraction curves, resulting in much stronger interfacial interactions in $G/SiO_x$ than in $hBN(MoS_2)/SiO_x$. In particular, the separation process of 2D crystal tips in our set-up dictates a relatively gradual reduction of the interfacial adhesion force between two adjacent 2D crystal flakes and between $hBN(MoS_2)$ nanomesas and $SiO_x$ substrate. As a result, such sudden detachment with single/multiple plateau force jumps in the $G/SiO_x$ adhesion curves cannot be interpreted as a consequence of experimental noises, thermal fluctuations, and mechanical instabilities of the probe, as they are roughly equally present in all our $F–d$ measurements.

From over hundred interfacial adhesion measurements for the $G/SiO_x$ interaction, we identified three distinct $F–d$ curves, each describing gradually broken contacts (i.e., weak interaction without any force jump), suddenly broken contacts (i.e., strong interaction with a single force jump), and a transition from gradually to suddenly broken contacts (i.e., mild interaction with multiple force jumps), as shown in Fig. 3a. To provide a rational explanation of the origin of such distinctive interfacial behavior in the $G/SiO_x$ heterostructure, we begin by addressing quantitatively to what degree the interfacial adhesion of $G/SiO_x$ interfaces (and also intact and aged G homointerfaces for comparative purposes) is controlled by the conformity of the tip-attached G nanomesa to the underlying substrate morphology. To this end, a series of interfacial adhesion measurements over a pressure range of 0–10 MPa was conducted at the interface of G crystal tip/pre-annealed $SiO_x$ substrate (top panel of Fig. 3b) and G crystal tip/pre-annealed G substrate (bottom panel of Fig. 3b). This set-up only allowed us to study the interaction of $G/SiO_x$ weaker than that between G/G (red circles in Fig. 3b), otherwise the separation takes place across the thickness of G nanomesa (blue squares in Fig. 3b).

It is observed from Fig. 3b (top panel) that the G crystal tip requires a contact pressure of ≥3 MPa to conform closely to the $SiO_x$ surface, thereby enhancing the IAE of the $G/SiO_x$ interface from $0.131 \pm 0.038$ J m$^{-2}$ at zero pressure to $0.289 \pm 0.034$ J m$^{-2}$ at 10 MPa. In contrast, both the intact G homointerface (blue squares in the top panel of Fig. 3b) and the aged G homointerface (bottom panel in Fig. 3b) suggest a constant IAE value of $0.328 \pm 0.022$ and $0.263 \pm 0.032$ J m$^{-2}$, respectively, almost entirely independent of the pressure, indicative of flat and dangling bond-free G/G interfaces. It is also evident from the top panel in Fig. 3b that graphene flakes are not exfoliated from the tip-attached G nanomesa onto the $SiO_x$ at very low pressure (< 2 MPa) and only 10 and 20% of contacts at 2 and 3 MPa, respectively, result in the exfoliation of graphene flakes, indicating the significant contribution of the conformal adhesion to the overall interfacial adhesion strength of the $G/SiO_x$ interface. More importantly, we observe that abrupt detachment events with single/multiple force jumps in the retraction curves of $G/SiO_x$ take place more frequently at higher pressure in such a way that

nearly all contacts are suddenly broken at the pressure load of ≥4 MPa with IAE values roughly >0.221 J m$^{-2}$. Surprisingly, this value is very close to theoretical calculations of the intrinsic vdW interaction energy (0.230 J m$^{-2}$) at the $G/SiO_x$ interface obtained for the multilayer graphene blister tests on the $SiO_x$ substrate under pressure loading[20]. Hence, while a continuous decrease in the retraction curves can be attributed to the long-range vdW interaction of $G/SiO_x$ with IAE values typically <0.221 J m$^{-2}$, direct observation of single/multiple force jumps at stronger $G/SiO_x$ interfaces can be hypothesized to be the result of formation of short-range chemical bonds at the interface. Both experimental and theoretical results confirm that G flakes supported on $SiO_x$ exhibits much higher chemical reactivity than suspended G flakes, mainly due to the combined action of inhomogeneously distributed charge puddles (induced by polar adsorbates, such as water molecules on the silanol surface and by ionized impurities, such as $Na^+$ ions trapped on $SiO_x$) and larger topographic corrugations (induced by thermal fluctuation and vdW interaction at the $G/SiO_x$ interface)[33–35]. As such, hydrogen and oxygen molecules preferentially bind to apexes of corrugated G due to the combined contribution from the enhanced elastic and electronic energies of convex regions on the G surface[36].

In addition, the specific chemical reactivity of the carbon atoms with accessible and highly active electrons at the edge of the G flakes can also contribute to the formation of chemical bonds, as observed in a number of our retraction curves (e.g., see the magenta curve in Fig. 3a and also see Supplementary Note 2.6). This, coupled with our molecular dynamics (MD) observations that force jumps in the retraction curves can only be achieved by breaking short-range chemical bonds at the $G/SiO_x$ interface (Fig. 3c), provides further support for the hypothesis that more likely hydrogen bonds and/or less likely covalent bonds are formed between G and $SiO_x$. Notably, from Fig. 3b, only ~22% chemical bond-induced improvement in the IAE of $G/SiO_x$ (i.e., from $0.221 \pm 0.030$ to $0.284 \pm 0.046$ J m$^{-2}$) under relatively low pressure (of the order of few MPa) leads us to believe that (1) vdW interactions are still the dominant mechanism of adhesion at the $G/SiO_x$ interface; and (2) the formation of hydrogen bonds (e.g., C–H...O–Si, C–O...H–O–Si, and C–O–H...O–Si in the absence of contaminants and $C–H...O_{\backslash H} - H...O–Si$ and C–O...H–O–H...O–Si in the presence of water molecules) rather than covalent bonds could result in such force jumps in the retraction curves; nonetheless, the formation of any covalent bonds between G and $SiO_x$ (e.g., C–O–Si and C–Si[37,38]) cannot be completely ruled out because the effect of localized tensile strain and charge transfer on the chemical activity level of the corrugated G is poorly understood.

It is also to be noted that, in contrast to MD-calculated retraction curves in Fig. 3c, the number of force jumps in Fig. 3a does not necessarily correspond to the number of chemical bonds at the $G/SiO_x$ interface. One may argue from an interfacial fracture standpoint that, once the restoring force of the probe cantilever exceeds the strength of the $G/SiO_x$ interaction (i.e., the pull-off force), interfacial nano-sized cracks start to form due to the localized nano delamination across the separation plane and propagate until complete interfacial fracture occurs[2]. As such, the interfacial fracture of $G/SiO_x$ heterostructure is a combined action of the external pull-off force and the internal adhesion force (i.e., vdW and/or non-vdW forces). For the case of the vdW-only interaction of $G/SiO_x$, both a smaller pull-off force and the smooth and slow propagation of nanocracks contribute to the relatively gradual reduction of the interfacial adhesion force (purple curve in Fig. 3a). In contrast, faster crack propagation in the stronger vdW+non-vdW interaction of $G/SiO_x$, which is triggered by a larger pull-off force, results in the abrupt force drop

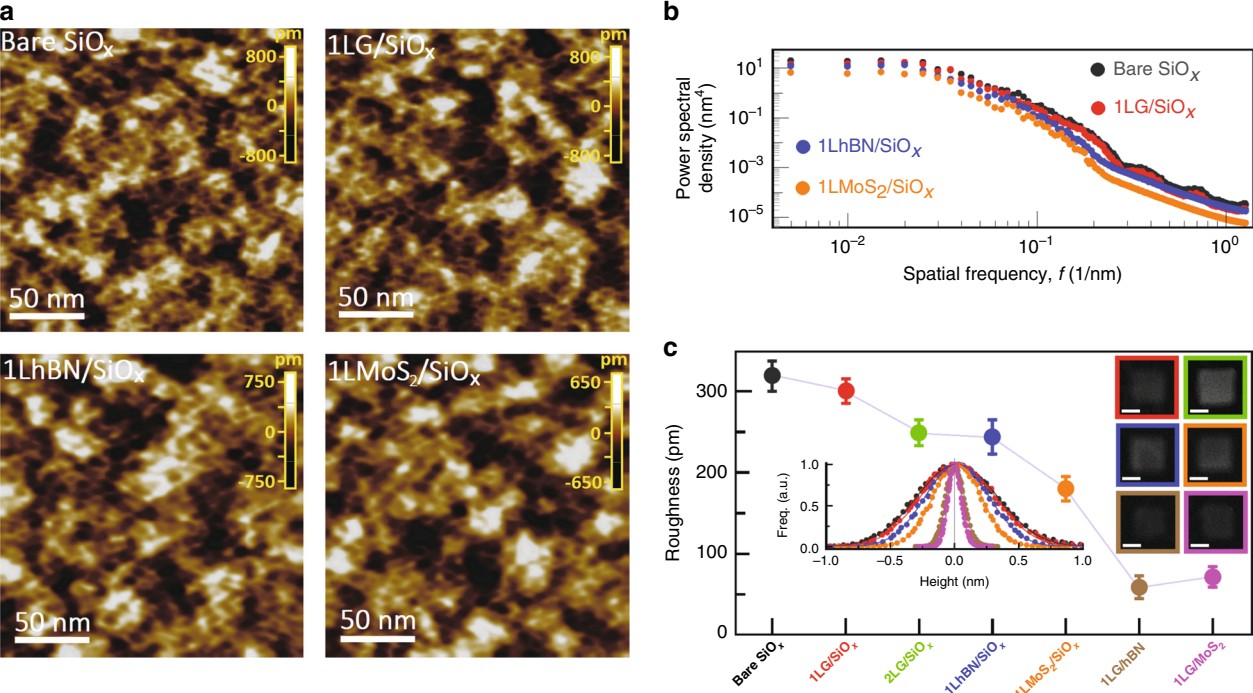

**Fig. 4 Surface roughness measurements of different heterostructures. a** High-resolution AFM images of the surface roughness of bare $SiO_x$, monolayer graphene (1LG), monolayer hBN (1LhBN), and monolayer $MoS_2$ (1LMoS$_2$) on $SiO_x$ substrate. **b** PSD profiles corresponding to the images in **a**. **c** Surface roughness measurements of different heterostructures. Error bars show the spread of data over several independent measurements of different flakes. Left inset: histogram of the height distribution (surface roughness) for bare $SiO_x$, 1LG/$SiO_x$, 1LhBN/$SiO_x$, 1LMoS$_2$/$SiO_x$, 1LG/hBN, and 1LG/MoS$_2$ substrates. Solid lines are Gaussian fits to the distribution. Right inset: representative SEM images of 2D crystal square mesas exfoliated onto the substrate. Scale bar in each is 5 μm. **b**, **c** and the insets of **c** share the same color legend.

in the retraction curves immediately upon the initiation of the separation process. As the nanocracks are continuously propagating and the pull-off force becomes progressively smaller and smaller, the chemical bonds (i.e., the anchoring spots) gain the ability to pin the nanocrack tips and thus momentarily retard the crack propagation. Such unique crack arresting behavior at the G/$SiO_x$ interface gives rise to very short signals in the retraction curves through a significant decrease in the force drop rate, making the detection of the chemical bonds possible in our set-up (brown and magenta curves in Fig. 3a). However, in the case of suddenly broken contacts with a single force jump (e.g., orange curve in Fig. 3a), as the number of the interfacial chemical bonds increases, larger and larger pull-off forces are required to initiate the interfacial fracture of the G/$SiO_x$, thereby much faster nanocrack propagation at the beginning of the separation process causes all interfacial chemical bonds to suddenly break and thus no longer allows our set-up to capture the crack-arresting behavior during propagation. Furthermore, while both the pull-off and interfacial adhesion forces are primarily responsible for developing the interfacial nanocrack growth and separation (see step-like events in the retraction curves), when the pull-off force approaches zero, further pull-off force needs to build up to overcome possible chemical bonds at the edge of G flakes (magenta curve in Fig. 3a).

**Origin of distinctive interfacial adhesion behavior in G/$SiO_x$.**
To gain a sub-nanoscale insight into the origin of the distinctive interfacial behavior in the G/$SiO_x$ heterostructure specifically and into the underlying interaction mechanism of 2D crystals and $SiO_x$ in general, we perform 3D surface topographic measurements of single-layer 2D crystals (transfer-printed on the $SiO_x$ surface under a controlled contact pressure of 5 MPa) with the

power spectral density (PSD) analysis of the surface roughness data (Supplementary Note 3). Typical high-resolution $200 \times 200$ nm$^2$ AFM topographic images of bare $SiO_x$ and monolayer G, hBN, and $MoS_2$ flakes supported on $SiO_x$ and the PSD profiles corresponding to the images are shown in Fig. 4a, b, respectively. It is evident that highly random corrugations with sub-nanometer vertical dimension but few-nanometer lateral dimension in monolayer 2D crystals are imposed by the underlying $SiO_x$ substrate. In Fig. 4c, the average surface roughness of monolayer 2D crystal/$SiO_x$ heterostructures and histograms of the corresponding height distribution are presented, where the measurements from the bilayer of G on $SiO_x$ and monolayer of G on the hBN and $MoS_2$ substrates are also shown for comparative purposes. Our roughness measurements suggest that monolayer G exhibits the highest degree of conformation to the $SiO_x$ (roughness ratio: 0.94), followed by bilayer G (0.78), monolayer hBN (0.76), and monolayer $MoS_2$ (0.57). As expected, the topography of monolayer G on hBN and $MoS_2$ substrates is much smoother than that of monolayer G on $SiO_x$, suggesting an atomically flat contact at G/hBN and G/$MoS_2$ interfaces. Assuming that the conformation of 2D crystal flakes to the underlying $SiO_x$ substrate of similar corrugation pattern is proportional to their IAE but inversely proportional to the bending stiffness of the flakes[39] with a value of $D_{1LG} = 1.49$ eV, $D_{2LG} = 35.5$ eV, $D_{1LhBN} = 1.34$ eV, and $D_{1LMoS_2} = 11.7$ eV, (Supplementary Note 7), the smaller bending stiffness of monolayer hBN compared to monolayer and bilayer G, however, results in a smoother surface morphology, further confirming the stronger IAE at the G/$SiO_x$ interface. Moreover, our comparative study of the corrugation of bilayer G and monolayer $MoS_2$ with almost the same thickness (i.e., 0.670 nm in 2LG versus 0.645 nm in 1LMoS$_2$) also demonstrates that the adhesion of bilayer G to $SiO_x$ is much stronger than that of

**Table 1 Cohesion energy ($\Gamma$), intrinsic cleavage strength ($\sigma_s$) and intrinsic interlayer shear strength ($\tau_s$) of 2D crystals.**

| | Normal exfoliation technique | | | Shear exfoliation technique | | |
|---|---|---|---|---|---|---|
| | G | hBN | MoS$_2$ | G | hBN | MoS$_2$ |
| $\Gamma$ (J m$^{-2}$) | 0.328 ± 0.028 | 0.326 ± 0.026 | 0.482 ± 0.032 | 0.361 ± 0.014 | 0.372 ± 0.015 | 0.501 ± 0.017 |
| $\sigma_s$ (MPa) | 21.9 ± 0.9 | 17.2 ± 0.6 | 30.1 ± 1.1 | | | |
| $\tau_s$ (MPa) | | | | 6.02 ± 0.23 | 6.21 ± 0.25 | 8.35 ± 0.6 |

monolayer MoS$_2$. Such intimate and strong interaction of G/SiO$_x$ suggests that the electron-scattering sites across the interface as well as the convex sites of corrugated G result in the formation of short-range chemical bonds, which act as anchoring spots to locally pin G to the SiO$_x$ surface at the location of such chemically active sites[33–36]. Since monolayer G with high flexibility possesses more chemically active sites than multilayer G at the G/SiO$_x$ interface, stronger adhesion energy of monolayer to SiO$_x$ is expected, as previously confirmed by a pressurized blister test to be 0.45 ± 0.02 J m$^{-2}$ for monolayer G but 0.31 ± 0.03 J m$^{-2}$ for multilayer G[21].

## Discussion

While the IAE is obtained from the information of all points on the $F$–$d$ curve, which provides valuable insight into the whole separation process, cleavage strength and interlayer shear strength of 2D crystals (Table 1) can be obtained from the information of one single point on the $F$–$d$ curve (i.e., the pull-off force in Fig. 1b and the maximum shear force in Fig. 1c). Thanks to the known contact area in our set-up, the intrinsic cleavage strength of G, hBN, and MoS$_2$ crystals at room temperature is measured to be 21.9 ± 0.9, 17.2 ± 0.6, and 30.1 ± 1.1 MPa, respectively. Our average value for the cleavage strength of G is consistent with that (10.3–20.7 MPa) reported using static tests for polycrystalline graphite[40]. Also, the interlayer shear strength of G, hBN, and MoS$_2$ crystals for a nanomesa of width/diameter ~60 nm is calculated to be 6.02 ± 0.23, 6.21 ± 0.25, and 8.35 ± 0.6 MPa, respectively. Given that the interlayer shear strength is inversely proportional to the diameter (width) of the circular (square) mesas (see Supplementary Note 2.1), our values are in good agreement with those obtained by the AFM-assisted shearing technique for a 50-nm-radius G nanomesa (3.1 MPa)[17] and microforce shearing technique for a 12.6-nm-width MoS$_2$ nanomesa (25.3 ± 0.6 MPa)[41].

We already explained in Fig. 3a the origin of different trends in our $F$–$d$ curves for the G/SiO$_x$ heterostructure by means of interfacial fracture mechanics. Similarly, we believe that, for the case of the relatively weak vdW-only interaction (e.g., hBN/hBN in Fig. 1b), both a smaller pull-off force and the smooth and slow propagation of nanocracks contribute to the relatively gradual reduction of the interfacial adhesion force. In contrast, faster crack propagation in the relatively stronger vdW-only interaction (e.g., G/G in Fig. 1b), which is triggered by a larger pull-off force, results in the abrupt force drop in the retraction curves immediately upon the initiation of the separation process. However, in the case of suddenly broken contacts (e.g., MoS$_2$/MoS$_2$ in Fig. 1b and G/Cu in Fig. 1d), the more electron sharing at the interface, the larger pull-off force is required to initiate the interfacial fracture, thereby much faster nanocrack propagation at the beginning of the separation process causes a sudden break of the contact.

Although we showed in Fig. 2b, c that MoS$_2$ has a higher chemical activity than G, in particular, at higher temperatures, short-range chemical bonds are not formed at the MoS$_2$/SiO$_x$ interface. A possible reason is that, unlike the G crystal, the basal plane of MoS$_2$ is rather inert unless S vacancies are introduced into its basal plane[42]. Short-range chemical reactions between

MoS$_2$ and SiO$_x$ require (1) vacancy defects in the MoS$_2$ basal plane to directly bind H, O, and Si atoms to exposed Mo atoms and (2) a close conformation of MoS$_2$ to the underlying SiO$_x$ substrate. For the former one, our surface topographic measurements in the absence of the thermal annealing do not exhibit vacancy defects and grain boundaries in the basal plane of MoS$_2$ nanomesas. For the latter one, we showed that the degree of conformation of MoS$_2$ to the SiO$_x$ is much lower than that of G (Fig. 4). These two reasons make the formation of the chemical bonds at the MoS$_2$/SiO$_x$ interface almost impossible at least when the contact forms at room temperature, as is the case in Fig. 3 for the G/SiO$_x$ heterostructures.

In order to provide a valuable guideline for the fabrication of vdW heterostructures based on the vdW pick-up transfer techniques, we present a summary of the cohesion energy at the intact G, hBN, and MoS$_2$ homointerfaces (Supplementary Table 5) and the IAE of untreated and precooling-treated homostructures/ heterostructures (Supplementary Table 6), corresponding to the experimental data points in Figs. 1e and 2, respectively. These tables reveal that hBN may not successfully pick up G flakes from G and MoS$_2$ substrates, whereas the strong adhesion at the MoS$_2$/ hBN(G) interface makes MoS$_2$ a better candidate for the selective G and hBN pick-up from the G and hBN substrates. These tables also suggest that both G and MoS$_2$ can be used to pick up all three 2D crystals from the SiO$_x$ substrate at room temperature. However, the stronger adhesion of G to the SiO$_x$ substrate requires careful selection of the 2D crystals for the high-yield G pick-up, making hBN a relatively improper choice for such a purpose at room temperature. Moreover, the simple precooling treatment of the SiO$_x$ substrate before the mechanical exfoliation of 2D crystals can highly facilitate the 2D crystal pick-up by reducing the IAE at the 2D crystal/SiO$_x$ interface.

Since the effect of airborne surface contaminations and thermal annealing on the IAE of 2D crystals can be well understood by contact angle measurements, we herein discuss the interaction of 2D crystals with airborne contaminants and quantify the effect of surface contaminations and thermal annealing on the IAE of 2D crystals by characterizing intrinsic water wettability of fresh and aged surfaces of 2D crystals with a focus on the G crystal. Many studies on the wettability of G crystal along with very limited studies on the wettability of hBN and MoS$_2$ crystals all suggest that freshly cleaved crystals spontaneously adsorb airborne contamination upon the air exposure, leading to an increase in the water contact angle (WCA) of G[43], hBN[44], and MoS$_2$[45] crystals, respectively, from (64°, 63°, 69°) measured within few seconds of air exposure of fresh surfaces to a saturated value of (90°, 86°, 89°) within few hours of air exposure. Our further analysis on the temporal evolution of the adhesion energy (see Supplementary Note 6 for our detailed analysis) and contamination thickness measured on the mechanically exfoliated G surface during the first 60 min of air exposure reveals that its intrinsic IAE of 0.341 ± 0.025 J m$^{-2}$ obtained under ultrahigh vacuum or ultrahigh-purity argon atmosphere is well consistent with our experimental value of 0.328 ± 0.028 J m$^{-2}$ but drastically decreases within the first minute of air exposure and eventually approaches a saturated value of 0.15 ± 0.02 J m$^{-2}$ after 10 min (Fig. 5), which is smaller

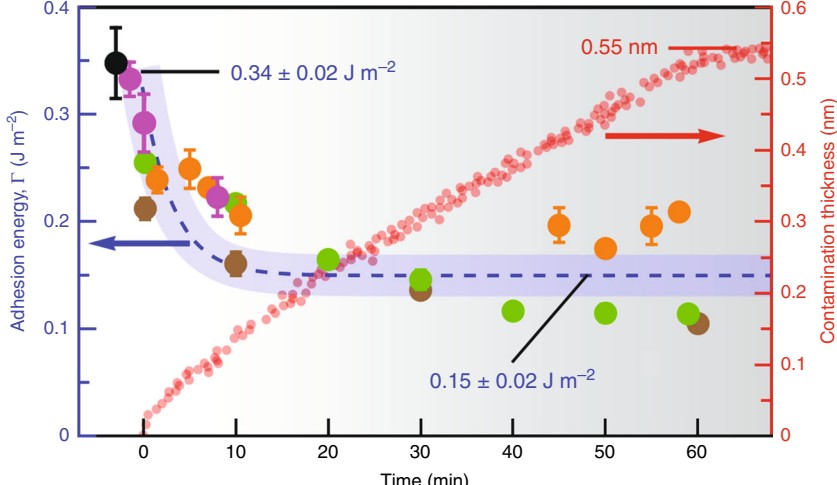

**Fig. 5 Adhesion energy evolution in G crystal.** Temporal evolution of the adhesion energy (left axis in blue) and contamination thickness (right axis in red) measured on the mechanically exfoliated HOPG surface during the first 60 min of air exposure. Adhesion energy is extracted from WCA measurements of ref. [46], ref. [47], ref. [48], ref. [49], and ref. [24] denoted by black, brown, orange, magenta, and green circles, respectively. Adsorbed contamination layers linearly grow within the first 60 min of air exposure, reaching a thickness of ~0.55 nm, and then the growth rate considerably decreases and plateaus at ~0.60 nm after several hours[24].

than our IAE value of $0.233 \pm 0.035 \,\mathrm{J\,m^{-2}}$ upon room-temperature storage for 1 h. This could be attributed to the presence of the contact pressure and the vdW interaction between the layers in our experiments, which may still play a role to squeeze away the contaminants even at room temperature, leaving cleaner interfaces with stronger interactions. We also note that a substantial decrease in the surface hydrocarbon level under vacuum, high-temperature (500–1000 °C) treatment during the WCA measurements results in the IAE recovery of the G crystal ($0.282 \pm 0.024 \,\mathrm{J\,m^{-2}}$), which is in good agreement with our IAE value of $0.268 \pm 0.028 \,\mathrm{J\,m^{-2}}$ at much lower temperature (300 °C), further confirming the dominant contribution of the vdW force to the IAE improvement.

In conclusion, we have used an AFM-assisted nanomanipulation technique to directly and precisely measure the weak interlayer vdW bonding at the fresh and aged vdW homointerfaces/heterointerfaces. Highly stronger interactions at the homointerface of $MoS_2$ than G and hBN suggested possible sharing electrons in the interlayer region of TMDs beyond a simple vdW-only interaction. After quantifying the effect of airborne contaminants and humidity on the interfacial adhesion level, we revealed to what degree contaminated heterointerfaces can recover their IAE upon thermal annealing through precise temperature control of nanocontact interfaces. We showed that the precooling treatments can significantly improve the interfacial adhesion of the hBN, G, and $MoS_2$ crystals regardless of the subsequent annealing temperature. Our combined experimental and atomistic analysis also suggested that the formation of short-range chemical bonds only in $G/SiO_x$ heterostructures can elucidate the mechanistic origin of the distinctive strong adhesion behavior between G and $SiO_x$ beyond the widely accepted vdW interaction. Our precise nanoscale quantification of weak interlayer vdW bonding in 2D materials and vdW heterostructures not only provides a reliable basis for theoretical calculations but also can be of fundamental and technological importance for the mass production and continued development of such promising materials in modern electronic devices.

## Methods

**AFM-assisted experimental set-up.** All AFM measurements were performed under controlled ambient conditions ($T = 22$ °C and 15% RH) by a Park XE-70

microscope, which was isolated from mechanical floor vibration by a microscope vibration isolator and also from acoustic vibration, ambient light disturbance, and air flow by a closed box. We determined the noise floor of our AFM set-up to be consistently <0.3 Å throughout the measurements. Three small pieces of $SiO_x/Si$ substrate were simultaneously loaded onto the AFM stage, including 2D crystal flakes mechanically exfoliated with adhesive tape on microheater arrays that are prefabricated on the $SiO_x/Si$ substrate (piece#1), 25-nm-thick polymer glue (poly (3,4-ethylenedioxythiophene):poly(styrene sulfonate) with D-sorbitol (PEDOT:PSS (D-sorbitol))) coated on the $SiO_x/Si$ substrate (piece#2), and pre-patterned bulk 2D crystal stamps with 50–100-nm-thick square and circular nanomesas of 55–65 nm in width and diameter, respectively (piece#3). To minimize the effect of the relative tilt angle, all three pieces were attached to a larger piece of $SiO_x/Si$ substrate precoated by the ultrathin glue film, followed by placing the larger piece onto a multistage Peltier cooling element equipped with a tilt control mechanism (angle resolution: ±0.5°) beneath the cooling stage (Supplementary Note 1).

**Surface preparation of 2D crystal flakes.** Instead of immediately removing all 2D crystal-loaded adhesive tapes from piece#1 to complete the mechanical exfoliation onto the microheaters, we only peeled off the tape containing the 2D crystal flakes of interest (G, hBN, or $MoS_2$) for the interfacial adhesion measurements, thereby enabling much better control over the possible adsorption of airborne contaminants onto the fresh surface of 2D crystals. We prepared aged substrates by two different aging conditions: (1) the freshly exfoliated 2D crystal substrate and the bare $SiO_x/Si$ substrate were directly exposed to the ambient air for 1 h at room temperature; (2) 2D crystal flakes were freshly exfoliated on a precooled (−15 °C) $SiO_x/Si$ substrate and the sample was kept at this temperature for 15 min, followed by the air exposure of the exfoliated 2D crystal substrate and the underlying $SiO_x/Si$ substrate for 1 h at room temperature. A similar method was used to aged tip-attached nanomesas for the subsequent contact with their corresponding aged substrates where fresh 2D crystal tips were simply obtained by our previously developed AFM-assisted shear exfoliation technique[2] (Supplementary Note 1).

**Fabrication of nano-sized 2D crystal mesas.** An ~100-nm-thick bilayer of polymethyl methacrylate (PMMA) 495 K (60 nm)/950 K (40 nm) is spin coated onto the freshly cleaved surface of 1-mm-thick highly oriented pyrolytic graphite (HOPG) (SPI, Grade 1, with a mosaic spread value of 0.4°), hBN (grade A, with single-crystal domains over 100 μm), and $MoS_2$ (429MS-AB, natural single crystals from Canada) substrates, baked each layer for 10 min at 120 °C to evaporate the solvent, and then patterned by electron beam lithography. After developing the exposed PMMA area in 1:3 MIBK/NMP, a 10-nm-thick aluminum film is deposited by thermal evaporation, followed by lift-off process in acetone. The unprotected HOPG, hBN, and $MoS_2$ areas are thinned down by using a reactive ion etching system with pure $O_2$ (precursor flow rate: 10 sccm, RF power: 40 W, pressure: 10 mTorr), $CHF_3/Ar/O_2$ (10/5/2 sccm, 30 W, 10 mTorr) and $SF_6$ (20 sccm, 100 W, 20 mTorr) reactive gases, respectively. Square (circular) mesas with a width (diameter) of 55–65 nm and etch depth of 50–100 nm emerge from 2D crystal substrates during the plasma etch. After plasma etching, the sample is soaked in 0.1 mol/l KOH water solution for ~3 min to remove the Al layer, followed by an annealing process at 600 °C under constant $Ar/H_2$ flow for 1 h to

remove any resist/metallic residues from 2D crystal substrates (Supplementary Notes 1.1 and 1.2).

**Attachment of 2D crystal nanomesas to the AFM tip.** For $F-d$ measurements, a highly doped silicon AFM probe (NANOSENSORS$^{TM}$, ATEC-FM, with a nominal spring constant of 2.8 N m$^{-1}$ and a typical tip radius of curvature better than 10 nm) was used where the tip is positioned at the very end of the cantilever and pointing outward, which provides a more accurate positioning of the tip apex. Since our experiments require a flat plateau at the apex parallel to the piece#1 surface on which all interfacial adhesion measurements were conducted, we scanned the tip in contact mode on its SiO$_x$ surface to achieve an atomically flat surface with a root mean square (RMS) roughness of <1 nm. The in situ flattened tip was next moved from piece#1 to piece#2 and coated with a very thin layer of polymer glue by putting the tip apex in gentle contact with the PEDOT:PSS(D-sorbitol) film. For the precise attachment of 2D crystal nanomesa to the glue-coated flattened apex, the tip was moved from piece#2 to piece#3, followed by locating the nanomesas by the non-contact AFM topographic measurements. The glue-coated tip apex was then moved to the center of the selected 2D crystal nanomesa and held in contact with the nanomesa for 10 min. Afterwards, the tip was gently pulled away from the substrate surface in a direction perpendicular (parallel) to the single basal plane of 2D crystal, leading to pulling off (shearing) the upper section of the nanomesa (attached to the tip apex) from the lower section (fixed to the 2D crystal substrate) (Supplementary Note 1.3). Compared to 2D crystal micro-sized mesas, the tip-attached nano-sized mesas alone offered four striking features in our set-up: (1) the presence of a single-crystalline grain across the whole nanomesa is guaranteed, enabling an atomically defect-free contact interface; (2) substantially more reliable and robust IAE measurements can be achieved under any possible small relative tilting angle between the tip-attached mesa and the substrate, assuring perfect face-to-face contact during approach–retract tip manipulation (Supplementary Fig. 8); (3) high-resolution topographic images in non-contact mode can still be taken to locate 2D crystal flakes of interest for the subsequent IAE measurements (see the AFM image taken by the G crystal tip in the left inset of Fig. 1a); and (4) the nano-sized contact area with a significantly smaller interfacial adhesion force allows using the AFM probe with a lower spring constant and thus higher force resolution.

**Temperature control of nanocontact interface.** While an AFM Peltier-based cooling stage was used to probe the IAE of 2D crystal tips on substrate surfaces at subzero temperature (−15 °C), the IAE measurements at elevated temperatures (up to 300 °C) on the target substrates were conducted by means of localized Joule heating of ultralow power microheaters (Supplementary Notes 1.4 and 1.5). Using the microheater can not only significantly alleviate the adverse effect of high temperature on the AFM probe by locally heating the substrate (Supplementary Fig. 4) but also provide a uniform temperature distribution over the heated 2D flakes, which are in direct contact with the heating lines (Supplementary Figs. 5 and 6). For each temperature change, enough time was given to the 2D flakes to reach steady-state temperature. Then the 2D crystal tip was engaged with the sample surface at a pressure of 5 MPa (unless otherwise noted) for 15 min to reach thermal equilibrium and then the substrate cooled back down to room temperature to perform the interfacial adhesion measurements. Similarly, we conducted a series of the interfacial adhesion measurements at subzero temperature by first cooling the substrate surface down to −15 °C using a multistage Peltier element and then removing the 2D crystal-loaded adhesive tape from piece#1 (Supplementary Note 1.5).

**Force–displacement measurements.** All retraction $F-d$ curves between the 2D crystal tips and the untreated/precooling-treated substrates were obtained under controlled ambient conditions in the near-equilibrium regime, which was achieved by an ultralow noise floor of <0.3 Å, an ultralow noise AFM controller with the $Z$ scanner's vertical resolution of better than 0.1 Å, and also using a very slow (quasi-static) pulling rate of 1 nm/s. Very careful adjustment of the $Z$ servo gain to suppress any possible oscillation of the $Z$ scanner could further make the retraction measurements in the near-equilibrium regime possible (Supplementary Note 2). In order to calculate the IAE per unit area ($\Gamma$, J m$^{-2}$) from the recorded retraction–displacement curves, we integrate the retraction force as a function of the piezo displacement (light blue-shaded area in Fig. 1b, d), followed by dividing the resulting adhesion energy by the known contact area at the interface. However, in order to extract the IAE from the shear $F-d$ curves, the interfacial adhesion force opposing new surface formation is first obtained as $F_a(x) = \Gamma[\mathrm{d}A(x)/\mathrm{d}x]$, where $x$ represents the lateral displacement of the mobile section of the mesa with respect to the initial position, and $A(x)$ is the overlap area of the top and bottom sections of the mesa as a function of $x$. For a square mesa of width $w$ and a circular mesa of diameter $D$, the maximum shear force, $F_s^m$, in the shear $F-d$ curves that is required to initiate sliding (i.e., $F_a$ at $x = 0$) can be related to the IAE by $\Gamma w$ and $\Gamma D$, respectively (Supplementary Note 2.1). While the cleavage strength can be obtained by $\sigma_{33} = P/A$, where $P$ is the pull-off force and $A$ is the interface area, by definition of the interlayer shear strength at the sliding interface, $\tau_s = F_s^m/A$ (Supplementary Note 2.1). The accuracy of $F-d$ measurements can be limited by the uncertainty in the determination of the interfacial contact area and spring constant of the AFM

probe (Supplementary Note 2.2). In order to create a known contact area, we used 2D crystal tips with a very well-defined geometric shape parallel to the substrate, enabling an atomically clean and flat contact interface. Our interfacial adhesion measurements reveal that the tilting angle between the tip and the substrate is smaller than 1°, indicating perfect face-to-face contact during measurements (Supplementary Note 2.3). We reduced the second main source of uncertainty in our measurements by determining the stiffness of the AFM cantilever by means of three different methods and took their mean value as the static normal (3.05 ± 0.05 N m$^{-1}$) and axial (8.60 ± 0.40 N m$^{-1}$) spring constants of the probe, suggesting a relative calibration error of 2 and 5%, respectively (Supplementary Note 2.4). The same AFM probe was used throughout the experiments to ensure accurate correlation between all interfacial adhesion measurements. The laser spot was also kept at the same position on the lever to avoid any changes in the force measurements. After performing all the measurements, the spring constant of the probe was again determined in ambient conditions to make sure that the cooling/local annealing of the substrate has no appreciable effect on its stiffness, yielding the spring constant still within the uncertainty range of our measurements. We also note that the random crystalline orientation at the interface of 2D crystal tips and 2D crystal substrates has no appreciable effect on the IAE measurements (Supplementary Note 2.5). Our further analysis of $F-d$ curves also confirms the negligible effect of the possible edge functionalization of nanomesas (due to the etching process) on the IAE measurements (Supplementary Note 2.6).

**Adhesion of G/SiO$_x$ under different contact pressures.** We conducted a series of interfacial adhesion measurements over a pressure range of 0–10 MPa at the interface of G crystal tip/pre-annealed SiO$_x$ substrate and G crystal tip/pre-annealed G substrate. To further minimize experimental uncertainty, a 2 μm × 2 μm smooth region of the SiO$_x$ (G) substrate with an RMS surface roughness of 0.305 nm (0.077 nm) was first located by non-contact AFM roughness measurements and then 10 contacts with 100-nm interval spacing were formed at each pressure load under a very clean environment, allowing us to perform all measurements within a very small region in close proximity to the microheater (see the SEM image in the inset of bottom panel in Fig. 3b). Moreover, prior to each pressure increment, SiO$_x$ and G substrates are annealed at 300 °C for 30 min to remove any possible adsorbed contaminations and then the G crystal tip/pre-annealed substrate interface is further annealed at 300 °C for 15 min, followed by the new round of adhesion measurements.

**3D surface topography measurements.** Monolayers of G, hBN, and MoS$_2$ square flakes of 10 μm in width were shear exfoliated from a flat polydimethylsiloxane stamp onto the pre-annealed SiO$_x$ substrate at a contact pressure of 5 MPa. We used an ultrasharp tip with 2 nm nominal radius of curvature (<5 nm guaranteed) and sparing constant of 39.1 N m$^{-1}$ in the non-contact mode and in the attractive regime (with a frequency shift of −10 Hz and free amplitude of 7.5 nm) under ambient conditions and then determined the noise floor of the AFM system, being consistently <0.3 Å. To provide a more accurate and comprehensive description of the surface roughness both in vertical and lateral directions, the AFM image data were analyzed by the 2D PSD function rather than the standard RMS roughness (Supplementary Note 3).

**X-ray photoelectron spectroscopy.** XPS measurements were performed on both freshly exfoliated and pre-annealed MoS$_2$ samples at excitation energy of 1486.6 eV. Residual electrostatic charging effects were taken into account by applying an offset to the spectra with a reference signal of C (1$s$) at a binding energy of 284.6 eV. In order to eliminate any interference between the dominant Mo (3$d$) and S (2$s$) features, we used the less intense Mo (3$p$) photoelectron signal for the quantification purposes. Each set of peaks was fitted by a 70% Gaussian–30% Lorentzian function. In addition, peaks of spin–orbit doublets Mo (3$p_{3/2}$ and 3$p_{1/2}$) were set to have an area ratio in accordance with quantum degeneracy values (i.e., 2:1 for 3$p_{3/2}$ and 3$p_{1/2}$ orbitals; Supplementary Note 4).

**Classical MD simulations.** To gain an in-depth understanding of underlying mechanisms associated with the interaction of G crystal and the SiO$_x$ substrate, we performed classical MD simulations using the LAMMPS simulator at room temperature. Four 98.2 Å × 102.1 Å G layers with AB stacking were placed at a distance of 3.0 Å above an amorphous SiO$_x$ substrate (143.3 × 146.5 × 21.3 Å$^3$) while the flattened tip was modeled by a tapered silicon (001) layer. To hold the system in space, 2 Å of the SiO$_x$ substrate from the bottom was treated as rigid throughout the simulation. We adopted reactive empirical bond order potential function to model the intralayer carbon–carbon interactions within the same G layer while the free G edges were passivated by hydrogen. A registry-dependent interlayer potential that can accurately describe the overall cohesion, corrugation, equilibrium spacing, and compressibility of few-layer G was implemented in the LAMMPS code to model the carbon–carbon interaction between G flakes. Tersoff potential and Stillinger–Weber potential were utilized for the modeling of SiO$_x$ substrate and silicon (001) layer, respectively. We used a standard 12-6 Lennard-Jones potential for describing Si-C and O-C long-range vdW interactions according to the Universal Force Field model and the Lorentz–Berthelot mixing rules, whereas O-C and Si-C covalent bonds at the G/SiO$_x$ interface were modeled by the Tersoff potential.

The glue between the tip and the few-layer G nanomesa was simply modeled by applying the Lennard-Jones potential between the silicon layer and the topmost G layer using a larger Si–C interaction energy. The calculations were conducted in the NVT ensemble using the Nosé–Hoover thermostat and Newton's equations of motion were integrated using the velocity Verlet algorithm with a time step of 1 fs. The total interfacial force (i.e., vdW and non-vdW forces) and relative displacement between the innermost G layer and the $SiO_x$ substrate were simultaneously monitored as the tapered silicon (001) layer was pulled in the normal direction with a constant speed of $10^{-2}$ Å/ps.

## Data availability

The authors declare that the data supporting the findings of this study are available within the paper and its supplementary information files.

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

## Acknowledgements

We acknowledge support from Michigan Center for Materials Characterization $(MC)^2$ and Lurie Nanofabrication Facility (LNF) at the University of Michigan. This work was supported in part by the National Science Foundation under Grant No. CMMI-1636132 and in part by the University of Michigan Rackham Predoctoral Fellowship. We also thank Saeedeh Noroozi from Molecular & Behavioral Neuroscience Institute (MBNI) at the University of Michigan for her assistance with MD simulations.

## Author contributions

H.R. and W.L. conceived the work. H.R. fabricated the devices and performed the experiments. H.R. carried out the finite element and MD simulations and analyzed the data. H.R. and W.L. contributed in the discussion and interpretation of the results. H.R. wrote the manuscript with comments from W.L.

## Competing interests

The authors declare no competing interests.
