## [Peer Review File · Nature Communications]

Reviewers' comments:

Reviewer #1 (Remarks to the Author):

The issue of interfacial adhesion in 2D materials is an important issue given the tremendous interest in various 2D heterostructures in recent years. The attempt to quantify the adhesion between different surfaces by the authors is timely and potentially of high impact. The authors devised a new experimental method to estimate the adhesion energy and also to evaluate the effect of different annealing and/or surface treatments. The work is very extensive with a large amount of data, which can attract interest from the researchers in the field.

Unfortunately, the details of the experiments are not transparent in the paper, which makes it difficult to grasp the key results. The authors should revise the manuscript substantially, with additional measurements, if necessary.

1. The authors explain the experimental method in the main text and also in the supplementary information. However, I had a hard time figuring out what the authors actually measured. For example, In Fig. 2a (ditto in Fig. 2b&c), the figure seems to suggest that the authors measured the adhesion between a monolayer graphene on SiO_x/Si and the graphene on the AFM tip. However, I am not sure if this is the case because nowhere in the manuscript was I able to find the description on the sample on the substrate. For each measurements, the authors should explicitly describe what they are measuring: monolayer graphene to monolayer graphene, monolayer graphene to thick graphite, graphite to graphite, etc. If the sample on the substrate is not a monolayer, its thickness data should be provided.

2. Figure 3b suggests that the adhesion energy between graphene layers is larger than the adhesion energy between graphene and the SiO_x substrate at all contact pressure. How can one exfoliate monolayer graphene on SiO_x substrates? Shouldn't the adhesion between the substrate the first layer of graphene be larger than the adhesion between the graphene layers in order to obtain monolayer?

3. One critical parameter that was not controlled in this work is the twist angle between the top and bottom layers. It is well known that the adhesion between different materials depends on the twist angle. The authors should devise a way or carry out additional TEM measurements to determine the twist angles.

4. I am not sure if the findings can be translated to real situation directly. In real situation, there are areas in the interface where bubbles form or residues are trapped. Such irregularities may dominate the actual adhesion. The current work on the other hand deals with the 'intrinsic' adhesion in which bubbles or large amount of residues are not involved. I am not saying this is not important. But I would like to see how the current findings can be translated into the real situation.

Reviewer #2 (Remarks to the Author):

The authors reported a set of experimental measurement results of the homo-interfacial and hetero-interfacial interactions between 2D crystals and the interfacial interaction between 2D crystals and substrate SiO_x with AFM tip-attached 2D crystal nanomesas under ambient conditions. They adopted a precooling-treated method to handle the substrate and examined the interfacial interactions on both the intact and aged samples under different annealing temperatures to quantify the effect of airborne contaminants and humidity on the interfacial adhesion level. Abrupt detachment of graphite (G) nanomesa on SiO_x at high annealing temperature was observed and in view of that, the authors exhibit the key factor of conformity of the tip-attached G nanomesa to the underlying substrate morphology.

The experimental method and results are interesting, but the manuscript is far from mature for publication. The following questions and concerns should be clarified.

Within the procedure of attaching the 2D crystal nanomesas to AFM-tip, the initial detaching F-d curve is crucial because it could ideally provide the cohesion energy with perfect, clean, aligned interface without absorptions from ambient air. Whether the F-d curves which are measured at intact homointerfaces showed in Figs.1b, 2a represent the initial curves? Are all the later measurements at the same bottom nanomesa included the influence of absorptions from ambient air? What's the difference between the initial curve and later measurements at the same bottom nanomesa?

Main concerns :

To measure the interfacial interaction of 2D layered crystals, vacuum environment should be necessary, as the cleavage surfaces can be contaminated soon when exposed to air.

The scatter in such measurements is inevitable, how many measurements are made and how stable the measurement during all the measuring procedure should be clearly presented.

The long introduction is nearly no relevant to the topic of the work. Literatures of related theoretical and experimental investigation on the interfacial interaction should be introduced and carefully discussed, and previous results should be compared.

Minor technique issues:

1) A relatively gradual reduction of the interfacial adhesion force is observed at homointerfaces (in manuscript, page 3, paragraph 2, line 10). It is stated that the short-range vdW interaction is the major separation mechanism in 2DLMs (in Supplementary Information, page 17, line 13). But in Fig. 1b, the distance of gradual reduction is $\sim 10\text{nm}$, which is larger than general feature distance of short-range vdW interaction. Can the authors respond to that inconsistency?

2) Can the authors explain the reason that why only the graphite was etched to circle shape compared with BN and MoS₂.

3) In Fig. 1c, the shear force of MoS₂ keeps stable and G goes down gradually. What is the mechanism?

4) Among the measurements of 2D crystal heterointerfaces in Fig. 2b, the tip-nanomesa seems have already stayed in ambient air for a long time and the tip-nanomesa may become so-called untreated (with the definition in manuscript, page 3, line 10) tip-nanomesa. Is there any difference on comparing the results of treated and untreated substrates with the influence of untreated tip-mesa.

5) In Fig. 2a, IAE values as a function of annealing temperatures at the intact hBN homointerface, it is evident that the IAE is not independent of the annealing temperature, especially at 300 °C (inconsistent to conclusion in manuscript, page 5, paragraph 2, line 6). Based on that, the conclusion which is stated that nearly full recovery of the intrinsic IAE only at the hBN homointerface upon annealing at 300 °C (in manuscript, page 6, line 2) need to be discussed.

6) In Fig. 2b, at the MoS₂/G interface, the mechanism that the IAE drops when the annealing temperature changes from -15 to 22 °C should be explained.

7) In Fig. 3a, what's the proportion of the existence of point of bond at edge among all the measured curves under 5 MPa? And I think in the process of the fabrication of 2D crystal nanomesa, the etching procedure would functionalize or chemically modify the edge of the nanomesa, and the edge would become more active than pure edge. The phenomena of bond at edge may be from that functionalized edge. (in manuscript, page 10, line 2).

8) Some writing errors need to be revised. E.g. adhesive tape (Methods, page 12, line 9 in manuscript and in Supplementary Information, page 2, paragraph 2, line 3).

Reviewer #3 (Remarks to the Author):

In this manuscript, the authors performed comprehensive quantitative measurements on the interfacial adhesion energy (IAE) of two-dimensional (2D) layered materials, van der Waals heterostructures and 2D materials on SiO₂ substrates. Their results showed that the MoS₂ has the

maximum IAE, compared to graphene and h-BN, independent of the annealing temperatures. They also quantified the effect of airborne contaminants and humidity on the interfacial adhesion level and revealed that the thermal annealing can sufficiently affect the IAE at both the contaminated homo- and heterointerfaces. Regarding the IAE on SiO₂ substrates, they measured the highest value for graphene and the lowest one for the MoS₂, attributed to their different degrees of conformation to the SiO₂ and the formation of short-range chemical bonds in the G/SiO_x. These results, if reliable, would be highly appealing to the community of 2D materials, from both theoretical and experimental points of view. However, since the measurement on vdw-like force of 2D materials is very tricky, the authors need to further confirm their results and strengthen their conclusions by performing further characterizations. The authors should adequately address my following issues before I can make any recommendation.

1. The authors glued multilayer 2D materials of different shapes to the AFM tips and then measured the interlayer interaction force between the tip-attached 2D materials nanomesa and the underlying samples. I have a concern that the interlayer interaction within the nanomesa is also dominated by vdw interaction, which may compete with the vdw-like tip-sample interaction and lead to big errors and even wrong results. How can the authors make sure that the force is measured just at the tip-sample interface, not within the nanomesa.
2. Previous measurements reported a stronger van der Waals interaction at the graphite-BN interface than that at the graphite-MoS₂ interface, while the authors' results do not follow this trend clearly. They should explain why and make detailed comparison to previous results. The interlayer interaction force of 2D materials has been measured by many groups, at least for graphene.
3. The thermal fluctuation at an increased temperature will increase the interlayer distance of 2D materials and therefore tend to decouple the interlayer interaction. This, however, is in contrast to the authors' results that the IAE increases with increasing the temperature. I'm not convinced by their analyses and explanation.
4. The MoS₂ has a higher chemical activity than graphene. Why graphene is shown to chemically bond to SiO₂ but MoS₂ does not? Here, I'm also not convinced by the mechanism proposed for explaining the larger IAE of graphene on SiO₂.
5. All abbreviations in this paper, such as 'G', 'SEM' and 'HOPG', need explanations.
6. For the force-displacement measurements, how to make sure the pressure exerted by the nanomesas to the sample is close to zero so that the displacement is governed only by the intrinsic interfacial interaction force.

Response to Reviews

We thank the reviewers for their careful reviews and valuable suggestions; we have addressed all the comments in the revised manuscript. All revised parts were highlighted in the manuscript.

Reviewer #1 (Remarks to the Author):

The issue of interfacial adhesion in 2D materials is an important issue given the tremendous interest in various 2D heterostructures in recent years. The attempt to quantify the adhesion between different surfaces by the authors is timely and potentially of high impact. The authors devised a new experimental method to estimate the adhesion energy and also to evaluate the effect of different annealing and/or surface treatments. The work is very extensive with a large amount of data, which can attract interest from the researchers in the field.

We thank the respected reviewer for his/her valuable inputs and comments on the manuscript.

Unfortunately, the details of the experiments are not transparent in the paper, which makes it difficult to grasp the key results. The authors should revise the manuscript substantially, with additional measurements, if necessary.

Following the reviewer's comment, we substantially revised the main text in order to make it easier to follow and more independent from the Supplementary Information. In the following, we provide a point-by-point response to the respected reviewer's comments.

1. The authors explain the experimental method in the main text and also in the supplementary information. However, I had a hard time figuring out what the authors actually measured. For example, In Fig. 2a (ditto in Fig. 2b&c), the figure seems to suggest that the authors measured the adhesion between a monolayer graphene on SiO_x/Si and the graphene on the AFM tip. However, I am not sure if this is the case because nowhere in the manuscript was I able to find the description on the sample on the substrate. For each measurements, the authors should explicitly describe what they are measuring: monolayer graphene to monolayer graphene, monolayer graphene to thick graphite, graphite to graphite, etc. If the sample on the substrate is not a monolayer, its thickness data should be provided.

In this study, we have reported adhesion measurements through well-defined interactions of AFM tip-attached 2D crystal nanomesas with mechanically exfoliated 2D crystal substrates and SiO_x substrate. As such, we only measured the adhesion between ultrathin 2D crystals, such as graphite/graphite (G/G), G/hBN, etc. Following the reviewer's comment, we modified the main text as follows:

Page 2, Paragraph 4:

To this end, force–displacement (F – d) curves with piconewton–subnanometer resolution are recorded upon retraction of AFM tip-attached 2D crystal nanomesas from tens-of-nm-thick fresh and aged 2D

crystal substrates and bare SiO_x/Si substrate under controlled ambient conditions in the near equilibrium regime (Fig.1a).

2. Figure 3b suggests that the adhesion energy between graphene layers is larger than the adhesion energy between graphene and the SiO_x substrate at all contact pressure. How can one exfoliate monolayer graphene on SiO_x substrates? Shouldn't the adhesion between the substrate the first layer of graphene be larger than the adhesion between the graphene layers in order to obtain monolayer?

As mentioned in the manuscript, Fig. 3b suggests the significant contribution of the conformal adhesion to the overall interfacial adhesion strength of the G/SiO_x interface. As a result, the interfacial adhesion between G/SiO_x is pressure dependent and could be weaker or stronger than the adhesion between G/G depending on the number of short-range chemical bonds formed at the G/SiO_x interface. When the adhesion of G/SiO_x is stronger than that of G/G, the separation takes place across the thickness of the nanomesa (i.e., between G/G, please see blue squares in Fig. 3b) and therefore we are not able to measure the strong adhesion at the G/SiO_x interface. In other words, we can only measure the adhesion of G/SiO_x weaker than that of G/G (red circles in Fig. 3b).

To make it clear, we modified the main text as follows:

Page 8, Paragraph 4:

To this end, a series of interfacial adhesion measurements over a pressure range of 0-10 MPa was conducted at the interface of G crystal tip/pre-annealed SiO_x substrate (top panel of Fig. 3b) and G crystal tip/pre-annealed G substrate (bottom panel of Fig. 3b). This setup only allowed us to study the interaction of G/SiO_x weaker than that between G/G (red circles in Fig. 3b), otherwise the separation takes place across the thickness of G nanomesa (blue squares in Fig. 3b).

3. One critical parameter that was not controlled in this work is the twist angle between the top and bottom layers. It is well known that the adhesion between different materials depends on the twist angle. The authors should devise a way or carry out additional TEM measurements to determine the twist angles.

The first direct, accurate experimental measurement of the interfacial adhesion energy of the graphite using the micro-force sensing probe shows that the interlayer twist angle ranging from 0° (perfect AB stacking) to 54° has only a weak effect (about 5.4%) on the interfacial adhesion energy [1]. Moreover, the critical adhesion forces of G-wrapped AFM tip on G, hBN and MoS₂ substrates reveal that the random crystalline orientation has negligible effect on the interfacial adhesion force measurements whose very small standard deviations of about 1% are closely related to the instability point at which the tip is pulled off from the sample surface [2]. Raman spectroscopy measurements of interfacial coupling in AB and twisted multilayer graphene [3, 4] and MoS₂ [5] also reveal a small effect (5.9% in multilayer graphene and 6.1% in MoS₂) of the interlayer twist angle on the out-of-plane elastic constants which are a measure of the interfacial adhesion energy at least near the equilibrium position of two neighboring layers. Therefore, we believe that the relative orientation of 2D vdW crystals may contribute relatively little to their overall interfacial adhesion energy. This can also be supported by comparing our standard deviations of the measured interfacial adhesion energy at intact G/G, hBN/hBN and MoS₂/MoS₂ interfaces (at which we expect

perfect AB stacking with no contamination) and those at the pre-cooling treated G/G, hBN/hBN and MoS₂/MoS₂ interfaces (at which we believe there could be interlayer lattice mismatch with small amount of contamination). The average of the standard deviations of the interfacial adhesion energy at the intact and the pre-cooling treated interfaces is ± 0.027 and ± 0.029 (G/G), ± 0.026 and ± 0.033 (hBN/hBN) and ± 0.033 and ± 0.037 (MoS₂/MoS₂), respectively, indicating 7%, 27% and 12% increase, respectively, in the standard deviations due to the effect of both interlayer lattice misorientation and the interfacial contamination.

[1] W. Wang, S. Dai, X. Li, J. Yang, D. J. Srolovitz and Q. Zheng, Measurement of the cleavage energy of graphite, *Nature Communications*, vol. 6, p. 7853, 2015.

[2] B. Li, J. Yin, X. Liu, H. Wu, J. Li, X. Li and W. Guo, "Probing van der Waals interactions at two-dimensional heterointerfaces," *Nature Nanotechnology*, vol. 10.10138, pp. s41565-019-0405-2, 2019.

[3] J.B. Wu, Z.X. Hu, et al. Interface coupling in twisted multilayer graphene by resonant Raman spectroscopy of layer breathing modes. *ACS Nano*, 9 (2015) 7440-7449.

[4] C. Cong, T. Yu. Enhanced ultra-low-frequency interlayer shear modes in folded graphene layers. *Nature Communications*. 5 (2014) 4709.

[5] K. Jin, D. Liu, Y. Tian. Enhancing the interlayer adhesive force in twisted multilayer MoS₂ by thermal annealing treatment. *Nanotechnology*, vol. 26, 405708, 2015.

Following the reviewer's comment, we added the following section to Supplementary Information, Page 11, as follows:

Section S2.5. Effect of crystalline orientation on IAE measurements

Owing to the random crystalline orientation at the interface of 2D crystal tips and 2D crystal substrates, the effect of such interlayer lattice mismatch on the IAE measurements should be investigated. A comparison between our standard deviations of the measured interfacial adhesion energy at intact G/G, hBN/hBN and MoS₂/MoS₂ interfaces (at which we expect perfect AB stacking with no contamination) and those at the pre-cooling treated G/G, hBN/hBN and MoS₂/MoS₂ interfaces (at which we believe there could be interlayer lattice mismatch with small amount of contamination) reveals that the relative orientation of 2D vdW crystals may contribute relatively little to their overall interfacial adhesion energy. The average of the standard deviations of the interfacial adhesion energy at the intact and the pre-cooling treated interfaces is ± 0.027 and ± 0.029 (G/G), ± 0.026 and ± 0.033 (hBN/hBN) and ± 0.033 and ± 0.037 (MoS₂/MoS₂), respectively, indicating 7%, 27% and 12% increase, respectively, in the standard deviations due to the effect of both interlayer lattice misorientation and the interfacial contamination. This is also confirmed by the first direct experimental measurement of the interfacial adhesion energy of the G crystal using the micro-force sensing probe, showing that the interlayer twist angle ranging from 0° (perfect AB stacking) to 54° has only a weak effect (about 5.4%) on the interfacial adhesion energy [7]. Moreover, the critical adhesion forces of G-wrapped AFM tip on G, hBN and MoS₂ substrates reveal that the random crystalline orientation has negligible effect on the interfacial adhesion force measurements whose very small standard deviations of about 1% are closely related to the instability point at which the tip is pulled off from the sample surface [8].

4. I am not sure if the findings can be translated to real situation directly. In real situation, there are areas in the interface where bubbles form or residues are trapped. Such irregularities may dominate the actual adhesion. The current work on the other hand deals with the 'intrinsic' adhesion in which bubbles or large

amount of residues are not involved. I am not saying this is not important. But I would like to see how the current findings can be translated into the real situation.

While adhesion measurements of 2D crystals in literature have been performed either in high vacuum or under a contamination-free environment [6, 7], airborne contaminants are an inevitable part of any vdW heterostructures, as the respected reviewer correctly mentioned. Therefore, addressing quantitatively to what degree their interfacial adhesion energy is influenced by interfacial contaminants and nanoblisters and how to effectively remove them is of fundamental and technological importance for the continued development of such promising materials. As an important step toward this goal, we quantified the effect of airborne contaminants and humidity on the interfacial adhesion level and revealed to what degree contaminated heterointerfaces can recover their interfacial adhesion energy upon thermal annealing. We even took one step further and showed that a simple but very effective precooling treatment can significantly improve the interfacial adhesion of hBN, G and MoS₂ regardless of the subsequent annealing temperature.

[6] W. Wang, S. Dai, X. Li, J. Yang, D. J. Srolovitz and Q. Zheng, Measurement of the cleavage energy of graphite, *Nature Communications*, vol. 6, p. 7853, 2015.

[7] B. Li, J. Yin, X. Liu, H. Wu, J. Li, X. Li and W. Guo, "Probing van der Waals interactions at two-dimensional heterointerfaces," *Nature Nanotechnology*, vol. 10.10138, pp. s41565-019-0405-2, 2019.

We thank the reviewer for the question. Owing to the importance of translating the current findings into the real situation, the above discussion along with the significance of accurate quantification of interfacial adhesion of 2D crystals in many different practical applications were discussed in the revised **Introduction, Paragraphs 1-5.**

Reviewer #2 (Remarks to the Author):

The authors reported a set of experimental measurement results of the homo-interfacial and hetero-interfacial interactions between 2D crystals and the interfacial interaction between 2D crystals and substrate SiO_x with AFM tip-attached 2D crystal nanomesas under ambient conditions. They adopted a precooling-treated method to handle the substrate and examined the interfacial interactions on both the intact and aged samples under different annealing temperatures to quantify the effect of airborne contaminants and humidity on the interfacial adhesion level. Abrupt detachment of graphite (G) nanomesa on SiO_x at high annealing temperature was observed and in view of that, the authors exhibit the key factor of conformity of the tip-attached G nanomesa to the underlying substrate morphology. The experimental method and results are interesting, but the manuscript is far from mature for publication. The following questions and concerns should be clarified.

We thank the respected reviewer for his/her valuable inputs and comments on the manuscript. In the following, we provide a point-by-point response to the reviewer's comments.

Within the procedure of attaching the 2D crystal nanomesas to AFM-tip, the initial detaching *F-d* curve is crucial because it could ideally provide the cohesion energy with perfect, clean, aligned interface without absorptions from ambient air. Whether the *F-d* curves which are measured at intact homointerfaces showed in Figs.1b, 2a represent the initial curves? Are all the later measurements at the same bottom nanomesa included the influence of absorptions from ambient air? What's the difference between the initial curve and later measurements at the same bottom nanomesa?

As we showed schematically in Figs. S1a-d in Supplementary Information, during the attachment of nanomesas to the glue-coated tip, *F-d* curves for the intact interfaces (Fig. 1b in the main text) can be recorded as the tip is gently pulled away from the substrate, leading to pulling off the upper section of the nanomesa (attached to the tip apex) from the lower section of the nanomesa (fixed to the 2D crystal substrate) (Fig. S1a). This allowed us to report the cohesion energy of intact G/G, hBN/hBN and MoS₂/MoS₂ in Fig. 1e (not in Fig. 2a). However, in order to measure the interfacial adhesion of G/G, hBN/hBN and MoS₂/MoS₂ in the presence of airborne contaminants, we brought the AFM tip-attached 2D crystal nanomesas into contact with the mechanically exfoliated 2D crystal substrates (Fig. S1c), where both nanomesa and substrate adsorb airborne contamination upon the air exposure. We then reported the interfacial adhesion energy of contaminated G/G, hBN/hBN and MoS₂/MoS₂ upon retraction of the 2D crystal tip from the substrate surfaces (Fig. 2a).

In short, while the initial *F-d* curves were acquired from the separation of nanomesas (that emerge from 2D crystal substrates during the plasma etch) somewhere across their thickness, the later measurements were performed by the separation of 2D crystal tip from the mechanically exfoliated 2D crystal substrates (not from the same bottom nanomesa). We also observed larger pull-off forces in the initial *F-d* measurements at the intact interfaces than in the later measurements at the contaminated interfaces, well consistent with the reported IAE values.

To make it clear, we added the following paragraph to **Supplementary Information, Section S2, Page 6**:

In order to make sure that the $F-d$ curves are measured at the tip-sample interface not within the nanomesa, we first measured the intrinsic cohesion energy of 2D crystals (**Fig. 1e** and gray circles in **Fig. 2a**), confirming that the cohesion energy across the 2D crystal nanomesa is larger than the interfacial adhesion energy at all 2D crystal tip-sample interfaces. We also observed larger pull-off forces at the intact interfaces compared to contaminated interfaces, well consistent with our reported IAE values. Therefore, the separation most likely takes place at the tip-sample interface rather than somewhere across the thickness of the tip-attached nanomesa. Moreover, for each tip-attached 2D crystal, we formed all contacts with 1 μm interval spacing within the same distance from the heating line, allowing us to easily locate and scan all contact spots (using the non-contact AFM mode) for any possible exfoliation of monolayer or few layers from tip-attached 2D crystal onto the sample. For the spot with exfoliated mono/few-layer 2D crystal, the area under the corresponding $F-d$ curve was considered as the intrinsic cohesion energy rather than the interfacial adhesion energy at the tip-sample interface.

Main concerns: To measure the interfacial interaction of 2D layered crystals, vacuum environment should be necessary, as the cleavage surfaces can be contaminated soon when exposed to air.

While adhesion measurements of 2D crystals in literature have been performed either in high vacuum or under a contamination-free environment [1, 2], airborne contaminants are an inevitable part of any vdW heterostructures, as the respected reviewer is well aware. Therefore, addressing quantitatively to what degree their interfacial adhesion energy is influenced by interfacial contaminants and nanoblisters and how to effectively remove them is of fundamental and technological importance for the continued development of such promising materials. As an important step toward this goal, we quantified the effect of airborne contaminants and humidity on the interfacial adhesion level and revealed to what degree contaminated heterointerfaces can recover their interfacial adhesion energy upon thermal annealing.

[1] W. Wang, S. Dai, X. Li, J. Yang, D. J. Srolovitz and Q. Zheng, Measurement of the cleavage energy of graphite, *Nature Communications*, vol. 6, p. 7853, 2015.

[2] B. Li, J. Yin, X. Liu, H. Wu, J. Li, X. Li and W. Guo, "Probing van der Waals interactions at two-dimensional heterointerfaces," *Nature Nanotechnology*, vol. 10.10138, pp. s41565-019-0405-2, 2019.

Following the reviewer's question and owing to the importance of translating the current findings into the real situation, the above discussion along with the significance of accurate quantification of interfacial adhesion of 2D crystals in many different practical applications were discussed in **Introduction, Paragraphs 1-5, as follows:**

Two-dimensional layered materials (2DLMs), such as graphene, hexagonal boron nitride (hBN), transition metal dichalcogenides (TMDs: e.g., MoS_2 and WS_2) and many others, with strong in-plane covalent bonding and weak out-of-plane van der Waals (vdW) interactions exhibit a unique combination of high elasticity, extreme mechanical flexibility, visual transparency, and superior (opto)electronic performance, making them ideally suited to modern devices, such as photovoltaic devices, hybrid electrochemical capacitors, lithium- and sodium-ion batteries, hydrogen evolution catalysis, transistors,

photodetectors, DNA detection, and memory devices [1]. However, intrinsic ultrahigh surface to volume ratio in 2DLMs requires intimate knowledge of interfacial adhesion between two adjacent layers of similar or dissimilar 2DLMs and also between 2DLMs and their supporting substrate. In particular, such a fundamental mechanical property plays a central role not only in synthesis, transfer and manipulation of 2DLMs but also in fabrication, integration and performance of 2DLMs–incorporated devices.

In general, fabrication of 2D systems involves transferring 2DLMs from their growth substrate or bulk stamp onto a target substrate using different transfer–printing techniques. A better understanding of the adhesion energy between 2DLMs and the various substrates involved is highly desired as an essential step toward enhancing the transfer efficiency and thickness uniformity of printed flakes and thus producing high-quality, large–scale 2D electronic device arrays at micro and nanoscales [2]. The interfacial adhesion between 2DLMs and their neighbors is also an important parameter for the mechanical integrity of the device whose operation is highly influenced by slippage and delamination of 2DLMs during thermal and mechanical loadings. As such, a 2DLM needs to make secure contact not only with supporting substrates and metallic interconnects in 2DLMs–based devices but also with other 2DLMs and encapsulation layers in vdW heterostructure devices [3, 4].

Newly emerged vdW heterostructures – stacks of individual monolayer flakes of different 2DLMs assembled layer by layer – offer a variety of new physical properties, thanks to the full spectrum of electronic properties in 2DLMs, from conducting graphene, to semiconducting TMDs, to insulating hBN. An essential feature of such heterostructures is atomically clean interfaces to achieve the best device performance – any interfacial contamination (e.g., blisters) results in deterioration of transport properties [5]. As such, wet transfer and dry pick-and-lift transfer techniques are widely used for assembly of vdW heterostructures. However, both direct mechanical assembly techniques rely strongly on vdW interactions between the 2D crystals, and, as a result, an accurate quantification of interfacial adhesion between different 2DLMs is crucial for the mass production of blister–free vdW heterostructures.

Fascinating interlayer vdW-dependent properties of similar and dissimilar 2DLMs provide a unique opportunity to study the nature of electronic structure and band alignment, interfacial thermal and electrical resistance, ion intercalation and deintercalation process, interfacial nanofluidic transport and drug delivery behavior, photon absorption and photocurrent/photovoltaic production, interfacial charge polarization and redistribution, spin–orbit coupling and many others in layered materials-based devices [6, 7, 8, 9, 10]. Notably, interfacial electrical, mechanical, optoelectronic, magnetic and thermal properties of layered materials can also interact in a rather complex way. For instance, formation of any delamination-motivated surface corrugations in 2DLMs can give rise to local strain distribution and curvature-induced rehybridization, which modify the electronic structure and local charge distribution, create polarized carrier puddles and dipole moment, induce pseudomagnetic fields and thus alter magnetic, optical and electrical properties as well as chemical surface reactivity [11]. Moreover, the vdW interaction as a key medium for the stress transfer both within and across the interface of 2DLMs can highly impact their thermal and electrical properties in such a way that a 2D layered system can act as a heat conductor or insulator and/or a semimetal or electrical insulator through strain engineering [12, 13, 14].

The interfacial physical and chemical behavior of layered materials becomes even more complicated when we consider that airborne contaminants are an inevitable part of any vdW heterostructures and therefore addressing quantitatively to what degree their interfacial adhesion energy (IAE) is influenced by interfacial contaminants and nanoblisters and how to effectively remove them is of fundamental and technological importance for the continued development of such promising materials. Despite the

significance of such a fundamental property for any layered materials, there have been relatively limited experimental and theoretical methods with significant diversity in the reported IAE values for 2DLMs in general and graphite (G) crystals specifically, where the exact cause of the variation in their IAE values has also remained to be elucidated by a comprehensive and accurate experimental technique [15, 16, 17, 18, 19, 20, 21].

The scatter in such measurements is inevitable, how many measurements are made and how stable the measurement during all the measuring procedure should be clearly presented.

Regarding the adhesion of 2D homo/heterostructures, for each 2D crystal, we considered 15 thermally-connected crystal flakes on each of which 10 individual adhesion measurements were taken from different locations of the flake surface at each annealing temperature. Our reported experimental errors for each data measurement confirm the reproducibility of our measurements. Regarding the adhesion of G/SiO_x under different contact pressures, a 2 μ m \times 2 μ m smooth region of SiO_x substrate was first located and then, 10 contacts with 100 nm interval spacing were formed at each pressure load under a very clean environment.

We provided the aforementioned information along with more details of our experiments in Methods (Adhesion of G/SiO_x under different contact pressures, Page 16) and Supplementary Information (Section S2. Interfacial adhesion energy (IAE) measurements, Page 6) as follows:

Adhesion of G/SiO_x under different contact pressures

We conducted a series of interfacial adhesion measurements over a pressure range of 0-10 MPa at the interface of G crystal tip/pre-annealed SiO_x substrate and G crystal tip/pre-annealed G substrate. To further minimize experimental uncertainty, a 2 μ m \times 2 μ m smooth region of the SiO_x (G) substrate with an RMS surface roughness of 0.305nm (0.077nm) was first located by non-contact AFM roughness measurements and then, ten contacts with 100 nm interval spacing were formed at each pressure load under a very clean environment, allowing us to perform all measurements within a very small region in close proximity to the microheater (see the SEM image in the inset of bottom panel in Fig. 3b). Moreover, prior to each pressure increment, SiO_x and G substrates are annealed at 300°C for 30 min to remove any possible adsorbed contaminations and then the G crystal tip/pre-annealed substrate interface is further annealed at 300°C for 15 min, followed by the new round of adhesion measurements.

Section S2. Interfacial adhesion energy (IAE) measurements

All retraction $F-d$ curves between 2D crystal tips and untreated/precooling-treated substrates were obtained under controlled ambient conditions in the near-equilibrium regime. For each 2D crystal substrate, we considered 15 thermally-connected crystal flakes on each of which 10 individual adhesion measurements at a contact pressure of 5 MPa (unless otherwise noted) were taken from different locations of the flake surface at each annealing temperature to confirm the reproducibility. The contact time (dwell time) of 2D crystal tips with the substrate was 15 min to reach thermal equilibrium at the contact interface. The approach speed was set to be 10 nm/s while a very slow pulling rate of 1 nm/s was used so that the tip remains in thermodynamic equilibrium with the substrate upon tip retraction. Such a slow pulling rate was achieved by using a 16-bit digital-to-analog converter in low voltage mode with an ultralow noise AFM controller which significantly improved the Z scanner's vertical resolution to 0.1 Å

at the expense of limiting the Z scanner's motion range. Very careful adjustment of the Z servo gain to suppress any possible oscillation of the Z scanner combined with an ultralow noise floor ($<0.3 \text{ \AA}$) in our setup could further make the retraction measurements in the near-equilibrium regime possible. In order to measure the cohesion energy, during the attachment of nanomesas to the glue-coated tip, $F-d$ curves were recorded as the tip was gently pulled away from the substrate surface in a direction perpendicular (parallel) to the single basal plane of 2D crystal, leading to pulling off (shearing) the upper section of the nanomesa (attached to the tip apex) from the lower section (fixed to the 2D crystal substrate). The annealing temperature for the case of cohesion measurements (studied after completion of our interfacial adhesion experiments) was controlled by a Kapton heater while the probe was fully retracted ($\sim 4 \text{ cm}$).

The long introduction is nearly no relevant to the topic of the work. Literatures of related theoretical and experimental investigation on the interfacial interaction should be introduced and carefully discussed, and previous results should be compared.

Following the respected reviewer's comment, we modified Introduction (Page 2, Paragraphs 1-3) to make it shorter and also to better reflect experimental and theoretical methods on the interfacial interaction of 2D crystals through adding seven more references [3-9]. We have also performed a very comprehensive comparison study of the interfacial adhesion energy of 2D homo/heterostructures and 2D crystal/SiO_x in the main text (Page 4, Paragraph 4; Page 7, Paragraph 2; Page 11, Paragraph 1; and Page 12, Paragraph 1) and Supplementary Information (Section S5. Comparative studies of IAE) and cited all relevant papers in the field (over 40 references, please see, for instance, Tables S1, S3, S4, S7 in the following) along with their detailed discussions. We also prepare Introduction in such a way that the significance of the accurate quantification of interfacial adhesion of 2D crystals could be highlighted in many different practical applications. This makes the present work accessible to a broader audience beyond the specialists in our field, and, as a result, can excite the immediate interest of the very broad readership of Nature Communications.

- [3] W. Wang, S. Dai, X. Li, J. Yang, D. J. Srolovitz and Q. Zheng, "Measurement of the cleavage energy of graphite," *Nature Communications*, vol. 6, p. 7853, 2015.
- [4] C. C. Vu, S. Zhang, M. Urbakh, Q. Li, Q. C. He and Q. Zheng, "Observation of normal-force-independent superlubricity in mesoscopic graphite contacts," *Physical Review B*, vol. 94, p. 081405, 2016.
- [5] E. Koren, E. Lörtscher, C. Rawlings, A. W. Knoll and U. Duerig, "Adhesion and friction in mesoscopic graphite contacts," *Science*, vol. 348, no. 6235, pp. 679-683, 2015.
- [6] D. M. Tang, D. G. Kvashnin, S. Najmaei, Y. Bando, K. Kimoto, P. Koskinen, P. M. Ajayan, B. I. Yakobson, P. B. Sorokin, J. Lou and D. Golberg, "Nanomechanical cleavage of molybdenum disulphide atomic layers," *Nature Communications*, vol. 5, p. 3631, 2014.
- [7] B. Li, J. Yin, X. Liu, H. Wu, J. Li, X. Li and W. Guo, "Probing van der Waals interactions at two-dimensional heterointerfaces," *Nature Nanotechnology*, vol. 10.10138, pp. s41565-019-0405-2, 2019.
- [8] J. D. Wood, C. M. Harvey and S. Wang, "Adhesion toughness of multilayer graphene films," *Nature Communications*, vol. 8, p. 1952, 2017.
- [9] S. P. Koenig, N. G. Boddeti, M. L. Dunn and J. S. Bunch, "Ultrastrong adhesion of graphene membranes," *Nature Nanotechnology*, vol. 6, pp. 543-546, 2011.

The revisions and related texts are listed below

Main text, Page 2, Paragraphs 1-3

Newly emerged vdW heterostructures – stacks of individual monolayer flakes of different 2DLMs assembled layer by layer – offer a variety of new physical properties, thanks to the full spectrum of electronic properties in 2DLMs, from conducting graphene, to semiconducting TMDs, to insulating hBN. An essential feature of such heterostructures is atomically clean interfaces to achieve the best device performance – any interfacial contamination (e.g., blisters) results in deterioration of transport properties [5]. As such, wet transfer and dry pick-and-lift transfer techniques are widely used for assembly of vdW heterostructures. However, both direct mechanical assembly techniques rely strongly on vdW interactions between the 2D crystals, and, as a result, an accurate quantification of interfacial adhesion between different 2DLMs is crucial for the mass production of blister-free vdW heterostructures.

Fascinating interlayer vdW-dependent properties of similar and dissimilar 2DLMs provide a unique opportunity to study the nature of electronic structure and band alignment, interfacial thermal and electrical resistance, ion intercalation and deintercalation process, interfacial nanofluidic transport and drug delivery behavior, photon absorption and photocurrent/photovoltaic production, interfacial charge polarization and redistribution, spin-orbit coupling and many others in layered materials-based devices [6, 7, 8, 9, 10]. Notably, interfacial electrical, mechanical, optoelectronic, magnetic and thermal properties of layered materials can also interact in a rather complex way. For instance, formation of any delamination-motivated surface corrugations in 2DLMs can give rise to local strain distribution and curvature-induced rehybridization, which modify the electronic structure and local charge distribution, create polarized carrier puddles and dipole moment, induce pseudomagnetic fields and thus alter magnetic, optical and electrical properties as well as chemical surface reactivity [11]. Moreover, the vdW interaction as a key medium for the stress transfer both within and across the interface of 2DLMs can highly impact their thermal and electrical properties in such a way that a 2D layered system can act as a heat conductor or insulator and/or a semimetal or electrical insulator through strain engineering [12, 13, 14].

The interfacial physical and chemical behavior of layered materials becomes even more complicated when we consider that airborne contaminants are an inevitable part of any vdW heterostructures and therefore addressing quantitatively to what degree their interfacial adhesion energy (IAE) is influenced by interfacial contaminants and nanoblisters and how to effectively remove them is of fundamental and technological importance for the continued development of such promising materials. Despite the significance of such a fundamental property for any layered materials, there have been relatively limited experimental and theoretical methods with significant diversity in the reported IAE values for 2DLMs in general and graphite (G) crystals specifically, where the exact cause of the variation in their IAE values has also remained to be elucidated by a comprehensive and accurate experimental technique [15, 16, 17, 18, 19, 20, 21].

Main text, Page 4, Paragraph 4

It is also worth making a comparison between our IAE results and those in the literature. Although many attempts have been made over the last six decades to measure the IAE of G crystal with the reported values ranging from 0.15–0.72 Jm⁻² (see Table S1), there are few direct measurements of intrinsic IAE values available for comparison. From the literature data, we found that our measurements for cohesion energy of G crystal are in excellent agreement with micro-force sensing probe measurements on 4 μm wide square mesas (0.37±0.01 Jm⁻² [15]) and AFM-assisted shearing measurements on 3 μm wide square mesas (0.35 Jm⁻² [16]), but inconsistent with the AFM-assisted shearing measurements on circular mesas of 100–600 nm in diameter (0.227±0.005 Jm⁻² [17]). We revisited the lateral stiffness

calibration of all probes used in ref. [17] by means of a 3D finite element simulation, predicting consistently stiffer (~ 1.6 times) probes than those described in the original work (Fig. S11). Using this modified lateral spring constant yields an IAE value of $0.362 \pm 0.008 \text{ Jm}^{-2}$, more consistent with our measurements. We also note that to the best of our knowledge no IAE measurement on the hBN homointerface yet exists, while there is only one measured IAE value of 0.22 Jm^{-2} at the MoS_2 homointerface using a nanomechanical cleavage technique [18], which is lower than half of our values. We believe that in their IAE calculations, a very low bending stiffness value of 0.92 eV was used for the monolayer MoS_2 which is even lower than that of monolayer G (1.49 eV) and monolayer hBN (1.34 eV) whose thicknesses are almost half of MoS_2 thickness, resulting in such a low IAE value in their MoS_2 homostructure (Section S7).

Main text, Page 7, Paragraph 2

During the revision of this paper, Li et al. reported the vdW interactions of G/hBN and G/ MoS_2 heterostructures, where a Si AFM tip wrapped with a thin G flake is brought into contact with pre-annealed G, hBN and MoS_2 substrates in high vacuum at room temperature [19]. Using a G-wrapped sharp tip with an unknown contact area only allows the measurements of the critical adhesion forces (i.e., pull-off forces, P) between G/G, G/hBN and G/ MoS_2 . Qualitatively speaking, their measurements showed that G experiences a weaker vdW interaction with hBN and G than MoS_2 , yielding a critical adhesion force ratio of $P_{\text{G/MoS}_2}/P_{\text{G/hBN}} = 1.079$ and $P_{\text{G/MoS}_2}/P_{\text{G/G}} = 1.028$. Similarly, our IAE ratios of $\Gamma_{\text{G/MoS}_2}/\Gamma_{\text{G/hBN}}$ and $\Gamma_{\text{G/MoS}_2}/\Gamma_{\text{G/G}}$ for the roughly similar experimental conditions (i.e., precooling-treated heterointerfaces annealed at 130°C) are 1.088 and 1.059, respectively, which are qualitatively in good agreement with their findings.

Main text, Page 11, Paragraph 1

Such intimate and strong interaction of G/ SiO_x suggests that the electron scattering sites across the interface as well as the convex sites of corrugated G result in the formation of short-range chemical bonds which act as anchoring spots to locally pin G to the SiO_x surface at the location of such chemically active sites [32, 33, 34, 35]. Since monolayer G with extreme flexibility possesses more chemically active sites than multilayer G at the G/ SiO_x interface, stronger adhesion energy of monolayer to SiO_x is expected, as previously confirmed by a pressurized blister test to be $0.45 \pm 0.02 \text{ Jm}^{-2}$ for monolayer G but $0.31 \pm 0.03 \text{ Jm}^{-2}$ for multilayer G [21].

Main text, Page 12, Paragraph 1

Our further analysis on the temporal evolution of the adhesion energy (see Section S6 for our detailed analysis) and contamination thickness measured on the mechanically exfoliated G surface during the first 60 min of air exposure reveals that its intrinsic IAE of $0.341 \pm 0.025 \text{ Jm}^{-2}$ obtained under ultrahigh vacuum or ultrahigh-purity argon atmosphere is well consistent with our experimental value of $0.328 \pm 0.028 \text{ Jm}^{-2}$ but drastically decreases within the first minute of air exposure and eventually approaches a saturated value of $0.15 \pm 0.02 \text{ Jm}^{-2}$ after 10 min (Fig. 5), which is smaller than our IAE value of $0.233 \pm 0.035 \text{ Jm}^{-2}$ upon room-temperature storage for 1 h. This could be attributed to the presence of the contact pressure and the vdW interaction between the layers in our experiments which may still play a role to squeeze away the contaminants even at room temperature, leaving cleaner interfaces with stronger interactions. We also note that a substantial decrease in the surface hydrocarbon level under vacuum, high-temperature ($500\text{--}1000^\circ\text{C}$) treatment during the WCA measurements results in the IAE recovery of the G crystal

($0.282 \pm 0.024 \text{ Jm}^{-2}$), which is in good agreement with our IAE value of $0.268 \pm 0.028 \text{ Jm}^{-2}$ at much lower temperature (300°C), further confirming the dominant contribution of the vdW force to the IAE improvement.

Section S5. Comparative studies of IAE

In this section, we perform a comprehensive comparison study on the interfacial adhesion energy (IAE) of 2D crystals and 2D crystal/ SiO_x heterostructures obtained from a wide range of experimental methods. While a vast majority of studies have been conducted on the interaction of G with G (Section S5.1) and SiO_x (Section S5.2) substrates with a wide range of reported IAE values, to the best of our knowledge, no IAE measurement at the hBN/hBN and hBN/ SiO_x interfaces yet exists, and also there are a very limited number of reports on the interaction of MoS_2 with MoS_2 (Section S5.1) and SiO_x (Section S5.2) substrates. We also note that, to the best of our knowledge, there is no direct IAE measurement on the 2D crystal heterostructures.

Section S5.1. Comparison study on cohesion energy of 2D crystal homostructures

Although many attempts have been made over the last six decades to measure the cohesion energy of G crystal with the reported values ranging from $0.15\text{--}0.72 \text{ Jm}^{-2}$ (Table S1), there are few direct measurements of cohesion energy available for comparison. From the literature data, we found that our measurements for cohesion energy of G crystal are in excellent agreement with micro-force sensing probe measurements on $4 \mu\text{m}$ wide square mesas ($0.37 \pm 0.01 \text{ Jm}^{-2}$ [13]) and AFM-assisted shearing measurements on $3 \mu\text{m}$ wide square mesas (0.35 Jm^{-2} [14]), but inconsistent with the recent AFM-assisted shearing measurements on circular mesas of $100\text{--}600 \text{ nm}$ in diameter ($0.227 \pm 0.005 \text{ Jm}^{-2}$ [15]).

Table S1. Cohesion energy of carbon nanotubes, few-layer graphene, and graphite.

Method	Sample	Stack	$\Gamma \text{ (J/m}^2\text{)}$	Ref
Heat of wetting	Graphite	N.A.	0.26 ± 0.03	[21]
Radial deformation of MWCNT	Collapsed MWCNT	(Non-)AB*	$0.15\text{--}0.31$	[22]
Thermal desorption	HOPG	AB	0.37 ± 0.03	[23]
MWCNT retraction	MWCNT	Non-AB	$0.28\text{--}0.4$	[24]
Deformation of thin sheets	HOPG	AB	0.19 ± 0.01	[25]
AFM pull-off force measurements	HOPG	Non-AB	0.319 ± 0.05	[26]
DWCNT inner-shell pull-out	DWCNT	Non-AB	0.436 ± 0.074	[27]
SEM peeling of MWCNT	Collapsed MWCNT on 1-LG Flattened MWCNT on 1-LG	(Non-)AB	0.40 ± 0.18 0.72 ± 0.32	[28]
AFM-assisted mechanical shearing	HOPG	Non-AB	0.227 ± 0.005	[15]
Self-retraction motion	HOPG	AB Non-AB	0.39 ± 0.02 0.37 ± 0.01	[13]
AFM-assisted mechanical shearing	HOPG	Non-AB	0.35	[14]
AFM nano-indentation	BLG/FLG onto FLG	Non-AB	0.307 ± 0.041	[29]
Atomic intercalation of neon ion	1LG onto HOPG	Non-AB	0.221 ± 0.095	[30]
Surface force balance	CVD-grown 1LG/1LG CVD-grown FLG/FLG	Non-AB Non-AB	0.230 ± 0.008 0.238 ± 0.006	[31]

Section S5.2. Comparison study on IAE of 2D crystal/ SiO_x

Despite many experimental studies devoted to the IAE determination of 2D crystals/ SiO_x heterostructures, no experimental data are available on the interaction of hBN/ SiO_x , whereas the reported IAE data on the interaction of G and MoS_2 with SiO_x are very diverse, ranging from $0.09\text{--}0.90 \text{ Jm}^{-2}$ at the G/ SiO_x interface (Table S3) and $0.08\text{--}0.48 \text{ Jm}^{-2}$ at the MoS_2 / SiO_x interface (Table S4). We believe that a

part of this large data scattering can be attributed to different surface properties of SiO_x during sample preparation, leading to different surface roughness, surface configurations (due to its amorphous nature), surface polarities, charge impurities, surface reactions with ambient humidity, and type of surface termination/defects (i.e., H-, Si- and O-terminated surfaces).

Table S3. Interlayer adhesion energy of carbon nanotubes, few-layer graphene, and graphite on SiO_x.

Method	Sample	Γ (J/m ²)	Ref
AFM nano-indentation	BLG/FLG	0.270±0.020	[29]
Pressurized blister	1LG	0.45±0.02	[32]
	2-5LG	0.31±0.03	
Pressurized blister	1LG	0.24	[33]
Pressurized blister	1LG	0.140±0.040	[34]
	5LG	0.160±0.060	
AFM with a microsphere tip	1LG	0.46±0.02	[35]
Intercalation of nanoparticles	5LG	0.302±0.056*	[36]
Infrared crack opening Interferometry	1LG	0.357±0.016	[37]
Nanoparticle-loaded blister	1LG	0.453±0.006	[38]
	3-5LG	0.317±0.003	
	10-15LG	0.276±0.002	
Intercalation of nanoparticles	FLG	0.567	[39]
Colorimetry technique	2LG	0.9	[40]
Interfacial nanoblister	1LG	0.093±0.001	[41]

Table S4. Interlayer adhesion energy of MoS₂ on SiO_x.

Method	Sample	Γ (J/m ²)	Ref
Intercalation of nanoparticles	FL	0.482	[39]
Pressurized blister	1L	0.212±0.037	[42]
	2L	0.166±0.004	
	3L	0.237±0.016	
	1L CVD	0.236±0.021	
Wrinkle	FL	0.170±0.033	[43]
Interfacial nanoblister	1L	0.082±0.001	[41]

Table S7. Summary of water contact angle measurements and corresponding IAE values of G crystal.

Notes	Measured within	WCA	$2\gamma_G$ (J/m ²)	Ref
Ultrahigh vacuum	3 sec	42±7°	0.348±0.033	[44]
Ambient air at 24°C/48% RH	10 sec	64.4°	0.232	[45]
	2 days	91.0°	0.093*	
550°C annealing in Ar	1 min	54.1°	0.286	
Ambient air at 22-25°C/20-40% RH	10 sec	64.4±2.9°	0.232±0.015	[46]
	7 days	97.0±1.8°	0.072±0.01*	
Ultrahigh vacuum for 15 h	N.A.	59°	0.260	
Ambient air at 22-25°C/20-40% RH	10 sec	68.6±7.1°	0.210±0.034	[47]
Ambient air at 22°C/50% RH	10 sec	68.2±2°	0.212±0.010	[48]
	1 day	90±0.1°	0.096±0.002*	
Ambient air at RT/40-50% RH	1.5 min	62.9±2.2°	0.239±0.012	[49]
Ambient air at RT/40-50% RH	5 min	61.8±3.3°	0.249±0.017	
	1 day	81.9±2.9°	0.129±0.012*	
600°C annealing in He	N.A.	51.4±2.0°	0.300±0.010	
Clean room at 21°C/40% RH	5 sec	53±5°	0.292±0.027	[50]
	8 min	66±3°	0.223±0.016	
	2 days	86±4°	0.112±0.016*	
Water vapor atmosphere	N.A.	58±2°	0.266±0.010	
Ultrahigh-purity argon atmosphere	1 min	45±3°	0.333±0.016	
Evacuation/1050°C annealing/vacuum	1 min	55±1°	0.281±0.005	
Evacuation/1000°C annealing/atmosphere	1 min	73±5°	0.187±0.025	
Ambient air at 22-25°C/20-40% RH	10 sec	60±0.1°	0.255±0.002	[51]

Minor technique issues: 1) A relatively gradual reduction of the interfacial adhesion force is observed at homointerfaces (in manuscript, page 3, paragraph 2, line 10). It is stated that the short-range vdW interaction is the major separation mechanism in 2DLMs (in Supplementary Information, page 17, line 13). But in Fig. 1b, the distance of gradual reduction is ~10nm, which is larger than general feature distance of short-range vdW interaction. Can the authors respond to that inconsistency?

In Fig. 1b, ~10 nm is the piezo displacement rather than the interlayer distance between two adjacent 2D crystal layers and thus does not represent the distance of short-range vdW interaction.

To make it clear, we modified the main text (Page 3, Paragraph 2) and Supplementary Information (Section S2.1, Page 6) as follows:

Main text, Page 3, Paragraph 2:

However, our $F-d$ curves upon tip approach display a small jump-to-contact force due to the hydrophobic nature of 2D crystal nanomesas (see, for instance, hBN/hBN approach curve in Fig. 1b), suggesting negligible effect of tip-sample capillary forces and bridging bubble ruptures on the retraction curves. For comparison purposes, we recorded rupture force curves of bridging nanobubbles formed between the *in situ* flattened Si tip and the SiO_x/Si substrate at the relative humidity of 70% (red curve in Fig. 1d). By closer inspection of the $F-d$ curves, we notice three fundamental differences in the separation mechanism between 2DLMs (e.g., blue curve in Fig. 1d) and bridging nanobubbles (e.g., red curve in Fig. 1d). First, the separation range in the 2DLMs (typically 5-10 nm) is almost an order of magnitude shorter than that in the nanobubbles (typically 50-80 nm), further supporting the claim that the short-range vdW interaction (rather than the long-range bubble deformation) is the major separation mechanism in 2DLMs. Second, contrary to the case of the bridging nanobubbles where the adhesion strongly depends on the retraction speed of the piezo [22], our $F-d$ analysis under various tip retraction rates in the range of 1-1000nm/s reveals no appreciable effect on the IAE of 2DLMs, indicative of the absence of any dynamic (viscous) forces in the separation mechanism of 2DLMs. Third, an abrupt drop in the retraction force curves of nanobubbles just prior to the complete separation could reflect the pinch-off process of the unstable bubble neck, whereas a relatively fast transition from a surface contact to a line contact during the separation process in 2DLMs can eventually lead to the sudden break of the line contact and thereby an abrupt force drop at the very end of the separation process.

Supplementary Information, Section S2.1, Page 6:

It is also worth pointing out that the reported distance between the initiation of the separation and full separation of the tip-sample in Fig. 1b is the piezo displacement (e.g., ~10 nm at hBN/hBN, ~9 nm at G/G and ~5 nm at MoS₂/ MoS₂ interfaces) rather than the interlayer distance between two adjacent 2D crystal layers and thus does not represent the distance of short-range vdW interaction at the tip-sample interface.

2) Can the authors explain the reason that why only the graphite was etched to circle shape compared with BN and MoS₂.

We designed our mask in such a way that arrays of G, hBN and MoS₂ nanomesas were fabricated with both square and circular cross sections and we used both shapes in our measurements. For instance, square and circular MoS₂ nanomesas were used in Fig. 1b and Fig. 2b, respectively.

To make it clear, we modified Supplementary Information (Section S1.2, Page 3) as follows:

Section S1.2. Fabrication of nano-sized 2D crystal mesas

A ~100-nm-thick bilayer of polymethyl methacrylate (PMMA) 495K (60 nm)/950K (40 nm) is spin coated onto the freshly cleaved surface of 1-mm-thick HOPG (SPI, Grade 1, with a mosaic spread value of 0.4°), hBN (grade A, with single crystal domains over 100 μ m) and MoS₂ (429MS-AB, natural single crystals from Canada) substrates, baked each layer for 10 min at 120 °C to evaporate the solvent and then patterned by electron beam lithography. The mask was designed in such a way that arrays of G, hBN and MoS₂ nanomesas were fabricated with both square and circular cross sections and both shapes were used in interfacial adhesion measurements. After developing the exposed PMMA area in 1:3 MIBK/NMP, a 10-nm-thick aluminum film is deposited by thermal evaporation, followed by lift-off process in acetone. The unprotected HOPG, hBN and MoS₂ areas are thinned down by using a reactive ion etching system with pure O₂ (precursor flow rate: 10 sccm, RF power: 40W, pressure: 10 mTorr), CHF₃/Ar/O₂ (10/5/2 sccm, 30W, 10 mTorr) and SF₆ (20 sccm, 100W, 20 mTorr) reactive gases, respectively. Square (circular) mesas with a width (diameter) of 55-65 nm and etch depth of 50-100 nm emerge from 2D crystal substrates during the plasma etch.

3) In Fig. 1c, the shear force of MoS₂ keeps stable and G goes down gradually. What is the mechanism?

Applying a lateral shear force to a square mesa of width w or a circular mesa of diameter D leads to the lateral displacement x of the upper section relative to the bottom section of the mesa and creation of new interface area $A(x)$. At the sliding interface, the total free energy may change by $U(x) = -\Gamma A(x)$. We next obtained the corresponding interfacial adhesion force opposing new surface formation as $F_a(x) = -dU(x)/dx = \Gamma dA(x)/dx$. For the circular/square mesa with the following new interface area

$$\text{Circular mesa: } A(x) = \frac{D^2}{2} \left[\cos^{-1} \left(\frac{x}{D} \right) - \frac{x}{D} \sqrt{1 - \left(\frac{x}{D} \right)^2} \right]$$

$$\text{Square mesa: } A(x) = w(w - x)$$

the corresponding interfacial adhesion forces can be written as

$$\text{Circular mesa: } F_a(x) = -\Gamma D \sqrt{1 - \left(\frac{x}{D} \right)^2}$$

$$\text{Square mesa: } F_a = -\Gamma w$$

As the respected reviewer can see, the interfacial adhesion force of the square MoS₂ mesa is constant while the interfacial adhesion energy of the circular G mesa is a function of x and decreases as x increases. For the interested reader, we added the aforementioned discussion to Method (Force-displacement measurements, Page 15) and Supplementary Information (Section S2.1, Page 7) as follows:

Force-displacement measurements

All retraction $F-d$ curves between the 2D crystal tips and the untreated/precooling-treated substrates were obtained under controlled ambient conditions in the near-equilibrium regime which was achieved by an ultralow noise floor of less than 0.3 Å, an ultralow noise AFM controller with the Z scanner's vertical resolution of better than 0.1 Å and also using a very slow (quasi-static) pulling rate of 1 nm/s. Very careful adjustment of the Z servo gain to suppress any possible oscillation of the Z scanner could further make the retraction measurements in the near-equilibrium regime possible (Section S2). In order to calculate the IAE per unit area (Γ , J/m²) from the recorded retraction-displacement curves, we integrate the retraction force as a function of the piezo displacement (light blue-shaded area in Figs. 1b and 1d), followed by dividing the resulting adhesion energy by the known contact area at the interface. However, in order to extract the IAE from the shear force-displacement curves, the interfacial adhesion force opposing new surface formation is first obtained as $F_a(x) = \Gamma[dA(x)/dx]$, where x represents the lateral displacement of the mobile section of the mesa with respect to the initial position, and $A(x)$ is the overlap area of the top and bottom sections of the mesa as a function of x . For a square mesa of width w and a circular mesa of diameter D , the maximum shear force in the shear $F-d$ curves which is required to initiate sliding (i.e., F_a at $x = 0$) can be related to the interfacial adhesion energy by Γw and ΓD , respectively (Section S2.1).

Section S2.1, Page 7

In order to extract the interfacial adhesion energy from the shear force-displacement curves, we first assumed a lateral shear force F_s being applied to a square mesa of width w or a circular mesa of diameter D , leading to the lateral displacement x of the upper section relative to the bottom section of the mesa and creation of new interface area $A(x)$. At the sliding interface, the total free energy may change by $U(x) = -\Gamma A(x)$. We next obtained the corresponding interfacial adhesion force opposing new surface formation as $F_a(x) = -dU(x)/dx = \Gamma dA(x)/dx$. For the circular/square mesa with the following new interface area

Circular mesa:
$$A(x) = \frac{D^2}{2} \left[\cos^{-1} \left(\frac{x}{D} \right) - \frac{x}{D} \sqrt{1 - \left(\frac{x}{D} \right)^2} \right]$$

Square mesa:
$$A(x) = w(w - x)$$

the corresponding interfacial adhesion forces can be written as

Circular mesa:
$$F_a(x) = -\Gamma D \sqrt{1 - \left(\frac{x}{D} \right)^2}$$

Square mesa:
$$F_a = -\Gamma w$$

By inspection, one can see that the maximum interfacial adhesion force required to initiate sliding the circular and square mesas can simply be given by ΓD and Γw , respectively.

4) Among the measurements of 2D crystal heterointerfaces in Fig. 2b, the tip-nanomesa seems have already stayed in ambient air for a long time and the tip-nanomesa may become so-called untreated(with the definition in manuscript, page 3, line 10) tip-nanomesa. Is there any difference on comparing the results of treated and untreated substrates with the influence of untreated tip-mesa.

As we mentioned in Methods and Supplementary Information, we used a similar method as for the (un)treated substrates to age the tip-attached nanomesas for the subsequent contact with their corresponding aged substrates. However, unlike the fresh 2D crystal substrates obtained by the conventional mechanical exfoliation technique, fresh 2D crystal tips were simply obtained by our previously developed AFM-assisted shear exfoliation technique [10].

[10] H. Rokni and W. Lu, "Nanoscale probing of interaction in atomically thin layered materials," *ACS Central Science*, pp. 288-297, 2018.

To make it clear, we revised Methods (**Surface preparation of 2D crystal flakes, Page 13**) as follows:

Surface preparation of 2D crystal flakes

Instead of immediately removing all 2D crystal-loaded adhesive tapes from the piece#1 to complete the mechanical exfoliation onto the microheaters, we only peeled off the tape containing the 2D crystal flakes of interest (G, hBN or MoS₂) for the interfacial adhesion measurements, thereby enabling much better control over the possible adsorption of airborne contaminants onto the fresh surface of 2D crystals. We prepared aged substrates by two different aging conditions: (1) the freshly exfoliated 2D crystal substrate and the bare SiO_x/Si substrate were directly exposed to the ambient air for 1 h at room temperature; (2) 2D crystal flakes were freshly exfoliated on a precooled (-15°C) SiO_x/Si substrate and the sample was kept at this temperature for 15 min, followed by the air exposure of the exfoliated 2D crystal substrate and the underlying SiO_x/Si substrate for 1 h at room temperature. A similar method was used to age tip-attached nanomesas for the subsequent contact with their corresponding aged substrates where fresh 2D crystal tips were simply obtained by our previously developed AFM-assisted shear exfoliation technique [2] (Section S1).

5) In Fig. 2a, IAE values as a function of annealing temperatures at the intact hBN homointerface, it is evident that the IAE is not independent of the annealing temperature, especially at 300 °C (inconsistent to conclusion in manuscript, page 5, paragraph 2, line 6). Based on that, the conclusion which is stated that nearly full recovery of the intrinsic IAE only at the hBN homointerface upon annealing at 300 °C (in manuscript, page 6, line 2) need to be discussed.

Within our experimental accuracy, we did not observe any systematic change in the IAE values at the intact 2D crystal homointerfaces at least up to 300 °C (gray circles in Fig. 2a) and, therefore, we could not conclude from Fig. 2a that the IAE at the intact homointerfaces is thermal-annealing dependent. In particular, Fig. 2a and Table S5 suggest that the IAE values at the intact hBN/hBN interface are $0.319 \pm 0.022 \text{ Jm}^{-2}$ and $0.312 \pm 0.027 \text{ Jm}^{-2}$ at -15°C and 300 °C, respectively, further indicating no appreciable change (only 2%) in the IAE of intact hBN/hBN upon the thermal annealing. Also, among 2D crystals under consideration, IAE of untreated hBN (red circles in Fig. 2a) approaches very closely to that of intact hBN (gray circles in Fig. 2a) at 300 °C, which is not the case for G and MoS₂ crystals, as evident from Fig. 2a.

To make it clear, we modified the main text as follows:

Page5, Paragraph 2:

It is evident from the gray circles in Fig. 2a (and also Table S5) that, upon the attachment of nanomesas to the AFM tip after the thermal annealing, the measured cohesion energy at the intact homointerfaces is, within our experimental accuracy, independent of the annealing temperatures.

6) In Fig. 2b, at the MoS₂/G interface, the mechanism that the IAE drops when the annealing temperature changes from -15 to 22 °C should be explained.

From Fig. 2a and Fig. 2b, we observed that the IAE of G/G, hBN/hBN and MoS₂/hBN increases from -15 to 22 °C whereas the IAE of MoS₂/MoS₂, G/hBN and MoS₂/G decreases from -15 to 22 °C. Since the IAE does not show a single trend when annealing from -15 to 22 °C, we could not reach a general explanation. For this kind no-trend behavior, it is generally thought to be material specific. In other materials, researchers have also found that adhesion can increase or decrease with temperature, without clear rules. While some studies explain the behavior to be a nature result of complicated interaction of interatomic potentials, such an explanation is not particularly useful since it does not provide any general trend guidance (or in fact, there is no trend). It could be that it is just an aggregate result of complicated interaction among surface structures and interatomic potentials, and there is no simple explanation. We will leave it for future studies. This may also pose an interesting problem for potential readers to explore. Following the reviewer's question, we made sure that we did not claim any monotonic trend of IAE on temperature.

7) In Fig. 3a, what's the proportion of the existence of point of bond at edge among all the measured curves under 5 MPa? And I think in the process of the fabrication of 2D crystal nanomesa, the etching procedure would functionalize or chemically modify the edge of the nanomesa, and the edge would become more active than pure edge. The phenomena of bond at edge may be from that functionalized edge. (in manuscript, page 10, line 2).

Almost 5% of our *F-d* measurements at the G/SiO_x interface exhibit the possible short-range bond at the edge of the G nanomesa. As the respected reviewer correctly mentioned, although 2D crystals possess intrinsic active edge sites, the etching process could also functionalize or chemically modify their edge. However, during the attachment of G, hBN and MoS₂ nanomesas to the AFM tip, we observed a mild detachment through the *F-d* curves, confirming no chemical bond/dangling bond both at the edge and along the sliding plane of nanomesas. As such, we believe that at least the edge of the most bottom layer of the tip-attached nanomesas is unlikely to be functionalized due to the etching process. In particular, we did not observe the chemical bond at the edge of tip-attached hBN and MoS₂ crystal in contact with SiO_x, further confirming the negligible effect of the possible edge functionalization on the IAE of 2D crystal/SiO_x heterostructures. For the interested reader and to be clearer, we added one section to Supplementary Information about the above discussion (Section S2.6. Possible edge functionalization of 2D crystals, Page 11) as follows:

Section S2.6. Possible edge functionalization of 2D crystals

Almost 5% of our *F-d* measurements at the G/SiO_x interface exhibit the possible short-range bond at the edge of the G nanomesa. Although 2D crystals possess intrinsic active edge sites, the etching process of nanomesas could also functionalize or chemically modify their edge. However, during the attachment of G, hBN and MoS₂ nanomesas to the AFM tip, we observed a mild detachment through the *F-d* curves,

confirming no chemical bond/dangling bond both at the edge and along the sliding plane of nanomesas. As such, we believe that at least the edge of the most bottom layer of the tip-attached nanomesas is unlikely to be functionalized due to the etching process. In particular, we did not observe the chemical bond at the edge of tip-attached hBN and MoS₂ crystals in contact with SiO_x, further confirming the negligible effect of the possible edge functionalization on the IAE of 2D crystal/SiO_x heterostructures.

8) Some writing errors need to be revised. E.g. adhesive tape (Methods, page 12, line 9 in manuscript and in Supplementary Information, page 2, paragraph 2, line 3).

We thank the respected reviewer for pointing this out. We have fixed the typos and also double-checked the whole manuscript for any other possible typos.

Reviewer #3 (Remarks to the Author):

In this manuscript, the authors performed comprehensive quantitative measurements on the interfacial adhesion energy (IAE) of two-dimensional (2D) layered materials, van der Waals heterostructures and 2D materials on SiO₂ substrates. Their results showed that the MoS₂ has the maximum IAE, compared to graphene and h-BN, independent of the annealing temperatures. They also quantified the effect of airborne contaminants and humidity on the interfacial adhesion level and revealed that the thermal annealing can sufficiently affect the IAE at both the contaminated homo- and heterointerfaces. Regarding the IAE on SiO₂ substrates, they measured the highest value for graphene and the lowest one for the MoS₂, attributed to their different degrees of conformation to the SiO₂ and the formation of short-range chemical bonds in the G/SiO_x. These results, if reliable, would be highly appealing to the community of 2D materials, from both theoretical and experimental points of view. However, since the measurement on vdW-like force of 2D materials is very tricky, the authors need to further confirm their results and strengthen their conclusions by performing further characterizations. The authors should adequately address my following issues before I can make any recommendation.

We thank the respected reviewer for his/her valuable inputs and generous comments on the manuscript. In the following, we provide a point-by-point response to the reviewer's comments.

1. The authors glued multilayer 2D materials of different shapes to the AFM tips and then measured the interlayer interaction force between the tip-attached 2D materials nanomesa and the underlying samples. I have a concern that the interlayer interaction within the nanomesa is also dominated by vdW interaction, which may compete with the vdW-like tip-sample interaction and lead to big errors and even wrong results. How can the authors make sure that the force is measured just at the tip-sample interface, not within the nanomesa.

We first measured the intrinsic cohesion energy of 2D crystals (Fig. 1e and gray circles in Fig. 2a), confirming that the interfacial cohesion energy across the 2D crystal nanomesa is larger than the interfacial adhesion energy at all 2D crystal tip-sample interfaces. That is the separation most likely takes place at the tip-sample interface rather than somewhere across the thickness of the tip-attached nanomesa. Moreover, for each tip-attached 2D crystal, we formed all contacts with 1 μ m interval spacing within the same distance from the heating line, allowing us to easily locate and scan all contact spots (using the non-contact AFM mode) for any possible exfoliation of monolayer or few layers from tip-attached 2D crystal onto the sample. For the spot with exfoliated mono/few-layer 2D crystal, the area under the corresponding $F-d$ curve was considered as the intrinsic cohesion energy rather than the interfacial adhesion energy at the tip-sample interface.

To make it clear, we added the above discussion to Supplementary Information (Section S2. Interfacial adhesion energy (IAE) measurements, Page 6) as follows:

Section S2, Page 6, Paragraph 2:

In order to make sure that the $F-d$ curves are measured at the tip-sample interface not within the nanomesa, we first measured the intrinsic cohesion energy of 2D crystals (Fig. 1e and gray circles in Fig. 2a), confirming that the cohesion energy across the 2D crystal nanomesa is larger than the interfacial

adhesion energy at all 2D crystal tip-sample interfaces. We also observed larger pull-off forces at the intact interfaces compared to contaminated interfaces, well consistent with our reported IAE values. Therefore, the separation most likely takes place at the tip-sample interface rather than somewhere across the thickness of the tip-attached nanomesa. Moreover, for each tip-attached 2D crystal, we formed all contacts with 1 μm interval spacing within the same distance from the heating line, allowing us to easily locate and scan all contact spots (using the non-contact AFM mode) for any possible exfoliation of monolayer or few layers from tip-attached 2D crystal onto the sample. For the spot with exfoliated mono/few-layer 2D crystal, the area under the corresponding $F-d$ curve was considered as the intrinsic cohesion energy rather than the interfacial adhesion energy at the tip-sample interface.

2. Previous measurements reported a stronger van der Waals interaction at the graphite-BN interface than that at the graphite-MoS₂ interface, while the authors' results do not follow this trend clearly. They should explain why and make detailed comparison to previous results. The interlayer interaction force of 2D materials has been measured by many groups, at least for graphene.

It would be great if the respected reviewer could kindly share papers(s) in which the vdW interaction of G/hBN has experimentally been shown to be stronger than that of G/MoS₂. However, very recently, Li et al. [1] reported the vdW interactions of G/hBN and G/MoS₂ heterostructures where a Si AFM tip wrapped with a thin G flake is brought into contact with pre-annealed G, hBN and MoS₂ substrates in high vacuum at room temperature. Using a G-wrapped sharp tip with an unknown contact area only allowed them to measure the critical adhesion forces (i.e., pull-off forces, P) between G/G, G/hBN and G/MoS₂. Qualitatively speaking, their measurements showed that G experiences a weaker vdW interaction with hBN and G than MoS₂, yielding a critical adhesion force ratio of $P_{G/MoS_2}/P_{G/hBN} = 1.079$ and $P_{G/MoS_2}/P_{G/G} = 1.028$. Similarly, our IAE ratios of $\Gamma_{G/MoS_2}/\Gamma_{G/hBN}$ and $\Gamma_{G/MoS_2}/\Gamma_{G/G}$ for the roughly similar experimental conditions (i.e., precooling-treated heterointerfaces annealed at 130°C) are 1.088 and 1.059, respectively, which are qualitatively in good agreement with their findings.

[1] B. Li, J. Yin, X. Liu, H. Wu, J. Li, X. Li and W. Guo, "Probing van der Waals interactions at two-dimensional heterointerfaces" Nature Nanotechnology, vol. 14, pp. 567-572, 2019.

Regarding a detailed comparison study, we performed a very comprehensive comparison study of the interfacial adhesion energy of 2D homo/heterostructures and 2D crystal/SiO_x in Supplementary Information (Section S5. Comparative studies of IAE) and cited all relevant papers in the field (over 40 references, please see, for instance, Tables S1, S3, S4, S7 in the following) along with detailed discussions. Following the respected reviewer's comment, we revised the main text in order to discuss in detail those comparative results that are essential for the general understanding of the present results (Page 4, Paragraph 4; Page 7, Paragraph 2; Page 11, Paragraph 1; and Page 12, Paragraph 1).

Main text, Page 4, Paragraph 4

It is also worth making a comparison between our IAE results and those in the literature. Although many attempts have been made over the last six decades to measure the IAE of G crystal with the reported values ranging from 0.15–0.72 Jm⁻² (see Table S1), there are few direct measurements of intrinsic IAE values available for comparison. From the literature data, we found that our measurements for cohesion energy of G crystal are in excellent agreement with micro-force sensing probe measurements

on 4 μm wide square mesas ($0.37\pm 0.01 \text{ Jm}^{-2}$ [15]) and AFM-assisted shearing measurements on 3 μm wide square mesas (0.35 Jm^{-2} [16]), but inconsistent with the AFM-assisted shearing measurements on circular mesas of 100–600 nm in diameter ($0.227\pm 0.005 \text{ Jm}^{-2}$ [17]). We revisited the lateral stiffness calibration of all probes used in ref. [17] by means of a 3D finite element simulation, predicting consistently stiffer (~ 1.6 times) probes than those described in the original work (Fig. S11). Using this modified lateral spring constant yields an IAE value of $0.362\pm 0.008 \text{ Jm}^{-2}$, more consistent with our measurements. We also note that to the best of our knowledge no IAE measurement on the hBN homointerface yet exists, while there is only one measured IAE value of 0.22 Jm^{-2} at the MoS_2 homointerface using a nanomechanical cleavage technique [18], which is lower than half of our values. We believe that in their IAE calculations, a very low bending stiffness value of 0.92 eV was used for the monolayer MoS_2 which is even lower than that of monolayer G (1.49 eV) and monolayer hBN (1.34 eV) whose thicknesses are almost half of MoS_2 thickness, resulting in such a low IAE value in their MoS_2 homostructure (Section S7).

Main text, Page 7, Paragraph 2

During the revision of this paper, Li et al. reported the vdW interactions of G/hBN and G/ MoS_2 heterostructures, where a Si AFM tip wrapped with a thin G flake is brought into contact with pre-annealed G, hBN and MoS_2 substrates in high vacuum at room temperature [19]. Using a G-wrapped sharp tip with an unknown contact area only allows the measurements of the critical adhesion forces (i.e., pull-off forces, P) between G/G, G/hBN and G/ MoS_2 . Qualitatively speaking, their measurements showed that G experiences a weaker vdW interaction with hBN and G than MoS_2 , yielding a critical adhesion force ratio of $P_{\text{G}/\text{MoS}_2}/P_{\text{G}/\text{hBN}}=1.079$ and $P_{\text{G}/\text{MoS}_2}/P_{\text{G}/\text{G}}=1.028$. Similarly, our IAE ratios of $\Gamma_{\text{G}/\text{MoS}_2}/\Gamma_{\text{G}/\text{hBN}}$ and $\Gamma_{\text{G}/\text{MoS}_2}/\Gamma_{\text{G}/\text{G}}$ for the roughly similar experimental conditions (i.e., precooling-treated heterointerfaces annealed at 130°C) are 1.088 and 1.059, respectively, which are qualitatively in good agreement with their findings.

Main text, Page 11, Paragraph 1

Such intimate and strong interaction of G/ SiO_x suggests that the electron scattering sites across the interface as well as the convex sites of corrugated G result in the formation of short-range chemical bonds which act as anchoring spots to locally pin G to the SiO_x surface at the location of such chemically active sites [32, 33, 34, 35]. Since monolayer G with extreme flexibility possesses more chemically active sites than multilayer G at the G/ SiO_x interface, stronger adhesion energy of monolayer to SiO_x is expected, as previously confirmed by a pressurized blister test to be $0.45\pm 0.02 \text{ Jm}^{-2}$ for monolayer G but $0.31\pm 0.03 \text{ Jm}^{-2}$ for multilayer G [21].

Main text, Page 12, Paragraph 1

Our further analysis on the temporal evolution of the adhesion energy (see Section S6 for our detailed analysis) and contamination thickness measured on the mechanically exfoliated G surface during the first 60 min of air exposure reveals that its intrinsic IAE of $0.341\pm 0.025 \text{ Jm}^{-2}$ obtained under ultrahigh vacuum or ultrahigh-purity argon atmosphere is well consistent with our experimental value of $0.328\pm 0.028 \text{ Jm}^{-2}$ but drastically decreases within the first minute of air exposure and eventually approaches a saturated value of $0.15\pm 0.02 \text{ Jm}^{-2}$ after 10 min (Fig. 5), which is smaller than our IAE value of $0.233\pm 0.035 \text{ Jm}^{-2}$ upon room-temperature storage for 1 h. This could be attributed to the presence of the contact pressure and the vdW interaction between the layers in our experiments which may still play a role to squeeze away the contaminants even at room temperature, leaving cleaner interfaces with stronger interactions.

We also note that a substantial decrease in the surface hydrocarbon level under vacuum, high-temperature (500–1000°C) treatment during the WCA measurements results in the IAE recovery of the G crystal ($0.282\pm 0.024 \text{ Jm}^{-2}$), which is in good agreement with our IAE value of $0.268\pm 0.028 \text{ Jm}^{-2}$ at much lower temperature (300°C), further confirming the dominant contribution of the vdW force to the IAE improvement.

Section S5. Comparative studies of IAE

In this section, we perform a comprehensive comparison study on the interfacial adhesion energy (IAE) of 2D crystals and 2D crystal/SiO_x heterostructures obtained from a wide range of experimental methods. While a vast majority of studies have been conducted on the interaction of G with G (Section S5.1) and SiO_x (Section S5.2) substrates with a wide range of reported IAE values, to the best of our knowledge, no IAE measurement at the hBN/hBN and hBN/SiO_x interfaces yet exists, and also there are a very limited number of reports on the interaction of MoS₂ with MoS₂ (Section S5.1) and SiO_x (Section S5.2) substrates. We also note that, to the best of our knowledge, there is no direct IAE measurement on the 2D crystal heterostructures.

Section S5.1. Comparison study on cohesion energy of 2D crystal homostructures

Although many attempts have been made over the last six decades to measure the cohesion energy of G crystal with the reported values ranging from 0.15–0.72 Jm⁻² (Table S1), there are few direct measurements of cohesion energy available for comparison. From the literature data, we found that our measurements for cohesion energy of G crystal are in excellent agreement with micro-force sensing probe measurements on 4 μm wide square mesas ($0.37\pm 0.01 \text{ Jm}^{-2}$ [13]) and AFM-assisted shearing measurements on 3 μm wide square mesas (0.35 Jm^{-2} [14]), but inconsistent with the recent AFM-assisted shearing measurements on circular mesas of 100–600 nm in diameter ($0.227\pm 0.005 \text{ Jm}^{-2}$ [15]).

Table S1. Cohesion energy of carbon nanotubes, few-layer graphene, and graphite.

Method	Sample	Stack	Γ (J/m ²)	Ref
Heat of wetting	Graphite	N.A.	0.26±0.03	[21]
Radial deformation of MWCNT	Collapsed MWCNT	(Non-)AB*	0.15–0.31	[22]
Thermal desorption	HOPG	AB	0.37±0.03	[23]
MWCNT retraction	MWCNT	Non-AB	0.28–0.4	[24]
Deformation of thin sheets	HOPG	AB	0.19±0.01	[25]
AFM pull-off force measurements	HOPG	Non-AB	0.319±0.05	[26]
DWCNT inner-shell pull-out	DWCNT	Non-AB	0.436±0.074	[27]
SEM peeling of MWCNT	Collapsed MWCNT on 1-LG	(Non-)AB	0.40±0.18	[28]
	Flattened MWCNT on 1-LG		0.72±0.32	
AFM-assisted mechanical shearing	HOPG	Non-AB	0.227±0.005	[15]
Self-retraction motion	HOPG	AB	0.39±0.02	[13]
		Non-AB	0.37±0.01	
AFM-assisted mechanical shearing	HOPG	Non-AB	0.35	[14]
AFM nano-indentation	BLG/FLG onto FLG	Non-AB	0.307±0.041	[29]
Atomic intercalation of neon ion	1LG onto HOPG	Non-AB	0.221±0.095	[30]
Surface force balance	CVD-grown 1LG/1LG	Non-AB	0.230±0.008	[31]
	CVD-grown FLG/FLG	Non-AB	0.238±0.006	

Section S5.2. Comparison study on IAE of 2D crystal/SiO_x

Despite many experimental studies devoted to the IAE determination of 2D crystals/SiO_x heterostructures, no experimental data are available on the interaction of hBN/SiO_x, whereas the reported IAE data on the interaction of G and MoS₂ with SiO_x are very diverse, ranging from 0.09–0.90 Jm⁻² at the

G/SiO_x interface (**Table S3**) and 0.08–0.48 Jm⁻² at the MoS₂/SiO_x interface (**Table S4**). We believe that a part of this large data scattering can be attributed to different surface properties of SiO_x during sample preparation, leading to different surface roughness, surface configurations (due to its amorphous nature), surface polarities, charge impurities, surface reactions with ambient humidity, and type of surface termination/defects (i.e., H-, Si- and O-terminated surfaces).

Table S3. Interlayer adhesion energy of carbon nanotubes, few-layer graphene, and graphite on SiO_x.

Method	Sample	Γ (J/m ²)	Ref
AFM nano-indentation	BLG/FLG	0.270±0.020	[29]
Pressurized blister	1LG	0.45±0.02	[32]
	2-5LG	0.31±0.03	
Pressurized blister	1LG	0.24	[33]
Pressurized blister	1LG	0.140±0.040	[34]
	5LG	0.160±0.060	
AFM with a microsphere tip	1LG	0.46±0.02	[35]
Intercalation of nanoparticles	5LG	0.302±0.056*	[36]
Infrared crack opening Interferometry	1LG	0.357±0.016	[37]
Nanoparticle-loaded blister	1LG	0.453±0.006	[38]
	3-5LG	0.317±0.003	
	10-15LG	0.276±0.002	
Intercalation of nanoparticles	FLG	0.567	[39]
Colorimetry technique	2LG	0.9	[40]
Interfacial nanoblister	1LG	0.093±0.001	[41]

Table S4. Interlayer adhesion energy of MoS₂ on SiO_x.

Method	Sample	Γ (J/m ²)	Ref
Intercalation of nanoparticles	FL	0.482	[39]
Pressurized blister	1L	0.212±0.037	[42]
	2L	0.166±0.004	
	3L	0.237±0.016	
	1L CVD	0.236±0.021	
Wrinkle	FL	0.170±0.033	[43]
Interfacial nanoblister	1L	0.082±0.001	[41]

Table S7. Summary of water contact angle measurements and corresponding IAE values of G crystal.

Notes	Measured within	WCA	$2\gamma_G$ (J/m ²)	Ref
Ultrahigh vacuum	3 sec	42±7°	0.348±0.033	[44]
Ambient air at 24°C/48% RH	10 sec	64.4°	0.232	[45]
	2 days	91.0°	0.093*	
550°C annealing in Ar	1 min	54.1°	0.286	
Ambient air at 22-25°C/20-40% RH	10 sec	64.4±2.9°	0.232±0.015	[46]
	7 days	97.0±1.8°	0.072±0.01*	
Ultrahigh vacuum for 15 h	N.A.	59°	0.260	
Ambient air at 22-25°C/20-40% RH	10 sec	68.6±7.1°	0.210±0.034	[47]
Ambient air at 22°C/50% RH	10 sec	68.2±2°	0.212±0.010	[48]
	1 day	90±0.1°	0.096±0.002*	
Ambient air at RT/40-50% RH	1.5 min	62.9±2.2°	0.239±0.012	[49]
Ambient air at RT/40-50% RH	5 min	61.8±3.3°	0.249±0.017	
	1 day	81.9±2.9°	0.129±0.012*	
600°C annealing in He	N.A.	51.4±2.0°	0.300±0.010	
Clean room at 21°C/40% RH	5 sec	53±5°	0.292±0.027	[50]
	8 min	66±3°	0.223±0.016	
	2 days	86±4°	0.112±0.016*	
Water vapor atmosphere	N.A.	58±2°	0.266±0.010	
Ultrahigh-purity argon atmosphere	1 min	45±3°	0.333±0.016	
Evacuation/1050°C annealing/vacuum	1 min	55±1°	0.281±0.005	
Evacuation/1000°C annealing/atmosphere	1 min	73±5°	0.187±0.025	
Ambient air at 22-25°C/20-40% RH	10 sec	60±0.1°	0.255±0.002	[51]

3. The thermal fluctuation at an increased temperature will increase the interlayer distance of 2D materials and therefore tend to decouple the interlayer interaction. This, however, is in contrast to the authors' results that the IAE increases with increasing the temperature. I'm not convinced by their analyses and explanation.

As we mentioned in the manuscript, all AFM measurements were performed at room temperature ($T = 22^\circ\text{C}$) under different annealing temperatures. As such, the interlayer distance of 2D materials remains unchanged when the temperature at the interface cools back down to room temperature for the subsequent *F-d* measurements and, therefore, has no contribution to the reported IAE of 2D crystals. Also, from Figs. 2a-c, the IAE may increase or decrease with the increase of the annealing temperature at the interface, as we discussed in the main text.

4. The MoS₂ has a higher chemical activity than graphene. Why graphene is shown to chemically bond to SiO₂ but MoS₂ does not? Here, I'm also not convinced by the mechanism proposed for explaining the larger IAE of graphene on SiO₂.

As the respected reviewer correctly mentioned, MoS₂ has a higher chemical activity than G, in particular, at higher temperatures, as confirmed by Figs. 2b and 2c, showing the negative impact of the annealing temperature on the IAE of MoS₂/G, MoS₂/hBN and MoS₂/SiO_x heterostructures due to the surface oxidation of MoS₂. However, unlike the G crystal, the basal plane of MoS₂ is rather inert unless S vacancies are introduced into its basal plane [2]. Short-range chemical reactions between MoS₂ and SiO_x require (1) vacancy defects in the MoS₂ basal plane to directly bind H, O and Si atoms to exposed Mo atoms and (2) a close conformation of MoS₂ to the underlying SiO_x substrate. For the former one, our surface topography measurements in the absence of the thermal annealing do not exhibit dangling bonds/vacancy defects and grain boundaries in the basal plane of MoS₂ nanomesas, whereas the high annealing temperature of MoS₂ can replace S atoms with O atoms to form the oxidized MoS₂ (MoO₃), **as confirmed by our x-ray photoelectron spectroscopy (XPS) measurements (Fig. S10)**. For the latter one, our surface study of the corrugation of bilayer G and monolayer MoS₂ with almost the same thickness (i.e., 0.670 nm in 2LG versus 0.645 nm in 1LMoS₂) demonstrates that the degree of conformation of MoS₂ to the SiO_x is much lower than that of G (Fig. 4). These two reasons can make the formation of the chemical bonds at the MoS₂/SiO_x interface almost impossible at least when the contact forms at room temperature, as is the case in Fig. 3 for the G/SiO_x heterostructures.

[2] J. Zhu, Z.C. Wang, H. Dai, Q. Wang, R. Yang, H. Yu, M. Liao, J. Zhang, W. Chen, Z. Wei, N. Li, L. Du, D. Shi, W. Wang, L. Zhang, Y. Jiang and G. Zhang. Boundary activated hydrogen evolution reaction on monolayer MoS₂, Nature Communications, 10: 1348 (2019).

To make it clear, we added this discussion in the main text as follows:

Page 11, Paragraph 2:

Figs. 2b and 2c suggest that MoS₂ has a higher chemical activity than G, in particular, at higher temperatures. However, unlike the G crystal, the basal plane of MoS₂ is rather inert unless S

vacancies are introduced into its basal plane [39]. Short-range chemical reactions between MoS₂ and SiO_x require (1) vacancy defects in the MoS₂ basal plane to directly bind H, O and Si atoms to exposed Mo atoms and (2) a close conformation of MoS₂ to the underlying SiO_x substrate. For the former one, our surface topography measurements in the absence of the thermal annealing do not exhibit vacancy defects and grain boundaries in the basal plane of MoS₂ nanomesas. For the latter one, we showed that the degree of conformation of MoS₂ to the SiO_x is much lower than that of G (Fig. 4). These two reasons make the formation of the chemical bonds at the MoS₂/SiO_x interface almost impossible at least when the contact forms at room temperature, as is the case in Fig. 3 for the G/SiO_x heterostructures.

5. All abbreviations in this paper, such as ‘G’, ‘SEM’ and ‘HOPG’, need explanations.

We thank the respected reviewer for pointing this out. We fixed these and also double-checked the whole manuscript.

graphite (G)

scanning electron microscopy (SEM)

highly oriented pyrolytic graphite (HOPG)

6. For the force-displacement measurements, how to make sure the pressure exerted by the nanomesas to the sample is close to zero so that the displacement is governed only by the intrinsic interfacial interaction force.

In the interface software XEP, we set the preload to be zero upon the tip approach, maintaining the zero applied normal force throughout the contact by adjusting the Z piezo scanner using the real-time feedback from the deflection of the AFM tip.

REVIEWER COMMENTS

Reviewer #1 (Remarks to the Author):

The authors revised the manuscript extensively to address the issues raised by the reviewers. The revisions and the authors' explanations in their 'Response' are mostly satisfactory. The manuscript is easier to follow as a result. I only have one minor suggestion. The Supplementary Information has all the figures at the end of the document. Since the SI will not be edited by the publisher, I suggest that the figures be placed within the text where they are mentioned so that the readers do not have to go back and forth.

Reviewer #2 (Remarks to the Author):

The authors have made great efforts to clarify the questions and concerns raised in my review report. Citation of related literature and detailed comparison with the data reported previously made more meaningful discussion of the present measurements. I thank the authors for all the improvement and the positive responses.

The most important advance for using nanomesas to measure the interfacial interactions of 2D materials is that the cleavage area of the mesa is known, so that the cleavage stress and critical shear stress can be determined from the maximum value of the measured force. In fact, the IAE is presented and analyzed in the manuscript per unit area. However, as the detaching F-d curves have different unloading shapes, the adhesive energy is lack of clean physical meaning. In comparing, the maximum stress should be more important to characterize the cleavage strength of different nanomesa of the 2D materials. It is suggested to present the force in form of per unit area, or in nominal stress. And compare them with theoretical prediction and previous measurements, especially for the first time to break the mesa as the new cleavage surface has not exposed to air pollution.

Also, after air exposure, using of stress may reflect the contact condition better.

More issues raised:

1. The authors added 7 closely related references in the introduction simply by stating "Despite the significance of such a fundamental property for any layered materials, there have been relatively limited experimental and theoretical methods with significant diversity in the reported IAE values for 2DLMs in general and graphite (G) crystals specifically, where the exact cause of the variation in their IAE values has also remained to be elucidated by a comprehensive and accurate experimental technique [15, 16, 17, 18, 19, 20, 21]."

The related important advances made by these references should be discussed as a background for readers to understand what new advances made in the present work.

2. In the main concern #3 of the last comment, I suggested that literatures of related theoretical investigation on the interfacial interaction should be introduced and discussed in the introduction, which should be addressed better.

3. In the response of technique issue #7, the authors declared that there is no chemical bond/dangling bond both at the edge and along the sliding plane of nanomesas based on the mild detachment through the F-d curves. This is not convinced without any characterizations of the sample and solid evidence. Especially, the etching procedure would also introduce significant charge pollution in the nanomesas, which is not considered, neither.

4. The phrases 2D material and van der Waals heterostructure in the title have the same meaning, thus, the title should be revised as "Direct Measurements of Interfacial Adhesion in 2D Materials in Ambient Air". Especially as the recommended measurement in vacuum has not conducted.

5. The abstract should be rewritten to include main results and conclusions, rather than the present one simply describing what work did but lack of necessary information about what

achieved for the readers.

6. The light blue-shaded area in Fig. 2b represents the integral of the retraction force as a function of the piezo displacement, however, the IAE means the integral of the adhesion force as a function of distance between the tip and substrate. Thus, the method for calculating IAE described in the main text should be explained in detail.

7. Figure 1c provides the shear force–lateral piezo displacement curves only with the range of piezo displacement from 0 to 12 nm, based on that, how did the authors calculate the value of interfacial adhesion energy?

8. In Fig. 3c, it is unreasonable that the calculated curves do not coincide with each other when the relative displacement is larger than 0.2 nm, because both curves represent the same van der Waals interaction between the graphene and SiO_x.

9. The reason why the relatively gradual reduction of the adhesion force was observed in the retraction force measurements (rather than a snap-back to zero force) as described in the revised manuscript (lines 101 ~103 in manuscript) is not positively explained except for the discussions of excluding the effect of tip-sample capillary forces. Besides, the red curve shown in Fig. 1d was measured in 70% humidity instead of 15%, which is not appropriate for the purpose of comparison.

10. The authors demonstrate that F–d curves upon tip approach display a small jump-to-contact force and suggest negligible effect of tip-sample capillary forces (revised manuscript, lines 103-106). Thus, the approach F–d curve of Si/SiO_x measurement (red curve in Fig. 1d) should be exhibited for comparison.

11. All three materials in the experiment are multi-layer flakes, thus, the comparison of the bending stiffness between different materials (lines 376 ~380 in manuscript) are in conflict with the recent paper (Phys. Rev. Lett. 123, 116101, 2019), the following conclusion should be reconsidered.

12. Did the separation take place across the thickness of G nanomesa for the data points Γ_{G/SiO_x} higher than $\Gamma_{G/G}$ in Fig. 3b?

13. The water layer between the Si-tip and substrate (Fig. 1d) should be called with corresponding terminology instead of “bubble” to avoid misleading the readers.

14. Some writing errors need to be revised. Eg. in red (legend of Fig. 2d for Si/SiO_x); volume and page numbers for reference 19; the scale bar in Fig. 1c.

15. The last, but most important issue is that the logic needs careful organization, the whole manuscript, each section, and the materials in the supplementary information.

Reviewer #3 (Remarks to the Author):

The authors have elaborated their responses to my comments raised in the first round of review. Basically, I'm satisfied with most of the replies. However, some issues remain in the revised manuscript that need further consideration by the authors.

1) Something mentioned in the authors' reply to my first comment are confusing. For example, tip-attached nanomesa, tip-attached 2D crystal, exfoliation of monolayer or few layers from tip-attached 2D crystal onto the sample, and contact spots are not easy to find what are referring to. The authors need to improve these statements and better add a figure for illustration.

2) The authors are suggested to explain or discuss why the interfacial cohesion energy across the 2D crystal nanomesa is larger than the interfacial adhesion energy at all 2D crystal tip-sample interfaces.

3) Some discussion on the results need to be strengthened. For example, in figure 1d, the behavior of separation between G-G is distinct from that between G-Cu. It is known that G-Cu interaction are very weak, about two-fold of the G-G interaction. Then why the former case displays a sharp detachment but the latter case shows a viscoelastic detachment? Theoretical calculations (first-principles etc) would be helpful here for offering deep insight into the interfacial detachment.

Response to Reviews

We thank the reviewers for their careful reviews and valuable suggestions; we have addressed all the comments in the revised manuscript. All revised parts were highlighted in the manuscript.

Reviewer #1 (Remarks to the Author):

The authors revised the manuscript extensively to address the issues raised by the reviewers. The revisions and the authors' explanations in their 'Response' are mostly satisfactory. The manuscript is easier to follow as a result. I only have one minor suggestion. The Supplementary Information has all the figures at the end of the document. Since the SI will not be edited by the publisher, I suggest that the figures be placed within the text where they are mentioned so that the readers do not have to go back and forth.

We thank the respected reviewer for all his/her valuable inputs and comments. Following the reviewer's suggestion, in Supplementary Information we have placed the tables and figures within the text where they are mentioned. So the readers do not have to go back and forth.

Reviewer #2 (Remarks to the Author):

The authors have made great efforts to clarify the questions and concerns raised in my review report. Citation of related literature and detailed comparison with the data reported previously made more meaningful discussion of the present measurements. I thank the authors for all the improvement and the positive responses.

We thank the respected reviewer for all his/her valuable inputs and comments. In the following, we provide a point-by-point response to the reviewer's comments.

The most important advance for using nanomesas to measure the interfacial interactions of 2D materials is that the cleavage area of the mesa is known, so that the cleavage stress and critical shear stress can be determined from the maximum value of the measured force. In fact, the IAE is presented and analyzed in the manuscript per unit area. However, as the detaching F - d curves have different unloading shapes, the adhesive energy is lack of clean physical meaning. In comparing, the maximum stress should be more important to characterize the cleavage strength of different nanomesa of the 2D materials. It is suggested to present the force in form of per unit area, or in nominal stress. And compare them with theoretical prediction and previous measurements, especially for the first time to break the mesa as the new cleavage surface has not exposed to air pollution. Also, after air exposure, using of stress may reflect the contact condition better.

As the respected reviewer mentioned, both interfacial adhesion energy and interfacial normal/shear strength are fundamental properties of 2D layered materials, each providing different insights into the interlayer mechanical properties of layered materials. While the interfacial normal/shear strength can be obtained from the information of one single point on the F - d curve (i.e., the maximum normal/shear force required to initiate detachment/sliding), the interfacial adhesion energy is obtained from the information of all points on the F - d curve, providing valuable insight into the whole separation process. The importance of both interfacial adhesion energy and interfacial normal/shear strength is also evident from the number of papers published after the first report of interfacial adhesion energy and of interfacial normal strength of graphite, dating back to 1954 and 1966, respectively. Therefore, following the tradition and recognizing that interfacial adhesion energy and interfacial strength are two fundamentally important and different properties, we decided to prepare two separate papers, each containing different experimental results and analyses with about 100 references and 20-30 pages of supplementary information.

While the present work has focused on the IAE of 2D materials, we are currently drafting a paper about the interlayer elastic properties of 2D materials (i.e., out-of-plane elastic constant C_{33} , interlayer shear elastic constant C_{44} , out-of-plane tensile/cleavage strength σ_{33} and interlayer shear strength τ_s) using our normal/shear force-displacement and Raman spectroscopy measurements. As the reviewer correctly mentioned, while the cleavage strength can be obtained by $\sigma_{33} = P/A$, where P is the pull-off force and A is the interface area, by definition of the interlayer shear strength at the sliding interface, $\tau_s = F_s^m/A$, where F_s^m is the maximum shear force required to initiate sliding and thus is equal to the interfacial adhesion force F_a at $x = 0$, where

Circular mesa:

$$F_a(x) = -\Gamma D \sqrt{1 - \left(\frac{x}{D}\right)^2}$$

Square mesa:

$$F_a = -\Gamma w$$

As a result, the interlayer shear strength can be given by

Circular mesa
$$\tau_s = \frac{4\Gamma}{\pi D}$$

Square mesa
$$\tau_s = \frac{\Gamma}{w}$$

As it can be seen, the interlayer shear strength is inversely proportional to the diameter (length) of the circular (rectangular) mesas, implying that the smaller mesas can unexpectedly sustain greater shear stresses. This distinct shear behavior of 2D materials confirms that the interlayer shear strength is size and shape dependent rather than being an intrinsic material property, well consistent with our new experimental results. In this ongoing research project, we have also collected the interlayer mechanical properties of 2D materials obtained from a wide range of experimental and theoretical methods reported since 1950's in more than 100 articles (please see, for example, the following table for few-layer graphene and graphite)

Table S2 | Interlayer elastic properties of few-layer graphene and graphite.

Method	Sample	Stack	c/2 (Å)	C ₃₃ (GPa)	C ₄₄ (GPa)	σ ₃ (MPa)	τ _s (MPa)	Ref
Ultrasonic test	Single-crystal graphite	AB			4.0±0.4			1
Ultrasonic/static tests	Pyrolytic graphite	Non-AB		36.5±1	0.18–0.35		0.88–2.45	2
Ultrasonic test	Pyrolytic graphite	Non-AB		36.6	0.281			3
Neutron scattering	HOPG	AB		37.1±0.5	4.6±0.2			4
Brillouin scattering	Graphite	AB			5.05±0.35			5
X-ray scattering	Single-crystal graphite	Non-AB	3.353	36.6±1.2				6
X-ray scattering	Single-crystal graphite	AB	3.356	38.7±0.7	5.0±0.3			7
X-ray diffraction	Single-crystal graphite		3.354	35.7±2.5				8
Raman spectroscopy	Few-layer graphene	AB			4.3			9
Specific heat	Canadian natural graphite	N.A.		>18	2.3			10
Thermal noise excitation	HOPG						2000–7000	11
Static tests	Polycrystalline graphite	Non-AB			4.0±0.05	10.3–20.7		12
Uniaxial-shear stress	Single-crystal graphite				4.5±0.6		0.25–0.75	13
Partial dislocation motion	HOPG						5–200	14
Self-retraction motion	HOPG	AB					100±40	15
Self-folding conformation	Few-layer graphene	Non-AB			0.36–0.49			16
Friction force microscope	HOPG	AB					80–120°	17
Modulated nanoindentation	HOPG			33±3				18
	10-layer epitaxial graphene			36±3				
AFM-assisted shearing	HOPG	Non-AB					0.5–3.1**	19
Self-retraction motion	HOPG	Non-AB					0.12***	20
AFM-assisted motion	HOPG	Non-AB					0.057	21

- [1] E.J. Seldin, in *Proceedings of Ninth Biennial Conference on Carbon*, Chestnut Hill, Massachusetts, 1969 (Defense Ceramic Information Center, Columbus, Ohio, 1969) p. 59.
- [2] O. L. Blakslee, D. G. Proctor, E. J. Seldin, G. B. Spence, and T. Weng, *J. Appl. Phys.* **41**, 3373 (1970).
- [3] Gauster, W. B. & Fritz, I. J. Pressure and temperature dependences of the elastic constants of compression-annealed pyrolytic graphite. *J. Appl. Phys.* **45**, 3309 (1974).
- [4] R. Nicklow, N. Wakabayashi, and H. G. Smith, *Phys. Rev. B* **5**, 4951 (1972).
- [5] M. Grimsditch, Shear elastic modulus of graphite, *J. Phys. C* **16**, L143 (1983).
- [6] Wada, N., Clarke, R. & Solin, S. A. X-ray compressibility measurements of the graphite intercalates KC₈ and KC₂₄. *Solid State Commun.* **35**, 675 (1980).
- [7] Bosak, A., Krisch, M., Mohr, M., Maultzsch, J. & Thomsen, C. Elasticity of single crystalline graphite: Inelastic X-ray scattering study. *Phys. Rev. B* **75**, 153408 (2007).

- [8] M. Hanfland, H. Beister, K. Syassen. Graphite under pressure: Equation of state and first-order Raman modes. *Phys. Review B*. 39, 12598 (1989).
- [9] P. H. Tan , W. P. Han , W. J. Zhao , Z. H. Wu , K. Chang , H. Wang , Y. F. Wang , N. Bonini, N. Marzari , N. Pugno , G. Savini , A. Lombardo , and A. C. Ferrari , *Nat. Mater.* 11, 294 (2012).
- [10] J. C. Bowman and J. A. Krumhansl, The low-temperature specific heat of graphite, *J. Phys. Chern. Solids* 6, 367 (1958).
- [11] Ding, X. D., Wang, Y. Z., Xiong, X. M., et al.: Measurement of shear strength for HOPG with scanning tunneling microscopy by thermal excitation method. *Ultramicroscopy* 115, 1–6 (2011)
- [12] E.J. Seldin, Stress–strain properties of polycrystalline graphites in tension and compression at room temperature, *Carbon* 4 (1966) 177–191.
- [13] Soule D E and Nezbeda C W, Direct Basal-Plane Shear in Single-Crystal Graphite. *J. Appl. Phys.* 39 5122(1968).
- [14] Snyder, S. R., Gerberich, W. W., White, H. S., Scanning tunneling-microscopy study of tip-induced transitions of dislocation-network structures on the surface of highly oriented pyrolytic graphite. *Phys. Rev. B* 47, 10823 (1993).
- [15] Z. Liu, J. Yang, F. Grey, J.Z. Liu, Y. Liu, Y. Wang, Y.L. Yang, Y. Cheng, Q.S. Zheng, Observation of microscale superlubricity in graphite. *Phys. Rev. Letters* 108, 205503 (2012).
- [16] X. Chen, C. Yi and C. Ke, Bending stiffness and interlayer shear modulus of few-layer graphene. *Appl. Phys. Lett.* 106, 101907 (2015).
- [17] G.S. Verhoeven, M. Dienwiebel, J.W.M. Frenken. Model calculations of superlubricity of graphite. *Phys. Rev. B* 70, 165418 (2004).
- [18] Y. Gao, A. Kim, S. Zhou, H.C. Chiu, D. Nelias, C. Berger, et al. Elastic Coupling between layers in two-dimensional materials. *Nature Materials*, 14, 714-720 (2015).
- [19] E. Koren, E. Lörtscher, C. Rawlings, A.W. Knoll, U. Duerig. Adhesion and friction in mesoscopic graphite contacts. *Science* Vol. 348 no. 6235 pp. 679–683 (2015).
- [20] W. Wang, S. Dai, X. Li, J. Yang, D. J. Srolovitz, Q. Zheng. Measurement of the cleavage energy of graphite. *Nature Communications*, 6 (2015):7853.
- [21] C.C. Vu, S. Zhang, et al. Observation of normal-force-independent superlubricity in mesoscopic graphite contacts, *Phys Rev B*, 081405 (2016).

More issues raised:

1. The authors added 7 closely related references in the introduction simply by stating “Despite the significance of such a fundamental property for any layered materials, there have been relatively limited experimental and theoretical methods with significant diversity in the reported IAE values for 2DLMs in general and graphite (G) crystals specifically, where the exact cause of the variation in their IAE values has also remained to be elucidated by a comprehensive and accurate experimental technique [15, 16, 17, 18, 19, 20, 21].”

The related important advances made by these references should be discussed as a background for readers to understand what new advances made in the present work.

Following the reviewer’s comment, we modified Introduction as follows:

Page 2, Paragraph 3:

The interfacial physical and chemical behavior of layered materials becomes even more complicated when we consider that airborne contaminants are an inevitable part of any vdW heterostructures and therefore addressing quantitatively to what degree their interfacial adhesion energy (IAE) is influenced by interfacial contaminants and nanoblisters and how to effectively remove them is of fundamental and technological importance for the continued development of such promising materials. However, many attempts have been made over the last six decades to measure the IAE of 2D crystals either in high vacuum or under a contamination-free environment. Among them, few direct IAE measurements of 2D crystals have been reported with a particular focus on graphite (G) crystal [15-21]. For instance, the IAE at the intact G/G homointerface was reported using micro-force sensing probe measurements on 4 μm

wide square mesas ($0.37 \pm 0.01 \text{ Jm}^{-2}$ [15]) and AFM-assisted shearing measurements on $3 \mu\text{m}$ wide square mesas (0.35 Jm^{-2} [16]) and circular mesas of 100–600 nm in diameter ($0.227 \pm 0.005 \text{ Jm}^{-2}$ [17]). Moreover, there is only one measured IAE value of 0.22 Jm^{-2} at the $\text{MoS}_2/\text{MoS}_2$ homointerface using a nanomechanical cleavage technique [18], whereas, to the best of our knowledge, no IAE measurement at the hBN/hBN homointerface yet exists. Also, the vdW interaction at G/hBN and G/ MoS_2 heterointerfaces was studied using a G-wrapped sharp tip with an unknown contact area, allowing the measurements of critical adhesion forces between G/G, G/hBN and G/ MoS_2 in high vacuum at room temperature [19]. Although a considerable number of experimental and theoretical methods have been proposed to study the IAE of 2D crystals in general and G crystal specifically, there is significant diversity in the reported IAE values, where the exact cause of the variation in their IAE values has still remained to be elucidated by a comprehensive and accurate experimental technique.

2. In the main concern #3 of the last comment, I suggested that literatures of related theoretical investigation on the interfacial interaction should be introduced and discussed in the introduction, which should be addressed better.

Given the experimental nature of our work, we have mainly focused on experimental results and relevant experimental techniques and used theoretical papers (see, for example, Ref [20] in the main text), where needed, to support our results. Forced by page limit, we have to provide theoretical investigation in a condensed form. The following table, for instance, contains the summary of binding energy (BE), exfoliation energy (EE) and cleavage energy (CE) of graphite crystal, showing the number of papers on the IAE of graphite obtained by various theoretical methods. As can be seen by the long list of reference, a detailed discussion of theoretical investigation of the IAE of 2D materials should be in a different context. The main text is at the limit and the reference in main text and supplementary information is already 100 references, which is not very common for a research paper (a review paper would be fine). This motivated us to draft a review paper on the vdW interaction of 2D materials with over 250 references, including both theoretical and experimental papers. We are discussing with the editor about this option.

Table 4 | Summary of interfacial adhesion energy of graphite obtained by various theoretical methods (in the unit of Jm^{-2}).

Method	Sample	d_0 (Å)	Interfacial adhesion energy	Ref
DFT-LDA	Graphite (AB)	3.310	$0.31^{(\text{BE})}$	22
vdW-DF	Graphite	3.760	$0.15^{(\text{BE})}$	23
vdW-DF	Graphite (AB)	3.590	$0.34^{(\text{EE})}$	24
vdW-DF	Graphite	3.350	$0.51^{(\text{EE})}$	25
vdW-DF	Graphene (AB)	3.600	$0.28^{(\text{EE})}$	26
	Graphite (AB)		$0.31^{(\text{CE})}; 0.29^{(\text{EE})}$	
Semiempirical-LDA+vdW	Graphite (AB)	3.313	$0.36^{(\text{BE})}$	27
QMC	Graphite (AB)	3.426	$0.34^{(\text{BE})}$	28
ACFDT-RPA	Graphite (AB)	3.340	$0.29^{(\text{BE})}$	29
	Graphite (AA)	3.420	$0.23^{(\text{BE})}$	
SAPT-DFT	Graphene (AB)	3.430	$0.26^{(\text{BE})}$	30
	Graphite (AB)	3.420	$0.30^{(\text{CE})}; 0.28^{(\text{EE})}; 0.28^{(\text{BE})}$	
DFT-D	Graphene (AB)	3.250	$0.31^{(\text{BE})}$	31
	Graphene (AA)	---	$0.19^{(\text{BE})}$	
	Graphite (AB)	3.220	$0.35^{(\text{BE})}$	
LCAO-S2+vdW	Graphite (AB)	3.340	$0.49^{(\text{BE})}$	32
	Graphite (AA)	3.600	$0.43^{(\text{BE})}$	
	Graphite (Incommensurate)	3.420	$0.43 \pm 0.02^{(\text{BE})}$	
DFT-LDA	Graphite (AB)	3.380	$0.15^{(\text{BE})}$	33
RPA	Graphite (AB)	3.340	$0.29^{(\text{BE})}$	33
vdW-DF1	Graphite (AB)	3.600	$0.30^{(\text{BE})}$	33
vdW-DF2	Graphite (AB)	3.470	$0.30^{(\text{BE})}$	33

PBE/vdW-DF1	Graphite (AB)	3.446	0.41 ^(BE)	33
VV10	Graphite (AB)	3.389	0.43 ^(BE)	33
PBE-D	Graphite (AB)	3.370	0.39 ^(BE)	33
DFT-LDA	Graphite (AB)	3.334	0.14 ^(BE)	34
	Graphite (AA)	3.622	0.09 ^(BE)	
LDA/DFT-D2	Graphite (AB)	2.989	0.70 ^(BE)	34
	Graphite (AA)	3.198	0.34 ^(BE)	
PBE/DFT-D2	Graphite (AB)	3.231	0.34 ^(BE) ; 0.34 ^(EE) ; 0.35 ^(CE) ; 0.35 ^(SE)	34
	Graphite (AA)	3.492	0.25 ^(BE)	
optPBE/vdW-DF	Graphite (AB)	3.447	0.39 ^(BE)	34
	Graphite (AA)	3.625	0.35 ^(BE)	
optB88/vdW-DF	Graphite (AB)	3.356	0.43 ^(BE)	34
	Graphite (AA)	3.545	0.36 ^(BE)	
rPW86/vdW-DF2	Graphite (AB)	3.524	0.32 ^(BE)	34
	Graphite (AA)	3.670	0.23 ^(BE)	
revB86b/vdW-DF2	Graphite (AB)	3.360	0.34 ^(BE)	35
cx13/vdW-DF	Graphite (AB)	3.330	0.37 ^(BE)	35
optB88/vdW-DF	Graphite (AB)	3.380	0.40 ^(BE)	35
optB86b/vdW-DF	Graphite (AB)	3.360	0.40 ^(BE)	35
optPBE/vdW-DF	Graphite (AB)	3.350	0.36 ^(BE)	35
C09/vdW-DF	Graphite (AB)	3.270	0.43 ^(BE)	35
PBE-MBD	Graphite (AB)	3.400	0.29 ^(BE)	36
DMC	Graphene (AB)	3.384	0.11 ^(BE)	37
	Graphene (AA)	3.495	0.07 ^(BE)	
SAPT-DFT	Graphene (AB)		0.48 ^(BE)	38
PBE-D/DFT	Graphene (AB)	3.310	0.26 ^(BE)	39
RPA	Graphite (AB)	3.370	0.29 ^(CE)	40
PBE+TS+SCS	Graphite (AB)	3.302	0.31 ^(BE)	41
PBE+TS+SCS+MBD	Graphite (AB)	3.350	0.29 ^(BE)	42
PBE+D2	Graphite (AB)	3.222	0.34 ^(BE)	43
PBE+D3	Graphite (AB)	3.483	0.29 ^(BE)	43
PBE+D3-BJ	Graphite (AB)	3.373	0.32 ^(BE)	43
PBE+TS	Graphite (AB)	3.333	0.50 ^(BE)	43
PBE+TS+SCS	Graphite (AB)	3.317	0.33 ^(BE)	43
9-6LJ/analytical model	Graphene (AA)	3.400	0.40 ^(BE)	44

- [22] Wang, Y., Scheerschmidt, K. & Gösele, U. Theoretical investigations of bond properties in graphite and graphitic silicon. *Phys. Rev. B* 61, 12864, (2000)
- [23] Rydberg, H. et al. Van der Waals Density Functional for Layered Structures. *Phys. Rev. Lett.* 91, (2003) 126402.
- [24] Ziambaras, E., Kleis, J., Schröder, E. & Hyldgaard, P. Potassium intercalation in graphite: A van der Waals density-functional study. *Phys. Rev. B* 76, (2007).
- [25] Ortman, F., Bechstedt, F. & Schmidt, W. Semiempirical van der Waals correction to the density functional description of solids and molecular structures. *Phys. Rev. B* 73, (2006).
- [26] Chakarova-Käck, S., Schröder, E., Lundqvist, B. & Langreth, D. Application of van der Waals Density Functional to an Extended System: Adsorption of Benzene and Naphthalene on Graphite. *Phys. Rev. Lett.* 96, (2006).
- [27] Hasegawa, M., Nishidate, K. & Iyetomi, H. Energetics of interlayer binding in graphite: The semiempirical approach revisited. *Phys. Rev. B* 76, (2007).
- [28] Spanu, L., Sorella, S. & Galli, G. Nature and Strength of Interlayer Binding in Graphite. *Phys. Rev. Lett.* 103, (2009).
- [29] Lebègue, S. et al. Cohesive Properties and Asymptotics of the Dispersion Interaction in Graphite by the Random Phase Approximation. *Phys. Rev. Lett.* 105, (2010).
- [30] R. Podeszwa. Interactions of graphene sheets deduced from properties of polycyclic aromatic hydrocarbons, *J. Chemical Phys.* 132, 044704; 2010.
- [31] I.V. Lebedeva, A.A. Knizhnik, A.M. Popov, Y.E. Lozovik, and B.V. Potapkin, Interlayer interaction and relative vibrations of bilayer graphene. *Phys. Chem. Chem. Phys.* 13, 5687-5695; 2011.
- [32] Savini, G. et al. Bending modes, elastic constants and mechanical stability of graphitic systems. *Carbon* 49, 62-69, (2011).
- [33] T. Bjorkman, A. Gulans, A.V. Krasheninnikov, and R.M. Nieminen, *J. Phys.: Condens. Matter* 24, 424218; 2012.
- [34] X. Chen, F. Tian, C. Persson, W. Duan, N.X. Chen. Interlayer interactions in graphites. *Science Reports*, 3(2013):3046.
- [35] T. Bjorkman, *J. Chem. Phys.* 141, 074708; 2014.
- [36] W. Gao, A. Tkatchenko, Sliding mechanisms in multilayered hexagonal boron nitride and graphene: The effects of directionality, thickness, and sliding constraints. *Phys. Rev. Lett.* 114, (2015), 096101.
- [37] E. Mostaani, N.D. Drummond, V.I. Fal'ko, Quantum Monte Carlo Calculations of the Binding Energy of Bilayer Graphene. *Phys. Rev. Lett.* 115 (2015), 115501.
- [38] W. Wang, Y. Zhang, T. Sun, Y.B. Wang, On the nature of the stacking interaction between two graphene layers. *Chameical Physics Letters*, 620 (2015) 46-49.

- [39] G. Levita, E. Molinari, T. Polcar, and M.C. Righi. First-principles comparative study on the interlayer adhesion and shear strength of transition-metal dichalcogenides and graphene. *Phys. Rev. B* 92, 085434;2015.
- [40] T. Gould, Z. Liu, J.Z. Liu, J.F. Dobson, Q. Zheng, S. Lebegue. Binding and interlayer force in the near-contact region of two graphite slabs: Experiment and theory. *J. Chem. Phys.* 139, 224704;2013.
- [41] V.V. Gobre and A. Tkatchenko. Scaling laws for van der Waals interactions in nanostructured materials. *Nat. Commun.* 4 (2013) 1-6.
- [42] A. Ambrosetti, A.M. Reilly, R.A. DiStasio and A. Tkatchenko. *J. Chem. Phys.* 140 (2014) 18A508.
- [43] C.R.C. Rego, L.N. Oliveira, P. Tereshchuk, J.L.F. Da Silva. Comparative study of van der Waals corrections to the bulk properties of graphite. *J. Phys.: Condens. Matter*, 27, 415502; 2015.
- [44] Z. Lu, M.L. Dunn, van der Waals adhesion of graphene membranes, *J. Appl. Phys.* 107, 044301 (2010).

3. In the response of technique issue #7, the authors declared that there is no chemical bond/dangling bond both at the edge and along the sliding plane of nanomesas based on the mild detachment through the *F-d* curves. This is not convinced without any characterizations of the sample and solid evidence. Especially, the etching procedure would also introduce significant charge pollution in the nanomesas, which is not considered, neither.

As mentioned in the main text and supplementary information, we exploited four different techniques to characterize the crystal structures of 2D materials for any defects and/or contaminations/charge pollutions at the interface: (1) scanning tunneling microscopy (please see top panel of Figs. 1b and c and also Fig. S2a); (2) atomic force microscopy (please see top panel of Fig. 1b and Figs. S2b and S2c); (3) shear force-displacement measurements (please see Fig. 1c); and (4) x-ray photoelectron spectroscopy (please see Fig. S10). All four techniques confirmed an atomically flat and defect-free surface at the separation plane of 2D crystals.

In addition, Gongyang et al. [45] used a micro-force sensing probe to study the effect of chemical/physical bonds (induced by the reactive ion etching at the edge of $4\ \mu\text{m}\times 4\ \mu\text{m}$ graphite mesas) on the IAE of graphite through a direct shearing technique. They showed that the shear force in the presence of chemical/physical bonds at the edge ($\sim 19.2\ \mu\text{N}$, equivalent to $4.80\ \text{J}/\text{m}^2$) is an order of magnitude larger than that in the absence of chemical/physical bonds ($\sim 1.48\ \mu\text{N}$, equivalent to $0.37\ \text{J}/\text{m}^2$). From Figs. 1c and 1e in the main text, the IAE value of graphite crystal obtained by our shear force measurements is consistently less than $0.361\pm 0.014\ \text{J}/\text{m}^2$, indicating that at least the edge of the most bottom layer of the tip-attached nanomesas (where the sliding/separation takes place) is unlikely to be functionalized due to the etching process. Moreover, Gongyang et al. [45] showed that the effect of chemical/physical bonds on the shear force can be eliminated by annealing the graphite mesas at $150\ ^\circ\text{C}$. This, coupled with the fact that our measured cohesion energy at the intact homointerfaces is independent of the annealing temperatures (gray circles in Fig. 2a) further confirms that the chemical/physical bonds near/at the edge of nanomesas have no appreciable effect on our measurements even at room temperature.

To make it clearer, we totally changed **Supplementary Information, Section S2.6. Possible edge functionalization of 2D crystals, Page 11**, as follows

Section S2.6. Possible edge functionalization of 2D crystals

Among all our IAE measurements at 2D crystal/2D crystal and 2D crystal/ SiO_x interfaces, only $\sim 5\%$ of *F-d* curves at the G/ SiO_x interface exhibit the possible short-range bond at the edge of the G nanomesa. Although 2D crystals possess intrinsic active edge sites, the etching process of nanomesas could also

functionalize or chemically modify their edge. Gongyang et al. [52] used a micro-force sensing probe to study the effect of chemical bonds (induced by the reactive ion etching at the edge of $4\ \mu\text{m}\times 4\ \mu\text{m}$ graphite mesas) on the cohesion energy of G/G through a direct shear force technique. They showed that the cohesion energy in the presence of chemical bonds at the edge ($\sim 4.80\ \text{J/m}^2$) is an order of magnitude larger than that in the absence of chemical/physical bonds ($\sim 0.37\ \text{J/m}^2$). From **Figs. 1c** and **1e** in the main text, the cohesion energy of G crystal obtained by our shear force measurements is consistently less than $0.361\pm 0.014\ \text{J/m}^2$, confirming no chemical bond/dangling bond both at the edge and along the sliding plane of G nanomesas. Similarly, the level of cohesion energy in hBN and MoS₂ crystals dictates no chemical bond at their edge. Moreover, Gongyang et al. [52] showed that the effect of chemical bonds on the shear force can be eliminated by annealing the G micromesas at 150 °C. This, coupled with the fact that our measured cohesion energy at the intact homointerfaces is independent of the annealing temperatures (gray circles in **Fig. 2a**) further confirms that the chemical bonds near/at the edge of nanomesas have no appreciable effect on our measurements even at room temperature. As such, we believe that at least the edge of the most bottom layer of the tip-attached nanomesas (where the sliding/separation takes place) is unlikely to be functionalized due to the etching process, and, therefore, the edge functionalization has no contribution to the overall IAE measurements. Moreover, the observation of chemical bonds at the edge of G nanomesa in some *F-d* curves of G/SiO_x heterostructures can be attributed to the intrinsic active edge sites in the most bottom layer of the G crystal tip rather than any possible edge functionalization due to the etching process of G nanomesas.

[45] Gongyang et al., Eliminating delamination of graphite sliding on diamond-like carbon. *Carbon* 132 (2018) 444-450.

4. The phrases 2D material and van der Waals heterostructure in the title have the same meaning, thus, the title should be revised as “Direct Measurements of Interfacial Adhesion in 2D Materials in Ambient Air”. Especially as the recommended measurement in vacuum has not conducted.

We thank the reviewer’s suggestion. We have added “in Ambient Air” to our title. “2D materials” term is commonly used for 2D single crystals (e.g., G, hBN, MoS₂, etc) while vdW heterostructures, which are created from stacking different 2D single crystals, appear independently in the title of many papers (please see, for example, K. S. Novoselov et al. “2D materials and van der Waals heterostructures”. *Science*, 2016:353,6298). Since we have studied the IAE of both naturally-formed 2D materials and man-made vdW heterostructures, we decided to choose the title as such to make sure that the reader can find in our work the IAE information for both types of layered materials.

In addition, we modified our abstract to better reflect the reviewer’s comment about our ambient-air measurements, as follows:

Abstract, Page 1, 3rd Sentence

We use an atomic force microscopy (AFM) technique to report precise adhesion measurements in ambient air through well-defined interactions of AFM tip-attached 2D crystal nanomesas with 2D crystal and SiO_x substrates.

5. The abstract should be rewritten to include main results and conclusions, rather than the present one simply describing what work did but lack of necessary information about what achieved for the readers.

We appreciate the reviewer's valuable suggestion and have re-written our abstract to better reflect what was achieved. In this work we have quantified, for the first time, the effect of airborne contaminants, pre-cooling treatments, thermal annealing on the IAE of similar/dissimilar vdW heterostructures and 2D crystal/SiO_x heterostructures. As evident from different sections of the main text, there are many different findings with a lot of detailed discussions about how each of the aforementioned parameters influence the IAE at G/G, hBN/hBN, MoS₂/MoS₂, G/hBN, MoS₂/G, MoS₂/hBN, G/SiO_x, hBN/SiO_x and MoS₂/SiO_x interfaces. While it is not possible to squeeze all quantitative information into Abstract due to the 150-words limit and the journal requirement to provide a general introduction to the topic as part of the Abstract, we make qualitative statements of the achievements in our Abstract.

Following the reviewer's comment, we modified Abstract to qualitatively reflect our key findings as follows:

Abstract

Interfacial adhesion energy is a fundamental property of two-dimensional (2D) layered materials and van der Waals (vdW) heterostructures due to their intrinsic ultrahigh surface to volume ratio, making adhesion forces extremely strong in many processes related to fabrication, integration and performance of devices incorporating 2D crystals. However, direct quantitative characterization of adhesion behavior of fresh and aged homo/heterointerfaces at nanoscale has remained elusive. We use an atomic force microscopy (AFM) technique to report precise adhesion measurements in ambient air through well-defined interactions of AFM tip-attached 2D crystal nanomesas with 2D crystal and SiO_x substrates. We quantify how different levels of short-range dispersive and long-range electrostatic interactions respond to airborne contaminants and humidity upon thermal annealing. We show that a simple but very effective precooling treatment can protect 2D crystal substrates against the airborne contaminants and thus boost the adhesion level at the interface of similar and dissimilar vdW heterostructures. Our combined experimental and computational analysis also reveals a distinctive interfacial behavior in transition metal dichalcogenides and graphite/SiO_x heterostructures beyond the widely accepted vdW interaction.

6. The light blue-shaded area in Fig. 2b represents the integral of the retraction force as a function of the piezo displacement; however, the IAE means the integral of the adhesion force as a function of distance between the tip and substrate. Thus, the method for calculating IAE described in the main text should be explained in detail.

Following the reviewer's comment, we have added the detail in Methods and in Supplementary Information (see next page).

In addition, we would like to provide some discussions here. Butt et al. [46] provided a comprehensive review paper entitled "Force measurements with the atomic force microscope: Technique, interpretation and applications" with a detailed discussion about the calculation of the adhesion energy. In particular, *Section 3.2. Difference between approach and retraction* and *Fig. 13* of this paper specifically discuss about the calculation of the interfacial adhesion energy (please see *Eq. 3.2* and relevant discussions on *page 34* of this paper) using the area under the $F-Z_p$ curve (where Z_p is the piezo displacement) rather than the area under the $F-D$ curve (where $D (=Z_p - Z_c)$ is the tip-sample separation and Z_c is the cantilever deflection), which is quite consistent with our

calculations. The reason behind using the $F-Z_p$ curve is that we aim at measuring the internal adhesion energy consumed/dissipated at the interface during the tip/sample separation process. To do so, the external energy required to overcome the internal adhesion energy at the interface is presumed to be equal to the bending energy stored in the cantilever (i.e., the area under the $F-Z_p$ curve). Moreover, this method is very well-established and widely used in literature to report the interfacial adhesion energy using the conventional static force microscopy. For instance, we refer the respected reviewer to the work of Zhang et al. *Science*, 2017:356;434-437 (and also page 4 of their supplementary materials) who used the same method to study the interfacial vdW attraction between rutile TiO₂ nanocrystals. Also, please see the following paper, “Temperature effects on the friction characteristics of graphene” by Zhang et al, *Appl Phys Lett* (2015) as another useful resource.

We have elaborated the methods that we used to calculate the IAE from both retraction force-displacement and shear force-displacement curves in Methods (**Force-displacement measurements**) and in Supplementary Information (**Section S2.1. Calculation of IAE from $F-d$ curves**).

Force-displacement measurements

All retraction $F-d$ curves between the 2D crystal tips and the untreated/precooling-treated substrates were obtained under controlled ambient conditions in the near-equilibrium regime which was achieved by an ultralow noise floor of less than 0.3 Å, an ultralow noise AFM controller with the Z scanner’s vertical resolution of better than 0.1 Å and also using a very slow (quasi-static) pulling rate of 1 nm/s. Very careful adjustment of the Z servo gain to suppress any possible oscillation of the Z scanner could further make the retraction measurements in the near-equilibrium regime possible (Section S2). In order to calculate the IAE per unit area (Γ , J/m²) from the recorded retraction-displacement curves, we integrate the retraction force as a function of the piezo displacement (light blue-shaded area in Figs. 1b and 1d), followed by dividing the resulting adhesion energy by the known contact area at the interface. However, in order to extract the IAE from the shear force-displacement curves, the interfacial adhesion force opposing new surface formation is first obtained as $F_a(x) = \Gamma[dA(x)/dx]$, where x represents the lateral displacement of the mobile section of the mesa with respect to the initial position, and $A(x)$ is the overlap area of the top and bottom sections of the mesa as a function of x . For a square mesa of width w and a circular mesa of diameter D , the maximum shear force in the shear $F-d$ curves which is required to initiate sliding (i.e., F_a at $x = 0$) can be related to the interfacial adhesion energy by Γw and ΓD , respectively (Section S2.1).

Section S2.1. Calculation of IAE from $F-d$ curves

While the adhesion forces were calculated by the calibrated spring constant and the measured deflection signal of the AFM probe, the IAE per unit area (Γ , J/m²) was calculated by integrating the retraction force as a function of the piezo displacement, followed by dividing the resulting adhesion energy by the known contact area at the interface. The reason behind using the area under the force-piezo displacement curve is that for stiff interfacial contacts, the external energy required to overcome the internal adhesion energy consumed/dissipated at the interface is presumed to be equal to the bending energy stored in the cantilever. In fact, we did not measure the deformation of the sample as 2D nanomesas and substrates are almost rigid. Rather, we measured the separation energy as the nanocracks start to form due to the localized nano delamination during the retraction process and propagate at the separation plane until the complete separation takes place [1]. As we already discussed in detail in the

main text, the crack propagation at the contact interface results in the separation not the deformation does. This can also be immediately confirmed by our MD simulations in the main text where a tiny distance ($\sim 0.3\text{nm}$) between two adjacent layers results in full separation at the interface with negligible deformation. It is also worth pointing out that the reported distance between the initiation of the separation and full separation of the tip-sample in **Fig. 1b** is the piezo displacement (e.g., $\sim 10\text{ nm}$ at hBN/hBN, $\sim 9\text{ nm}$ at G/G and $\sim 5\text{ nm}$ at MoS₂/ MoS₂ interfaces) rather than the interlayer distance between two adjacent 2D crystal layers and thus does not represent the distance of short-range vdW interaction at the tip-sample interface.

In order to extract the interfacial adhesion energy from the shear force-displacement curves, we first assumed a lateral shear force F_s being applied to a square mesa of width w or a circular mesa of diameter D , leading to the lateral displacement x of the upper section relative to the bottom section of the mesa and creation of new interface area $A(x)$. At the sliding interface, the total free energy may change by $U(x) = -\Gamma A(x)$. We next obtained the corresponding interfacial adhesion force opposing new surface formation as $F_a(x) = -dU(x)/dx = \Gamma dA(x)/dx$. For the circular/square mesa with the following new interface area

Circular mesa:
$$A(x) = \frac{D^2}{2} \left[\cos^{-1}\left(\frac{x}{D}\right) - \frac{x}{D} \sqrt{1 - \left(\frac{x}{D}\right)^2} \right]$$

Square mesa:
$$A(x) = w(w - x)$$

the corresponding interfacial adhesion forces can be written as

Circular mesa:
$$F_a(x) = -\Gamma D \sqrt{1 - \left(\frac{x}{D}\right)^2}$$

Square mesa:
$$F_a = -\Gamma w$$

By inspection, one can see that the maximum interfacial adhesion force required to initiate sliding the circular and square mesas can simply be given by ΓD and Γw , respectively.

[46] Butt et al., Force measurements with the atomic force microscope: Technique, interpretation and applications. Surface Science Reports,2005;59(1-6):1-152.

7. Figure 1c provides the shear force–lateral piezo displacement curves only with the range of piezo displacement from 0 to 12 nm, based on that, how did the authors calculate the value of interfacial adhesion energy?

As we discussed in Methods (please see “Force-displacement measurements”) and Supplementary Information (please see “Section S2.1. Calculation of IAE from F - d curves”), the IAE values from the shear force-displacement curves in Fig. 1c can be calculated by taking the maximum interfacial adhesion force (required to initiate sliding the nanomesa) multiplied by the width/diameter of square/circular nanomesa, and, therefore, are independent of the lateral piezo displacement. In order to extract the interfacial adhesion energy from the shear force-displacement curves, we first assumed a lateral shear force F_s being applied to a square mesa of width w or a circular mesa of diameter D , leading to the lateral displacement x of the upper section relative to the bottom section of the mesa and creation of new interface area $A(x)$. At the sliding interface, the total free energy may change by $U(x) = -\Gamma A(x)$. We next obtained the corresponding interfacial adhesion force

opposing new surface formation as $F_a(x) = -dU(x)/dx = \Gamma dA(x)/dx$. For the circular/square mesa with the following new interface area

Circular mesa:
$$A(x) = \frac{D^2}{2} \left[\cos^{-1} \left(\frac{x}{D} \right) - \frac{x}{D} \sqrt{1 - \left(\frac{x}{D} \right)^2} \right]$$

Square mesa:
$$A(x) = w(w - x)$$

the corresponding interfacial adhesion forces can be written as

Circular mesa:
$$F_a(x) = -\Gamma D \sqrt{1 - \left(\frac{x}{D} \right)^2}$$

Square mesa:
$$F_a = -\Gamma w$$

By inspection, one can see that the maximum interfacial adhesion force required to initiate sliding the circular and square mesas can simply be given by ΓD and Γw , respectively.

8. In Fig. 3c, it is unreasonable that the calculated curves do not coincide with each other when the relative displacement is larger than 0.2 nm, because both curves represent the same van der Waals interaction between the graphene and SiO_x.

Although we defined the same vdW interactions (LJ potential) for both G/SiO_x heterostructures, one interaction is defined purely based on vdW forces (purple curve in Fig. 3c) and the other one is a combined action of vdW forces and short-range forces (magenta curve in Fig. 3c). Therefore, different interfacial interactions and, as a result, different relative distances between the most bottom graphene flake and SiO_x in each case dictate different interfacial adhesion forces up to 0.5 nm (not 0.2 nm) where the full separation takes place.

9. The reason why the relatively gradual reduction of the adhesion force was observed in the retraction force measurements (rather than a snap-back to zero force) as described in the revised manuscript (lines 101 ~103 in manuscript) is not positively explained except for the discussions of excluding the effect of tip-sample capillary forces. Besides, the red curve shown in Fig. 1d was measured in 70% humidity instead of 15%, which is not appropriate for the purpose of comparison.

Following the reviewer's comment, we added the following paragraph to "Discussion" section to further explain, from an interfacial fracture standpoint, the reason why the relatively gradual reduction of the adhesion force was observed in 2D crystals (Fig. 1b in the main text).

Discussion, Page 11

We already explained in Fig. 3a the origin of different trends in our F-d curves for the G/SiO_x heterostructure by means of interfacial fracture mechanics. Similarly, we believe that for the case of the relatively weak vdW-only interaction (e.g., hBN/hBN in Fig. 1b), both a smaller pull-off force and the smooth and slow propagation of nanocracks contribute to the relatively gradual reduction of the interfacial adhesion force. In contrast, faster crack propagation in the relatively stronger vdW-only interaction (e.g., G/G in Fig. 1b) which is triggered by a larger pull-off force, results in the abrupt force drop in the retraction curves immediately upon the initiation of the separation process. However, in the

case of suddenly broken contacts (e.g., MoS₂/MoS₂ in Fig. 1b and G/Cu in Fig. 1d), the more electron sharing at the interface, the larger pull-off force is required to initiate the interfacial fracture, thereby much faster nanocrack propagation at the beginning of the separation process causes a sudden break of the contact.

Regarding Fig. 1d, we aimed at comparing between the rupture force-displacement curve of capillary bridges and the retraction force-displacement curve of 2D crystals. To this end, higher humidity (70%) is required to form the nanomeniscus between the AFM tip and the SiO_x substrate. In other words, low humidity of 15% does not guarantee that the measured $F-d$ curve represents the pure rupture of capillary bridges with negligible contribution from the vdW interaction at the tip-substrate interface.

10. The authors demonstrate that $F-d$ curves upon tip approach display a small jump-to-contact force and suggest negligible effect of tip-sample capillary forces (revised manuscript, lines 103-106). Thus, the approach $F-d$ curve of Si/SiO_x measurement (red curve in Fig. 1d) should be exhibited for comparison.

For comparison purposes, the approach $F-d$ curve of Si/SiO_x must be presented at RH 15%, similar to the approach $F-d$ curve of hBN at RH 15% (gray curve in Fig. 1b). However, as we mentioned in response to the reviewer's comment #9, Fig. 1d aims at showing fundamental differences in the separation mechanism between 2D crystals at RH 15% and capillary bridges at RH 70%, and, as a result, adding the approach $F-d$ curve of Si/SiO_x at RH 15% makes Fig.1d more confusing for the reader. Moreover, using the approach $F-d$ curve of Si/SiO_x at RH 15% does not necessarily confirm the negligible effect of tip-sample capillary forces in 2D crystals. Rather, a small jump-to-contact force (in the order of few nN) at a small relative tip-sample distance (in the order of few nm) coupled with a dry contact-like shape of the approach curve can guarantee the negligible effect of tip-sample capillary forces in 2D crystals. In fact, the presence of long-range capillary forces does not allow the tip to (1) become as close to the sample as ~6nm (gray curve in Fig. 1b) and (2) suddenly jump to contact with the sample at a small jump-to-contact force of ~10 nN (gray curve in Fig. 1b).

To make it clearer, we modified the main text as follows:

Main text, Page 4, Paragraph 1, Lines 1-4:

However, the hydrophobic nature of 2D crystal nanomesas along with our $F-d$ approach curves which display a small jump-to-contact force of 8–12 nN at a small relative tip-sample distance of 5–6 nm (see, for instance, hBN/hBN approach curve in Fig. 1b) suggest dry contact at the interface with negligible effect of tip-sample capillary forces on the retraction curves.

11. All three materials in the experiment are multi-layer flakes, thus, the comparison of the bending stiffness between different materials (lines 376 ~380 in manuscript) are in conflict with the recent paper (Phys. Rev. Lett. 123, 116101, 2019), the following conclusion should be reconsidered.

Based on our discussions in the main text “Origin of distinctive interfacial adhesion behavior in G/SiO_x” on page 10-11, the bending stiffnesses of monolayer G, monolayer hBN and monolayer MoS₂ were considered to be 1.49eV [47], 1.34eV [47] and 11.7eV [48], respectively, which are both quantitatively and quantitatively consistent with those of 1.60 eV [49] (for monolayer G), 1.29 eV

[47] (for monolayer hBN) and 9.9 eV [50] (for monolayer MoS₂) used in the paper mentioned by the reviewer to calculate the bending stiffness of multilayer 2D materials (Phys. Rev. Lett. 123, 116101, 2019).

Moreover, in our work, the bending stiffness of bilayer G was considered to be 35.5 eV which was calculated by measuring the critical voltage for snap-through of pre-buckled graphene membranes [51]. This value also lies within the range of 3.15-110.4 eV predicted by Eq. 3 in the paper mentioned by the reviewer for the bilayer G (Phys. Rev. Lett. 123, 116101, 2019).

For the sake of completeness, we cited the aforementioned paper (Phys. Rev. Lett. 123, 116101, 2019) and also the following paper (Nature Communications, 6:8935, 2015) and modified the main text and the Supplementary Information, accordingly, as follows:

Main text, Page 5, Paragraph 1

We also note that, to the best of our knowledge, no IAE measurement on the hBN homointerface yet exists, while there is only one measured IAE value of 0.22 Jm⁻² at the MoS₂ homointerface using a nanomechanical cleavage technique [18], which is much lower than our values. Given that the bending stiffness of transition metal dichalcogenides is reported in the range of 10-16 eV [23], we believe that a very low bending stiffness value of 0.92 eV used in their calculations for the monolayer MoS₂ has resulted in such a low IAE value.

Section S7. Calculations of bending stiffness in 2D crystals

A direct measurement of in-plane elastic modulus of monolayer G (342±8 Nm⁻¹ [16]), bilayer G (645±16 Nm⁻¹ [16]), monolayer hBN (289±24 Nm⁻¹ [16]) and monolayer MoS₂ (180±60 Nm⁻¹ [17], 120±30 Nm⁻¹ [18]) was reported by AFM nanoindentation of suspended 2D crystal membranes. Also, the bending stiffness of monolayer G (1.49eV) [19], monolayer hBN (1.34eV) [19] and monolayer MoS₂ (11.7eV) [18] is obtained by first principles calculations, whose in-plane elastic modulus of monolayer 2D crystals is consistent with the aforementioned experimental values. In addition, the bending stiffness of bilayer G (35.5 eV) was calculated by measuring the critical voltage for snap-through of pre-buckled graphene membranes [20]. This value for the bilayer G also lies within the range of 3.15-110.4 eV predicted by a modified classical plate theory for the effective bending rigidity of multilayer graphene and 2D materials [53].

- [47] K. N. Kudin and G. E. Scuseria, "C₂F, BN, and C nanoshell elasticity from ab initio computations," *Physical Review B*, vol. 64, p. 235406, 2001.
- [48] R. C. Cooper, C. Lee, C. A. Marianetti, X. Wei, J. Hone and J. W. Kysar, "Nonlinear elastic behavior of two-dimensional molybdenum disulfide," *Physical Review B*, vol. 87, p. 035423, 2013.
- [49] D. B. Zhang et al. "Bending ultrathin graphene at the Margin of continuum mechanics," *Physical Review Letters*, vol. 106, p. 255503, 2011.
- [50] J. Zhao et al. "Two-dimensional membrane as elastic shell with proof on the folds revealed by three-dimensional atomic mapping," *Nature Communications*, vol. 19;6:8935, 2015.
- [51] N. Lindah, D. Midtvedt, J. Svensson, O. A. Nerushev, N. Lindvall, A. Isacson and E. E. B. Campbell, "Determination of the Bending Rigidity of Graphene via Electrostatic Actuation of Buckled Membranes," *Nano Letters*, vol. 12, no. 7, p. 3526–3531, 2012.

12. Did the separation take place across the thickness of G nanomesa for the data points ΓG/SiOx higher than ΓG/G in Fig. 3b?

Yes, it did. Fig. 3b shows that the interfacial adhesion between G/SiO_x is pressure dependent and could be weaker or stronger than the adhesion between G/G depending on the number of short-range chemical bonds formed at the G/SiO_x interface. When the adhesion of G/SiO_x is stronger than that of G/G, the separation takes place across the thickness of the nanomesa (i.e., between G/G, please see blue squares in Fig. 3b) and therefore we are not able to measure the strong adhesion at the G/SiO_x interface. In other words, we can only measure the adhesion of G/SiO_x weaker than that of G/G (red circles in Fig. 3b). The above discussion was reflected in the main text as follows:

Main text, Page 8, Paragraph 4:

To this end, a series of interfacial adhesion measurements over a pressure range of 0-10 MPa was conducted at the interface of G crystal tip/pre-annealed SiO_x substrate (top panel of Fig. 3b) and G crystal tip/pre-annealed G substrate (bottom panel of Fig. 3b). This setup only allowed us to study the interaction of G/SiO_x weaker than that between G/G (red circles in Fig. 3b), otherwise the separation takes place across the thickness of G nanomesa (blue squares in Fig. 3b).

13. The water layer between the Si-tip and substrate (Fig. 1d) should be called with corresponding terminology instead of “bubble” to avoid misleading the readers.

Following the reviewer’s comment, we replaced nanobubble/bridging nanobubble with water nanomeniscus/capillary nanobridge in Fig. 1d and throughout the manuscript.

14. Some writing errors need to be revised. Eg. in red (legend of Fig. 2d for Si/SiO_x); volume and page numbers for reference 19; the scale bar in Fig. 1c.

We thank the respected reviewer for pointing this out. We fixed these and also double-checked the whole manuscript.

Legend of Fig. 1d for SiO_x/Si:

Water bridge

Volume and page numbers for reference 19:

[19] B. Li, J. Yin, X. Liu, H. Wu, J. Li, X. Li and W. Guo, "Probing van der Waals interactions at two-dimensional heterointerfaces," *Nature Nanotechnology*, vol. 14, pp. 567-572, 2019.

Scale bar in Fig. 1c:

Scale bars indicate 50 nm.

15. The last, but most important issue is that the logic needs careful organization, the whole manuscript, each section, and the materials in the supplementary information.

The whole manuscript and Supplementary Information were read through one more time to make this work more organized and easier to follow for the reader, thanks to all three respected reviewers to greatly help us with their very constructive comments. We have also moved the tables and figures at the end in Supplementary Information to within the text where they are mentioned, so that it is easier to read.

Reviewer #3 (Remarks to the Author):

The authors have elaborated their responses to my comments raised in the first round of review. Basically, I'm satisfied with most of the replies. However, some issues remain in the revised manuscript that need further consideration by the authors.

1) Something mentioned in the authors' reply to my first comment are confusing. For example, tip-attached nanomesa, tip-attached 2D crystal, exfoliation of monolayer or few layers from tip-attached 2D crystal onto the sample, and contact spots are not easy to find what are referring to. The authors need to improve these statements and better add a figure for illustration.

We apologize for the confusion. To avoid any further confusion for the reader, we used “tip-attached 2D crystal nanomesa” and, as the shorter terminology, “tip-attached nanomesa” in Fig. 1a, legend of Fig. 1a and throughout the manuscript/Supplementary Information.

In particular, we further modified the supplementary information, Section S2 (which was added based on the reviewer's first comment in the first round of revision) as follows:

Section S2, Page 6, Paragraph 2:

In order to identify whether the $F-d$ curves are measured at the interface of the tip-attached 2D crystal nanomesa and the sample or within the thickness of the tip-attached 2D crystal nanomesa, we first measured the intrinsic cohesion energy of 2D crystals (Fig. 1e and gray circles in Fig. 2a), confirming that the cohesion energy across the 2D crystal nanomesa is larger than the interfacial adhesion energy at all 2D crystal tip-sample interfaces. We also observed larger pull-off forces at the intact interfaces compared to contaminated interfaces, well consistent with our reported IAE values. Therefore, the separation most likely takes place at the tip-sample interface rather than somewhere across the thickness of the tip-attached 2D crystal nanomesa. Moreover, for each tip-attached nanomesa, we formed all contacts with 1 μm interval spacing within the same distance from the heating line, allowing us to easily locate and scan all contact spots (using the non-contact AFM mode) for any possible exfoliation of monolayer or few layers of 2D crystal from tip-attached 2D crystal nanomesa onto the sample. For the contact spot with exfoliated mono/few-layer 2D crystal, the area under the corresponding $F-d$ curve was considered as the intrinsic cohesion energy rather than the interfacial adhesion energy at the tip-sample interface.

2) The authors are suggested to explain or discuss why the interfacial cohesion energy across the 2D crystal nanomesa is larger than the interfacial adhesion energy at all 2D crystal tip-sample interfaces.

We have performed all adhesion measurements in ambient air (rather than in high vacuum) to quantify the effect of airborne contaminants and humidity. Therefore, as we mentioned in the main text (please see, for instance, Page 5, Paragraph 2) after exposing the freshly exfoliated 2D crystal flakes to the ambient air, the IAE between similar vdW heterostructures (red circles in Fig. 2a) is consistently lower than their corresponding intrinsic value, mainly due to the possible adsorption of airborne contaminants (e.g., water and hydrocarbon molecules) onto the fresh surface of crystals, thereby reducing their overall free surface (Gibbs) energy.

Main text, Page 5, Paragraph 2

It is evident from the gray circles in **Fig. 2a** (and also Table S5) that, upon the attachment of nanomesas to the AFM tip after the thermal annealing, the measured cohesion energy at the intact homointerfaces is, within our experimental accuracy, independent of the annealing temperatures. However, after exposing the freshly exfoliated 2D crystal flakes to the ambient air, the IAE between similar vdW heterostructures (red circles in **Fig. 2a**) is consistently lower than their corresponding intrinsic value, mainly due to the possible adsorption of airborne contaminants (e.g., water and hydrocarbon molecules) onto the fresh surface of crystals, thereby reducing their overall free surface (Gibbs) energy. A 30% and 19% drop in the IAE of G/G and hBN/hBN, respectively, at room temperature suggests that G is more influenced by the airborne contaminants than hBN of similar lattice structure with only slightly larger (~1.8%) lattice constant.

3) Some discussion on the results need to be strengthened. For example, in figure 1d, the behavior of separation between G-G is distinct from that between G-Cu. It is known that G-Cu interaction are very weak, about two-fold of the G-G interaction. Then why the former case displays a sharp detachment but the latter case shows a viscoelastic detachment? Theoretical calculations (first-principles etc) would be helpful here for offering deep insight into the interfacial detachment.

The interfacial adhesion energy of G/Cu has been already measured by the double cantilever beam fracture mechanics test (1.05 J/m² [1] and 0.72 J/m² [2]) and the blister test (0.51 J/m² [3]), all showing a much stronger interaction at G/Cu than G/G, well consistent with our measurements of 1.02 J/m² for G/Cu and 0.33 J/m² for G/G. Moreover, as we mentioned in the last paragraph on Page 3, similar to the adhesion behavior at the MoS₂ homointerface, a sudden detachment of 2D crystal tips from metal substrates (e.g., Ni, Cu, Pt and Au) is observed with strong interfacial adhesion (see the *F-d* curve of G tip on the Cu substrate in Fig. 1d), suggesting that metal atoms share electrons with carbon atoms.

To make it clear, we added one paragraph to “Discussion” section to further explain, from an interfacial fracture standpoint, the reason why, unlike MoS₂ and Cu, the relatively gradual reduction of the adhesion force was observed only in hBN and G (Figs. 1b and 1d).

Discussion, Page 11

We already explained in **Fig. 3a** the origin of different trends in our *F-d* curves for the G/SiO_x heterostructure by means of interfacial fracture mechanics. Similarly, we believe that for the case of the relatively weak vdW-only interaction (e.g., hBN/hBN in **Fig. 1b**), both a smaller pull-off force and the smooth and slow propagation of nanocracks contribute to the relatively gradual reduction of the interfacial adhesion force. In contrast, faster crack propagation in the relatively stronger vdW-only interaction (e.g., G/G in **Fig. 1b**) which is triggered by a larger pull-off force, results in the abrupt force drop in the retraction curves immediately upon the initiation of the separation process. However, in the case of suddenly broken contacts (e.g., MoS₂/MoS₂ in **Fig. 1b** and G/Cu in **Fig. 1d**), the more electron sharing at the interface, the larger pull-off force is required to initiate the interfacial fracture, thereby much faster nanocrack propagation at the beginning of the separation process causes a sudden break of the contact.

[1] J. Seo et al. *Thin Solid Films*, 2015;584:170-175.

[2] T. Yoon et al. *Nano Letters*, 2012;12:1448-1452.

[3] Z. Cao et al. *Carbon*, 2014;69:390-400.

REVIEWER COMMENTS

Reviewer #3 (Remarks to the Author):

The authors have addressed my additional concerns and further improved the manuscript. Now, I can recommend publication of this manuscript.